# Rethinking the Expressive Power of GNNs via Graph Biconnectivity

**Bohang Zhang**[1][*]   **Shengjie Luo**[1][*]   **Liwei Wang**[1,2]   **Di He**[1]

[1]National Key Laboratory of General Artificial Intelligence,
 School of Intelligence Science and Technology, Peking University

[2]Center for Data Science, Peking University

`zhangbohang@pku.edu.cn`, `luosj@stu.pku.edu.cn`, `{wanglw,dihe}@pku.edu.cn`

## Abstract

Designing expressive Graph Neural Networks (GNNs) is a central topic in learning graph-structured data. While numerous approaches have been proposed to improve GNNs in terms of the Weisfeiler-Lehman (WL) test, generally there is still a lack of deep understanding of what additional power they can *systematically* and *provably* gain. In this paper, we take a fundamentally different perspective to study the expressive power of GNNs beyond the WL test. Specifically, we introduce a novel class of expressivity metrics via *graph biconnectivity* and highlight their importance in both theory and practice. As biconnectivity can be easily calculated using simple algorithms that have linear computational costs, it is natural to expect that popular GNNs can learn it easily as well. However, after a thorough review of prior GNN architectures, we surprisingly find that most of them are *not* expressive for *any* of these metrics. The only exception is the ESAN framework (Bevilacqua et al., 2022), for which we give a theoretical justification of its power. We proceed to introduce a principled and more efficient approach, called the Generalized Distance Weisfeiler-Lehman (GD-WL), which is provably expressive for all biconnectivity metrics. Practically, we show GD-WL can be implemented by a Transformer-like architecture that preserves expressiveness and enjoys full parallelizability. A set of experiments on both synthetic and real datasets demonstrates that our approach can consistently outperform prior GNN architectures.

## 1 Introduction

Graph neural networks (GNNs) have recently become the dominant approach for graph representation learning. Among numerous architectures, message-passing neural networks (MPNNs) are arguably the most popular design paradigm and have achieved great success in various fields (Gilmer et al., 2017; Hamilton et al., 2017; Kipf & Welling, 2017; Veličković et al., 2018). However, one major drawback of MPNNs lies in the limited expressiveness: as pointed out by Xu et al. (2019); Morris et al. (2019), they can never be more powerful than the classic 1-dimensional Weisfeiler-Lehman (1-WL) test in distinguishing non-isomorphic graphs (Weisfeiler & Leman, 1968). This inspired a variety of works to design provably more powerful GNNs that go beyond the 1-WL test.

One line of subsequent works aimed to propose GNNs that match the *higher-order* WL variants (Morris et al., 2019; 2020; Maron et al., 2019c;a; Geerts & Reutter, 2022). While being highly expressive, such an approach suffers from severe computation/memory costs. Moreover, there have been concerns about whether the achieved expressiveness is necessary for real-world tasks (Veličković, 2022). In light of this, other recent works sought to develop new GNN architectures with improved expressiveness while still keeping the message-passing framework for efficiency (Bouritsas et al., 2022; Bodnar et al., 2021b;a; Bevilacqua et al., 2022; Wijesinghe & Wang, 2022, and see Appendix A for more recent advances). However, most of these works mainly justify their expressiveness by giving *toy examples* where WL algorithms fail to distinguish, e.g., by focusing on regular graphs. On the theoretical side, it is quite unclear what additional power they can systematically and provably gain. More fundamentally, to the best of our knowledge (see Appendix D.1), there is still a lack of *principled* and *convincing* metrics beyond the WL hierarchy to formally measure the expressive power and to guide the design of provably better GNN architectures.

---

[*]Equal Contribution.

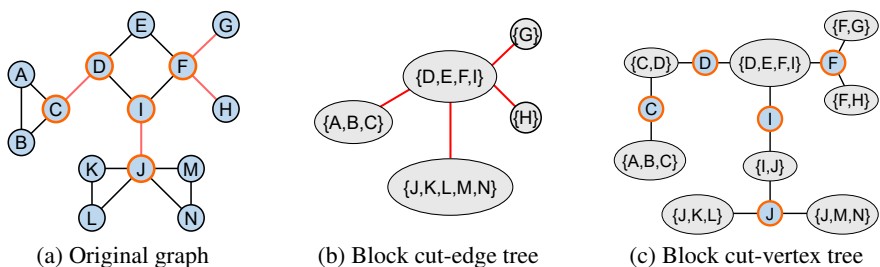

(a) Original graph      (b) Block cut-edge tree      (c) Block cut-vertex tree

Figure 1: An illustration of edge-biconnectivity and vertex-biconnectivity. Cut vertices/edges are outlined in bold red. Gray nodes in (b)/(c) are edge/vertex-biconnected components, respectively.

In this paper, we systematically study the problem of designing expressive GNNs from a novel perspective of *graph biconnectivity*. Biconnectivity has long been a central topic in graph theory (Bollobás, 1998). It comprises a series of important concepts such as cut vertex (articulation point), cut edge (bridge), biconnected component, and block cut tree (see Section 2 for formal definitions). Intuitively, biconnectivity provides a structural description of a graph by decomposing it into disjoint sub-components and linking them via cut vertices/edges to form a *tree* structure (cf. Figure 1(b,c)). As can be seen, biconnectivity purely captures the intrinsic structure of a graph.

The significance of graph biconnectivity can be reflected in various aspects. *Firstly*, from a theoretical point of view, it is a basic graph property and is linked to many fundamental topics in graph theory, ranging from path-related problems to network flow (Granot & Veinott Jr, 1985) and spanning trees (Kapoor & Ramesh, 1995), and is highly relevant to planar graph isomorphism (Hopcroft & Tarjan, 1972). *Secondly*, from a practical point of view, cut vertices/edges have substantial values in many real applications. For example, chemical reactions are highly related to edge-biconnectivity of the molecule graph, where the breakage of molecular bonds usually occurs at the cut edges and each biconnected component often remains unchanged after the reaction. As another example, social networks are related to vertex-biconnectivity, where cut vertices play an important role in linking between different groups of people (biconnected components). *Finally*, from a computational point of view, the problems related to biconnectivity (e.g., finding cut vertices/edges or constructing block cut trees) can all be efficiently solved using classic algorithms (Tarjan, 1972), with a computation complexity *equal to graph size* (which is the same as an MPNN). Therefore, one may naturally expect that popular GNNs should be able to learn all things related to biconnectivity without difficulty.

Unfortunately, we show this is not the case. After a thorough analysis of four classes of representative GNN architectures in literature (see Section 3.1), we find that surprisingly, none of them could even solve the *easiest* biconnectivity problem: to distinguish whether a graph has cut vertices/edges or not (corresponding to a graph-level binary classification). As a result, they obviously failed in the following harder tasks: (i) identifying all cut vertices (a node-level task); (ii) identifying all cut edges (an edge-level task); (iii) the graph-level task for general biconnectivity problems, e.g., distinguishing a pair of graphs that have non-isomorphic block cut trees. This raises the following question: *can we design GNNs with provable expressiveness for biconnectivity problems?*

We first give an *affirmative* answer to the above question. By conducting a deep analysis of the recently proposed Equivariant Subgraph Aggregation Network (ESAN) (Bevilacqua et al., 2022), we prove that the DSS-WL algorithm with *node marking* policy can precisely identify both cut vertices and cut edges. This provides a new understanding as well as a strong theoretical justification for the expressive power of DSS-WL and its recent extensions (Frasca et al., 2022). Furthermore, we give a fine-grained analysis of several key factors in the framework, such as the graph generation policy and the aggregation scheme, by showing that *neither* (i) the ego-network policy without marking *nor* (ii) a variant of the weaker DS-WL algorithm can identify cut vertices.

However, GNNs designed based on DSS-WL are usually sophisticated and suffer from high computation/memory costs. The **main contribution** in this paper is then to give a *principled* and *efficient* way to design GNNs that are expressive for biconnectivity problems. Targeting this question, we restart from the classic 1-WL algorithm and figure out a major weakness in distinguishing biconnectivity: the lack of *distance information* between nodes. Indeed, the importance of distance information is theoretically justified in our proof for analyzing the expressive power of DSS-WL. To this end, we introduce a novel color refinement framework, formalized as Generalized Distance Weisfeiler-Lehman (GD-WL), by directly encoding a general distance metric into the WL aggrega-

Table 1: Summary of theoretical results on the expressive power of different GNN models for various biconnectivity problems. We also list the time/space complexity (per WL iteration) for each WL algorithm, where $n$ and $m$ are the number of nodes and edges of a graph, respectively.

| Model | Section 3.1 | | | | Section 3.2 | | Section 4 | | |
|---|---|---|---|---|---|---|---|---|---|
| | MPNN | GSN | CWN | GraphSNN | ESAN | | Ours | | 3-IGN |
| WL variant | 1-WL | SC-WL | CWL | OS-WL | DSS-WL | DS-WL | SPD-WL | GD-WL | 2-FWL |
| Cut vertex | ✗ | ✗ | ✗ | ✗ | ✓ | ✗ | ✗ | ✓ | ✓ |
| Cut edge | ✗ | ✗ | ✗ | ✗ | ✓ | Unknown | ✓ | ✓ | ✓ |
| BCVTree | ✗ | ✗ | ✗ | ✗ | ✓ | Unknown | ✗ | ✓ | ✓ |
| BCETree | ✗ | ✗ | ✗ | ✗ | ✓ | Unknown | ✓ | ✓ | ✓ |
| Ref. Theorem | - | 3.1 | C.12 | C.13 | 3.2 | C.16 | 4.1 | 4.2, 4.3 | 4.6 |
| Time | $n+m$ | $n+m$ | - | $n+m$ | $n(n+m)$ | $n(n+m)$ | $n^2$ | $n^2$ | $n^3$ |
| Space[1] | $n$ | $n$ | - | $n$ | $n^2$ | $n$ | $n$ | $n$ | $n^2$ |

tion procedure. We first prove that as a special case, the Shortest Path Distance WL (SPD-WL) is expressive for all edge-biconnectivity problems, thus providing a novel understanding of its empirical success. However, it still cannot identify cut vertices. We further suggest an alternative called the Resistance Distance WL (RD-WL) for vertex-biconnectivity. To sum up, all biconnectivity problems can be provably solved within our proposed GD-WL framework.

**Practical Implementation**. The main advantage of GD-WL lies in its simplicity, efficiency and *parallelizability*. We show it can be easily implemented using a Transformer-like architecture by injecting the distance into Multi-head Attention (Vaswani et al., 2017), similar to Ying et al. (2021a). Importantly, we prove that the resulting Graph Transformer (called Graphormer-GD) is *as expressive as* GD-WL. This offers strong theoretical insights into the power and limits of Graph Transformers. Empirically, we show Graphormer-GD not only achieves perfect accuracy in detecting cut vertices and cut edges, but also outperforms prior GNN achitectures on popular benchmark datasets.

## 2 PRELIMINARY

**Notations**. We use $\{\ \}$ to denote sets and use $\{\!\{\ \}\!\}$ to denote multisets. The cardinality of (multi)set $\mathcal{S}$ is denoted as $|\mathcal{S}|$. The index set is denoted as $[n] := \{1, \cdots, n\}$. Throughout this paper, we consider simple undirected graphs $G = (\mathcal{V}, \mathcal{E})$ with no repeated edges or self-loops. Therefore, each edge $\{u, v\} \in \mathcal{E}$ can be expressed as a set of two elements. For a node $u \in \mathcal{V}$, denote its *neighbors* as $\mathcal{N}_G(u) := \{v \in \mathcal{V} : \{u, v\} \in \mathcal{E}\}$ and denote its *degree* as $\deg_G(u) := |\mathcal{N}_G(u)|$. A *path* $P = (u_0, \cdots, u_d)$ is a tuple of nodes satisfying $\{u_{i-1}, u_i\} \in \mathcal{E}$ for all $i \in [d]$, and its length is denoted as $|P| := d$. A path $P$ is said to be *simple* if it does not go through a node more than once, i.e. $u_i \neq u_j$ for $i \neq j$. The shortest path distance between two nodes $u$ and $v$ is denoted to be $\mathrm{dis}_G(u, v) := \min\{|P| : P \text{ is a path from } u \text{ to } v\}$. The *induced subgraph* with vertex subset $\mathcal{S} \subset \mathcal{V}$ is defined as $G[\mathcal{S}] = (\mathcal{S}, \mathcal{E}_\mathcal{S})$ where $\mathcal{E}_\mathcal{S} := \{\{u, v\} \in \mathcal{E} : u, v \in \mathcal{S}\}$.

We next introduce the concepts of connectivity, vertex-biconnectivity and edge-biconnectivity.

**Definition 2.1.** (**Connectivity**) A graph $G$ is *connected* if for any two nodes $u, v \in \mathcal{V}$, there is a path from $u$ to $v$. A vertex set $\mathcal{S} \subset \mathcal{V}$ is a *connected component* of $G$ if $G[\mathcal{S}]$ is connected and for any proper superset $\mathcal{T} \supsetneq \mathcal{S}$, $G[\mathcal{T}]$ is disconnected. Denote $\mathrm{CC}(G)$ as the set of all connected components, then $\mathrm{CC}(G)$ forms a *partition* of the vertex set $\mathcal{V}$. Clearly, $G$ is connected iff $|\mathrm{CC}(G)| = 1$.

**Definition 2.2.** (**Biconnectivity**) A node $v \in \mathcal{V}$ is a *cut vertex* (or *articulation point*) of $G$ if removing $v$ increases the number of connected components, i.e., $|\mathrm{CC}(G[\mathcal{V} \backslash \{v\}])| > |\mathrm{CC}(G)|$. A graph is *vertex-biconnected* if it is connected and does not have any cut vertex. A vertex set $\mathcal{S} \subset \mathcal{V}$ is a *vertex-biconnected component* of $G$ if $G[\mathcal{S}]$ is vertex-biconnected and for any proper superset $\mathcal{T} \supsetneq \mathcal{S}$, $G[\mathcal{T}]$ is not vertex-biconnected. We can similarly define the concepts of *cut edge* (or *bridge*) and *edge-biconnected component* (we omit them for brevity). Finally, denote $\mathrm{BCC}^{\mathrm{V}}(G)$ (resp. $\mathrm{BCC}^{\mathrm{E}}(G)$) as the set of all vertex-biconnected (resp. edge-biconnected) components.

Two non-adjacent nodes $u, v \in \mathcal{V}$ are in the same vertex-biconnected component iff there are two paths from $u$ to $v$ that do not intersect (except at endpoints). Two nodes $u, v$ are in the same edge-biconnected component iff there are two paths from $u$ to $v$ that do not share an edge. On the other hand, if two nodes are in different vertex/edge-biconnected components, any path between them

---

[1]The space complexity of WL algorithms may differ from the corresponding GNN models in training, e.g., for DS-WL and GD-WL, due to the need to store intermediate results for back-propagation.

must go through some cut vertex/edge. Therefore, cut vertices/edges can be regarded as "hubs" in a graph that link different subgraphs into a whole. Furthermore, the link between cut vertices/edges and biconnected components forms a *tree* structure, which are called the *block cut tree* (cf. Figure 1).

**Definition 2.3.** (**Block cut-edge tree**) The block cut-edge tree of graph $G = (\mathcal{V}, \mathcal{E})$ is defined as follows: $\mathrm{BCETree}(G) := (\mathrm{BCC}^{\mathrm{E}}(G), \mathcal{E}^{\mathrm{E}})$, where

$$\mathcal{E}^{\mathrm{E}} := \left\{ \{\mathcal{S}_1, \mathcal{S}_2\} : \mathcal{S}_1, \mathcal{S}_2 \in \mathrm{BCC}^{\mathrm{E}}(G), \exists u \in \mathcal{S}_1, v \in \mathcal{S}_2, \text{s.t. } \{u, v\} \in \mathcal{E} \right\}.$$

**Definition 2.4.** (**Block cut-vertex tree**) The block cut-vertex tree of graph $G = (\mathcal{V}, \mathcal{E})$ is defined as follows: $\mathrm{BCVTree}(G) := (\mathrm{BCC}^{\mathrm{V}}(G) \cup \mathcal{V}^{\mathrm{Cut}}, \mathcal{E}^{\mathrm{V}})$, where $\mathcal{V}^{\mathrm{Cut}} \subset \mathcal{V}$ is the set containing all cut vertices of $G$ and

$$\mathcal{E}^{\mathrm{V}} := \left\{ \{\mathcal{S}, v\} : \mathcal{S} \in \mathrm{BCC}^{\mathrm{V}}(G), v \in \mathcal{V}^{\mathrm{Cut}}, v \in \mathcal{S} \right\}.$$

The following theorem shows that all concepts related to biconnectivity can be efficiently computed.

**Theorem 2.5.** *(Tarjan, 1972) The problems related to biconnectivity, including identifying all cut vertices/edges, finding all biconnected components ($\mathrm{BCC}^{\mathrm{V}}(G)$ and $\mathrm{BCC}^{\mathrm{E}}(G)$), and building block cut trees ($\mathrm{BCVTree}(G)$ and $\mathrm{BCETree}(G)$), can all be solved using the Depth-First Search algorithm, within a computation complexity linear in the graph size, i.e. $\Theta(|\mathcal{V}| + |\mathcal{E}|)$.*

**Isomorphism and color refinement algorithms**. Two graphs $G = (\mathcal{V}_G, \mathcal{E}_G)$ and $H = (\mathcal{V}_H, \mathcal{E}_H)$ are *isomorphic* (denoted as $G \simeq H$) if there is an *isomorphism* (bijective mapping) $f : \mathcal{V}_G \to \mathcal{V}_H$ such that for any nodes $u, v \in \mathcal{V}_G$, $\{u, v\} \in \mathcal{E}_G$ iff $\{f(u), f(v)\} \in \mathcal{E}_H$. A color refinement algorithm is an algorithm that outputs a *color mapping* $\chi_G : \mathcal{V}_G \to \mathcal{C}$ when taking graph $G$ as input, where $\mathcal{C}$ is called the *color set*. A valid color refinement algorithm must preserve *invariance* under isomorphism, i.e., $\chi_G(u) = \chi_H(f(u))$ for isomorphism $f$ and node $u \in \mathcal{V}_G$. As a result, it can be used as a necessary test for graph isomorphism by comparing the multisets $\{\!\{\chi_G(u) : u \in \mathcal{V}_G\}\!\}$ and $\{\!\{\chi_H(u) : u \in \mathcal{V}_H\}\!\}$, which we call the *graph representations*. Similarly, $\chi_G(u)$ can be seen as the *node feature* of $u \in \mathcal{V}_G$, and $\{\!\{\chi_G(u), \chi_G(v)\}\!\}$ corresponds to the edge feature of $\{u, v\} \in \mathcal{E}_G$. All algorithms studied in this paper fit the color refinement framework, and please refer to Appendix B for a precise description of several representatives (e.g., the classic 1-WL and $k$-FWL algorithms).

**Problem setup**. This paper focuses on the following three types of problems with increasing difficulties. *Firstly*, we say a color refinement algorithm can distinguish whether a graph is vertex/edge-biconnected, if for any graphs $G, H$ where $G$ is vertex/edge-biconnected but $H$ is not, their graph representations are different, i.e. $\{\!\{\chi_G(u) : u \in \mathcal{V}_G\}\!\} \neq \{\!\{\chi_H(u) : u \in \mathcal{V}_H\}\!\}$. *Secondly*, we say a color refinement algorithm can identify cut vertices if for any graphs $G, H$ and nodes $u \in \mathcal{V}_G, v \in \mathcal{V}_H$ where $u$ is a cut vertex but $v$ is not, their node features are different, i.e. $\chi_G(u) \neq \chi_H(v)$. Similarly, it can identify cut edges if for any $\{u, v\} \in \mathcal{E}_G$ and $\{w, x\} \in \mathcal{E}_H$ where $\{u, v\}$ is a cut edge but $\{w, x\}$ is not, their edge features are different, i.e. $\{\!\{\chi_G(u), \chi_G(v)\}\!\} \neq \{\!\{\chi_H(w), \chi_H(x)\}\!\}$. *Finally*, we say a color refinement algorithm can distinguish block cut-vertex/edge trees, if for any graphs $G, H$ satisfying $\mathrm{BCVTree}(G) \not\simeq \mathrm{BCVTree}(H)$ (or $\mathrm{BCETree}(G) \not\simeq \mathrm{BCETree}(H)$), their graph representations are different, i.e. $\{\!\{\chi_G(u) : u \in \mathcal{V}_G\}\!\} \neq \{\!\{\chi_H(u) : u \in \mathcal{V}_H\}\!\}$.

## 3 INVESTIGATING KNOWN GNN ARCHITECTURES VIA BICONNECTIVITY

In this section, we provide a comprehensive investigation of popular GNN variants in literature, including the classic MPNNs, Graph Substructure Networks (GSN) (Bouritsas et al., 2022) and its variant (Barceló et al., 2021), GNN with lifting transformations (MPSN and CWN) (Bodnar et al., 2021b;a), GraphSNN (Wijesinghe & Wang, 2022), and Subgraph GNNs (e.g., Bevilacqua et al. (2022)). Surprisingly, we find most of these works are not expressive for *any* biconnectivity problems listed above. The only exceptions are the ESAN (Bevilacqua et al., 2022) and several variants, where we give a rigorous justification of their expressive power for both vertex/edge-biconnectivity.

### 3.1 COUNTEREXAMPLES

**1-WL/MPNNs**. We first consider the classic 1-WL. We provide two principled class of counterexamples which are formally defined in Examples C.9 and C.10, with a few special cases illustrated in Figure 2. For each pair of graphs in Figure 2, the color of each node is drawn according to the 1-WL color mapping. It can be seen that the two graph representations are the same. Therefore, 1-WL cannot distinguish any biconnectivity problem listed in Section 2.

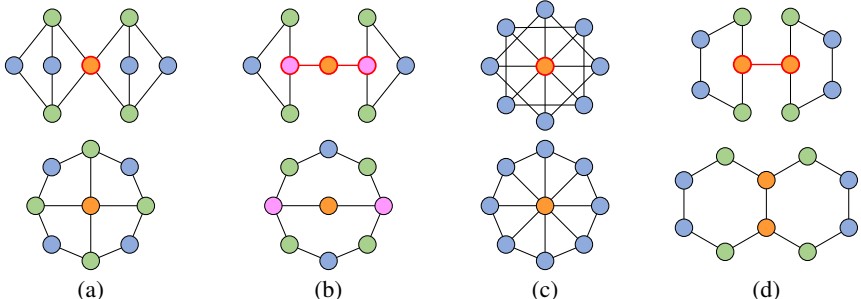

Figure 2: Illustration of four representative counterexamples (see Examples C.9 and C.10 for general definitions). Graphs in the first row have cut vertices (outlined in bold red) and some also have cut edges (denoted as red lines), while graphs in the second row do not have any cut vertex or cut edge.

**Substructure Counting WL/GSN**. Bouritsas et al. (2022) developed a principled approach to boost the expressiveness of MPNNs by incorporating *substructure counts* into node features or the 1-WL aggregation procedure. The resulting algorithm, which we call the SC-WL, is detailed in Appendix B.3. However, we show no matter what sub-structures are used, the corresponding GSN still cannot solve any biconnectivity problem listed in Section 2. We give a proof in Appendix C.2 for the *general* case that allows arbitrary substructures, based on Examples C.9 and C.10. We also point out that our negative result applies to the similar GNN variant in Barceló et al. (2021).

**Theorem 3.1.** *Let $\mathcal{H} = \{H_1, \cdots, H_k\}$, $H_i = (\mathcal{V}_i, \mathcal{E}_i)$ be any set of connected graphs and denote $n = \max_{i \in [k]} |\mathcal{V}_i|$. Then SC-WL (Appendix B.3) using the substructure set $\mathcal{H}$ cannot solve any vertex/edge-biconnectivity problem listed in Section 2. Moreover, there exist counterexample graphs whose sizes (both in terms of vertices and edges) are $O(n)$.*

**GNNs with lifting transformations (MPSN/CWN)**. Bodnar et al. (2021b;a) considered another approach to design powerful GNNs by using graph *lifting* transformations. In a nutshell, these approaches exploit higher-order graph structures such as cliques and cycles to design new WL aggregation procedures. Unfortunately, we show the resulting algorithms, called the SWL and CWL, still cannot solve any biconnectivity problem. Please see Appendix C.2 (Proposition C.12) for details.

**Other GNN variants**. In Appendix C.2, we discuss other recently proposed GNNs, such as Graph-SNN (Wijesinghe & Wang, 2022), GNN-AK (Zhao et al., 2022), and NGNN (Zhang & Li, 2021). Due to space limit, we defer the corresponding negative results in Propositions C.13, C.15 and C.16.

## 3.2 PROVABLE EXPRESSIVENESS OF ESAN AND DSS-WL

We next switch our attention to a new type of GNN framework proposed in Bevilacqua et al. (2022), called the Equivariant Subgraph Aggregation Networks (ESAN). The central algorithm in EASN is called the DSS-WL. Given a graph $G$, DSS-WL first generates a bag of vertex-shared (sub)graphs $\mathcal{B}_G^\pi = \{\!\{G_1, \cdots, G_m\}\!\}$ according to a graph generation policy $\pi$. Then in each iteration $t$, the algorithm refines the color of each node $v$ in each subgraph $G_i$ by jointly aggregating its neighboring colors in the own subgraph and across all subgraphs. The aggregation formula can be written as:

$$\chi_{G_i}^t(v) := \text{hash}\left(\chi_{G_i}^{t-1}(v), \{\!\{\chi_{G_i}^{t-1}(u) : u \in \mathcal{N}_{G_i}(v)\}\!\}, \chi_G^{t-1}(v), \{\!\{\chi_G^{t-1}(u) : u \in \mathcal{N}_G(v)\}\!\}\right), \quad (1)$$

$$\chi_G^t(v) := \text{hash}\left(\{\!\{\chi_{G_i}^t(v) : i \in [m]\}\!\}\right), \quad (2)$$

where hash is a perfect hash function. DSS-WL terminates when $\chi_G^t$ induces a stable vertex partition. In this paper, we consider *node-based* graph generation policies, for which each subgraph is associated to a specific node, i.e. $\mathcal{B}_G^\pi = \{\!\{G_v : v \in \mathcal{V}\}\!\}$. Some popular choices are node deletion $\pi_{\text{ND}}$, node marking $\pi_{\text{NM}}$, $k$-ego-network $\pi_{\text{EGO}(k)}$, and its node marking version $\pi_{\text{EGOM}(k)}$. A full description of DSS-WL as well as different policies can be found in Appendix B.4 (Algorithm 3).

A fundamental question regarding DSS-WL is how expressive it is. While a straightforward analysis shows that DSS-WL is strictly more powerful than 1-WL, an in-depth understanding on *what additional power* DSS-WL gains over 1-WL is still limited. The only new result is the very recent work of Frasca et al. (2022), who showed a 3-WL *upper bound* for the expressivity of DSS-WL. Yet, such a result actually gives a limitation of DSS-WL rather than showing its power. Moreover, there is a large gap between the highly strong 3-WL and the weak 1-WL. In the following, we take a different perspective and prove that DSS-WL is expressive for both types of biconnectivity problems.

**Theorem 3.2.** *Let $G = (\mathcal{V}_G, \mathcal{E}_G)$ and $H = (\mathcal{V}_H, \mathcal{E}_H)$ be two graphs, and let $\chi_G$ and $\chi_H$ be the corresponding DSS-WL color mapping with node marking policy. Then the following holds:*

- *For any two nodes $w \in \mathcal{V}_G$ and $x \in \mathcal{V}_H$, if $\chi_G(w) = \chi_H(x)$, then $w$ is a cut vertex if and only if $x$ is a cut vertex.*
- *For any two edges $\{w_1, w_2\} \in \mathcal{E}_G$ and $\{x_1, x_2\} \in \mathcal{E}_H$, if $\{\!\{\chi_G(w_1), \chi_G(w_2)\}\!\} = \{\!\{\chi_H(x_1), \chi_H(x_2)\}\!\}$, then $\{w_1, w_2\}$ is a cut edge if and only if $\{x_1, x_2\}$ is a cut edge.*

The proof of Theorem 3.2 is highly technical and is deferred to Appendix C.3. By using the basic results derived in Appendix C.1, we conduct a careful analysis of the DSS-WL color mapping and discover several important properties. They give insights on why DSS-WL can succeed in distinguishing biconnectivity, as we will discuss below.

**How can DSS-WL distinguish biconnectivity?** We find that a crucial advantage of DSS-WL over the classic 1-WL is that DSS-WL color mapping *implicitly* encodes *distance information* (see Lemma C.19(e) and Corollary C.24). For example, two nodes $u \in \mathcal{V}_G, v \in \mathcal{V}_H$ will have different DSS-WL colors if the distance set $\{\!\{\mathrm{dis}_G(u, w) : w \in \mathcal{V}_G\}\!\}$ differs from $\{\!\{\mathrm{dis}_H(v, w) : w \in \mathcal{V}_H\}\!\}$. Our proof highlights that distance information plays a vital role in distinguishing edge-biconnectivity when combining with color refinement algorithms (detailed in Section 4), and it also helps distinguish vertex-biconnectivity (see the proof of Lemma C.22). Consequently, our analysis provides a novel understanding and a strong justification for the success of DSS-WL in *two* aspects: the graph representation computed by DSS-WL intrinsically encodes distance and biconnectivity information, both of which are fundamental structural properties of graphs but are lacking in 1-WL.

**Discussions on graph generation policies**. Note that Theorem 3.2 holds for node marking policy. In fact, the ability of DSS-WL to encode distance information heavily relies on node marking as shown in the proof of Lemma C.19. In contrast, we prove that the ego-network policy $\pi_{\mathrm{EGO}(k)}$ cannot distinguish cut vertices (Proposition C.14), using the counterexample given in Figure 2(c). Therefore, our result shows an inherent advantage of node marking than the ego-network policy in distinguishing a class of non-isomorphic graphs, which is raised as an open question in Bevilacqua et al. (2022, Section 5). It also highlights a theoretical limitation of $\pi_{\mathrm{EGO}(k)}$ compared with its node marking version $\pi_{\mathrm{EGOM}(k)}$, a subtle difference that may not have received sufficient attention yet. For example, both the GNN-AK and GNN-AK-ctx architecture (Zhao et al., 2022) cannot solve vertex-biconnectivity problems since it is similar to $\pi_{\mathrm{EGO}(k)}$ (see Proposition C.15). On the other hand, the GNN-AK+ does not suffer from such a drawback although it also uses $\pi_{\mathrm{EGO}(k)}$, because it further adds distance encoding in each subgraph (which is more expressive than node marking).

**Discussions on DS-WL**. Bevilacqua et al. (2022); Cotta et al. (2021) also considered a weaker version of DSS-WL, called the DS-WL, which aggregates the node color in each subgraph without interaction across different subgraphs (see formula (10)). We show in Proposition C.16 that unfortunately, DS-WL with common node-based policies *cannot* identify cut vertices when the color of each node $v$ is defined as its associated subgraph representation $G_v$. This theoretically reveals the importance of cross-graph aggregation and justifies the design of DSS-WL. Finally, we point out that Qian et al. (2022) very recently proposed an extension of DS-WL that adds a final cross-graph aggregation procedure, for which our negative result may not hold. It may be an interesting direction to theoretically analyze the expressiveness of this type of DS-WL in future work.

## 4 GENERALIZED DISTANCE WEISFEILER-LEHMAN TEST

After an extensive review of prior GNN architectures, in this section we would like to formally study the following problem: can we design a principled and efficient GNN framework with provable expressiveness for biconnectivity? In fact, while in Section 3.2 we have proved that DSS-WL can solve biconnectivity problems, it is still far from enough. Firstly, the corresponding GNNs based on DSS-WL is usually sophisticated due to the complex aggregation formula (1), which inspires us to study whether simpler architectures exist. More importantly, DSS-WL suffers from high computational costs in both time and memory. Indeed, it requires $\Theta(n^2)$ space and $\Theta(nm)$ time per iteration (using policy $\pi_{\mathrm{NM}}$) to compute node colors for a graph with $n$ nodes and $m$ edges, which is $n$ times costly than 1-WL. Given the theoretical *linear* lower bound in Theorem 2.5, one may naturally raise the question of how to close the gap by developing more efficient color refinement algorithms.

We approach the problem by rethinking the classic 1-WL test. We argue that a major weakness of 1-WL is that it is agnostic to *distance information* between nodes, partly because each node can only "see" its *neighbors* in aggregation. On the other hand, the DSS-WL color mapping implicitly encodes distance information as shown in Section 3.2, which inspires us to formally study whether incorporating distance in the aggregation procedure is crucial for solving biconnectivity problems. To this end, we introduce a novel color refinement framework which we call Generalized Distance Weisfeiler-Lehman (GD-WL). The update rule of GD-WL is very simple and can be written as:

$$\chi_G^t(v) := \text{hash}\left(\{\!\{(d_G(v, u), \chi_G^{t-1}(u)) : u \in \mathcal{V}\}\!\}\right), \tag{3}$$

where $d_G$ can be an arbitrary distance metric. The full algorithm is described in Algorithm 4.

**SPD-WL for edge-biconnectivity**. As a special case, when choosing the *shortest path distance* $d_G = \text{dis}_G$, we obtain an algorithm which we call SPD-WL. It can be equivalently written as

$$\chi_G^t(v) := \text{hash}\left(\chi_G^{t-1}(v), \{\!\{\chi_G^{t-1}(u) : u \in \mathcal{N}_G(v)\}\!\}, \{\!\{\chi_G^{t-1}(u) : \text{dis}_G(v, u) = 2\}\!\}, \right.$$
$$\left. \cdots, \{\!\{\chi_G^{t-1}(u) : \text{dis}_G(v, u) = n - 1\}\!\}, \{\!\{\chi_G^{t-1}(u) : \text{dis}_G(v, u) = \infty\}\!\}\right). \tag{4}$$

From (4) it is clear that SPD-WL is strictly more powerful than 1-WL since it additionally aggregates the $k$-hop neighbors for all $k > 1$. There have been several prior works related to SPD-WL, including using distance encoding as node features (Li et al., 2020) or performing $k$-hop aggregation for some small $k$ (see Appendix D.2 for more related works and discussions). Yet, these works are either purely empirical or provide limited theoretical analysis (e.g., by focusing only on regular graphs). Instead, we introduce the general and more expressive SPD-WL framework with a rather different motivation and perform a systematic study on its expressive power. Our key result confirms that SPD-WL is fully expressive for all edge-biconnectivity problems listed in Section 2.

**Theorem 4.1.** *Let $G = (\mathcal{V}_G, \mathcal{E}_G)$ and $H = (\mathcal{V}_H, \mathcal{E}_H)$ be two graphs, and let $\chi_G$ and $\chi_H$ be the corresponding SPD-WL color mapping. Then the following holds:*

- *For any two edges $\{w_1, w_2\} \in \mathcal{E}_G$ and $\{x_1, x_2\} \in \mathcal{E}_H$, if $\{\!\{\chi_G(w_1), \chi_G(w_2)\}\!\} = \{\!\{\chi_H(x_1), \chi_H(x_2)\}\!\}$, then $\{w_1, w_2\}$ is a cut edge if and only if $\{x_1, x_2\}$ is a cut edge.*
- *If $\{\!\{\chi_G(w) : w \in \mathcal{V}_G\}\!\} = \{\!\{\chi_H(w) : w \in \mathcal{V}_H\}\!\}$, then $\text{BCETree}(G) \simeq \text{BCETree}(H)$.*

Theorem 4.1 is highly non-trivial and perhaps surprising at first sight, as it combines three seemingly unrelated concepts (i.e., SPD, biconnectivity, and the WL test) into a unified conclusion. We give a proof in Appendix C.4, which separately considers two cases: $\chi_G(w_1) \neq \chi_G(w_2)$ and $\chi_G(w_1) = \chi_G(w_2)$ (see Figure 2(b,d) for examples). For each case, the key technique in the proof is to construct an auxiliary graph (Definitions C.26 and C.34) that precisely characterizes the structural relationship between nodes that have specific colors (see Corollaries C.31 and C.40). Finally, we highlight that the second item of Theorem 4.1 may be particularly interesting: while distinguishing general non-isomorphic graphs are known to be hard (Cai et al., 1992; Babai, 2016), we show distinguishing non-isomorphic graphs with different block cut-edge trees can be much easily solved by SPD-WL.

**RD-WL for vertex-biconnectivity**. Unfortunately, while SPD-WL is fully expressive for edge-biconnectivity, it is not expressive for vertex-biconnectivity. We give a simple counterexample in Figure 2(c), where SPD-WL cannot distinguish the two graphs. Nevertheless, we find that by using a different distance metric, problems related to vertex-biconnectivity can also be fully solved. We propose such a choice called the *Resistance Distance* (RD) (denoted as $\text{dis}_G^R$). Like SPD, RD is also a basic metric in graph theory (Doyle & Snell, 1984; Klein & Randić, 1993) and has been widely used to characterize the relationship between nodes (Sanmartın et al., 2022; Velingker et al., 2022). Formally, the value of $\text{dis}_G^R(u, v)$ is defined to be the effective resistance between nodes $u$ and $v$ when treating $G$ as an electrical network where each edge corresponds to a resistance of one ohm.

RD has many elegant properties. First, it is a valid *metric*: indeed, RD is non-negative, semidefinite, symmetric, and satisfies the triangular inequality (see Appendix E.2). Moreover, similar to SPD, we also have $0 \leq \text{dis}_G^R(u, v) \leq n - 1$, and $\text{dis}_G^R(u, v) = \text{dis}_G(u, v)$ if $G$ is a tree. In Appendix E.2, we further show that RD is highly related to the graph Laplacian and can be efficiently calculated.

**Theorem 4.2.** *Let $G = (\mathcal{V}_G, \mathcal{E}_G)$ and $H = (\mathcal{V}_H, \mathcal{E}_H)$ be two graphs, and let $\chi_G$ and $\chi_H$ be the corresponding RD-WL color mapping. Then the following holds:*

- *For any two nodes $w \in \mathcal{V}_G$ and $x \in \mathcal{V}_H$, if $\chi_G(w) = \chi_H(x)$, then $w$ is a cut vertex if and only if $x$ is a cut vertex.*
- *If $\{\!\{\chi_G(w) : w \in \mathcal{V}_G\}\!\} = \{\!\{\chi_H(w) : w \in \mathcal{V}_H\}\!\}$, then $\text{BCVTree}(G) \simeq \text{BCVTree}(H)$.*

The form of Theorem 4.2 exactly parallels Theorem 4.1, which shows that RD-WL is fully expressive for vertex-biconnectivity. We give a proof of Theorem 4.1 in Appendix C.5. In particular, the proof of the second item is highly technical due to the challenges in analyzing the (complex) structure of the block cut-vertex tree. It also highlights that distinguishing non-isomorphic graphs that have different BCVTrees is much easier than the general case.

Combining Theorems 4.1 and 4.2 immediately yields the following corollary, showing that all biconnectivity problems can be solved within our proposed GD-WL framework.

**Corollary 4.3.** *When using both SPD and RD (i.e., by setting $d_G(u, v) := (\mathrm{dis}_G(u, v), \mathrm{dis}_G^{\mathrm{R}}(u, v))$), the corresponding GD-WL is fully expressive for both vertex-biconnectivity and edge-biconnectivity.*

**Computational cost**. The GD-WL framework only needs a complexity of $\Theta(n)$ space and $\Theta(n^2)$ time per-iteration for a graph of $n$ nodes and $m$ edges, both of which are strictly less than DSS-WL. In particular, GD-WL has the same space complexity as 1-WL, which can be crucial for large-scale tasks. On the other hand, one may ask how much computational overhead there is in preprocessing pairwise distances between nodes. We show in Appendix E that the computational cost can be trivially upper bounded by $O(nm)$ for SPD and $O(n^3)$ for RD. Note that the preprocessing step only needs to be executed once, and we find that the cost is negligible compared to the GNN architecture.

**Practical implementation**. One of the main advantages of GD-WL is its high degree of parallelizability. In particular, we find GD-WL can be easily implemented using a Transformer-like architecture by injecting distance information into Multi-head Attention (Vaswani et al., 2017), similar to the structural encoding in Graphormer (Ying et al., 2021a). The attention layer can be written as:

$$\mathbf{Y}^h = \left[\phi_1^h(\mathbf{D}) \odot \mathrm{softmax}\left(\mathbf{X}\mathbf{W}_Q^h(\mathbf{X}\mathbf{W}_K^h)^\top + \phi_2^h(\mathbf{D})\right)\right]\mathbf{X}\mathbf{W}_V^h, \tag{5}$$

where $\mathbf{X} \in \mathbb{R}^{n \times d}$ is the input node features of the previous layer, $\mathbf{D} \in \mathbb{R}^{n \times n}$ is the distance matrix such that $D_{uv} = d_G(u, v)$, $\mathbf{W}_Q^h, \mathbf{W}_K^h, \mathbf{W}_V^h \in \mathbb{R}^{d \times d_H}$ are learnable weight matrices of the $h$-th head, $\phi_1^h$ and $\phi_2^h$ are elementwise functions applied to $\mathbf{D}$ (possibly parameterized), and $\odot$ denotes the elementwise multiplication. The results $\mathbf{Y}^h \in \mathbb{R}^{n \times d_H}$ across all heads $h$ are then combined and projected to obtain the final output $\mathbf{Y} = \sum_h \mathbf{Y}^h \mathbf{W}_O^h$ where $\mathbf{W}_O^h \in \mathbb{R}^{d_H \times d}$. We call the resulting architecture Graphormer-GD, and the full structure of Graphormer-GD is provided in Appendix E.3.

It is easy to see that the mapping from $\mathbf{X}$ to $\mathbf{Y}$ in (5) is *equivariant* and simulates the GD-WL aggregation. Importantly, we have the following expressivity result, which precisely characterizes the power and limits of Graphormer-GD. We give a proof in Appendix E.3.

**Theorem 4.4.** *Graphormer-GD is at most as powerful as GD-WL in distinguishing non-isomorphic graphs. Moreover, when choosing proper functions $\phi_1^h$ and $\phi_2^h$ and using a sufficiently large number of heads and layers, Graphormer-GD is as powerful as GD-WL.*

**On the expressivity upper bound of GD-WL**. To complete the theoretical analysis, we finally provide an upper bound of the expressive power for our proposed SPD-WL and RD-WL, by studying the relationship with the standard 2-FWL (3-WL) algorithm.

**Theorem 4.5.** *The 2-FWL algorithm is more powerful than both SPD-WL and RD-WL. Formally, the 2-FWL color mapping induces a finer vertex partition than that of both SPD-WL and RD-WL.*

We give a proof in Appendix C.6. Using Theorem 4.5, we arrive at the concluding corollary:

**Corollary 4.6.** *The 2-FWL is fully expressive for both vertex-biconnectivity and edge-biconnectivity.*

## 5 EXPERIMENTS

In this section, we perform empirical evaluations of our proposed Graphormer-GD. We mainly consider the following two sets of experiments. *Firstly*, we would like to verify whether Graphormer-GD can indeed learn biconnectivity-related metrics easily as our theory predicts. *Secondly*, we would like to investigate whether GNNs with sufficient expressiveness for biconnectivity can also help real-world tasks and benefit the generalization performance as well. The code and models will be made publicly available at `https://github.com/lsj2408/Graphormer-GD`.

**Synthetic tasks**. To test the expressive power of GNNs for biconnectivity metrics, we separately consider two tasks: (i) Cut Vertex Detection and (ii) Cut Edge Detection. Given a GNN model

that outputs node features, we add a learnable prediction head that takes each node feature (or two node features corresponding to each edge) as input and predicts whether it is a cut vertex (cut edge) or not. The evaluation metric for both tasks is the graph-level accuracy, i.e., given a graph, the model prediction is considered correct only when all the cut vertices/edges are correctly identified. To make the results convincing, we construct a challenging dataset that comprises various types of hard graphs, including the regular graphs with cut vertices/edges and also Examples C.9 and C.10 mentioned in Section 3. We also choose several GNN baselines with different levels of expressive power: (i) classic MPNNs (Kipf & Welling, 2017; Veličković et al., 2018; Xu et al., 2019); (ii) Graph Substructure Network (Bouritsas et al., 2022); (iii) Graphormer (Ying et al., 2021a). The details of model configurations, dataset, and training procedure are provided in Appendix F.1.

The results are presented in Table 2. It can be seen that baseline GNNs cannot perfectly solve these synthetic tasks. In contrast, the Graphormer-GD achieves 100% accuracy on both tasks, implying that it can easily learn biconnectivity metrics even in very difficult graphs. Moreover, while using only SPD suffices to identify cut edges, it is still necessary to further incorporate RD to identify cut vertices. This is consistent with our theoretical results in Theorems 4.1, 4.2 and 4.4.

Table 2: Accuracy on cut vertex (articulation point) and cut edge (bridge) detection tasks.

| Model | Cut Vertex Detection | Cut Edge Detection |
|---|---|---|
| GCN (Kipf & Welling, 2017) | 51.5%±1.3% | 62.4%±1.8% |
| GAT (Veličković et al., 2018) | 52.0%±1.3% | 62.8%±1.9% |
| GIN (Xu et al., 2019) | 53.9%±1.7% | 63.1%±2.2% |
| GSN (Bouritsas et al., 2022) | 60.1%±1.9% | 70.7%±2.1% |
| Graphormer (Ying et al., 2021a) | 76.4%±2.8% | 84.5%±3.3% |
| Graphormer-GD (ours) | 100% | 100% |
| - w/o. Resistance Distance | 83.3%±2.7% | 100% |

**Real-world tasks**. We further study the empirical performance of our Graphormer-GD on the real-world benchmark: ZINC from Benchmarking-GNNs (Dwivedi et al., 2020). To show the scalability of Graphormer-GD, we train our models on both ZINC-Full (consisting of 250K molecular graphs) and ZINC-Subset (12K selected graphs). We comprehensively compare our model with prior expressive GNNs that have been publicly released. For a fair comparison, we ensure that the parameter budget of both Graphormer-GD and other compared models are around 500K, following Dwivedi et al. (2020). Details of baselines and settings are presented in Appendix F.2.

The results are shown in Table 3, where our score is averaged over four experiments with different seeds. We also list the per-epoch training time of different models on ZINC-subset as well as their model parameters. It can be seen that Graphormer-GD surpasses or matches all competitive baselines on the test set of both ZINC-Subset and ZINC-Full. Furthermore, we find that the empirical performance of compared models align with their expressive power measured by graph biconnectivity. For example, Subgraph GNNs that are expressive for biconnectivity also consistently outperform classic MPNNs by a large margin. Compared with Subgraph GNNs, the main advantage of Graphormer-GD is that it is simpler to implement, has stronger parallelizability, while still achieving better performance. Therefore, we believe our proposed architecture is both effective and efficient and can be well extended to more practical scenarios like drug discovery.

## 6 CONCLUSION

In this paper, we systematically investigate the expressive power of GNNs via the perspective of graph biconnectivity. Through the novel lens, we gain strong theoretical insights into the power and limits of existing popular GNNs. We then introduce the principled GD-WL framework that is fully expressive for all biconnectivity metrics. We further design the Graphormer-GD architecture that is provably powerful while enjoying practical efficiency and parallelizability. Experiments on both synthetic and real-world datasets demonstrate the effectiveness of Graphormer-GD.

There are still many promising directions that have not yet been explored. *Firstly*, it remains an important open problem whether biconnectivity can be solved more efficiently in $o(n^2)$ time using *equivariant* GNNs. *Secondly*, a deep understanding of GD-WL is generally lacking. For example, we conjecture that RD-WL can encode graph spectral (Lim et al., 2022) and is strictly more powerful than SPD-WL in distinguishing general graphs. *Thirdly*, it may be interesting to further investigate more expressive distance (structural) encoding schemes beyond RD-WL and explore how to encode them in Graph Transformers. *Finally*, one can extend biconnectivity to a hierarchy of higher-order variants (e.g., tri-connectivity), which provides a completely different view parallel to the WL hierarchy to study the expressive power and guide designing provably powerful GNNs architectures.

Table 3: Mean Absolute Error (MAE) on ZINC test set. Following Dwivedi et al. (2020), the parameter budget of compared models is set to 500k. We use * to indicate the best performance.

| Method | Model | Time (s) | Params | Test MAE | |
|---|---|---|---|---|---|
| | | | | ZINC-Subset | ZINC-Full |
| MPNNs | GIN (Xu et al., 2019) | 8.05 | 509,549 | 0.526±0.051 | 0.088±0.002 |
| | GraphSAGE (Hamilton et al., 2017) | 6.02 | 505,341 | 0.398±0.002 | 0.126±0.003 |
| | GAT (Veličković et al., 2018) | 8.28 | 531,345 | 0.384±0.007 | 0.111±0.002 |
| | GCN (Kipf & Welling, 2017) | 5.85 | 505,079 | 0.367±0.011 | 0.113±0.002 |
| | MoNet (Monti et al., 2017) | 7.19 | 504,013 | 0.292±0.006 | 0.090±0.002 |
| | GatedGCN-PE(Bresson & Laurent, 2017) | 10.74 | 505,011 | 0.214±0.006 | - |
| | MPNN(sum) (Gilmer et al., 2017) | - | 480,805 | 0.145±0.007 | - |
| | PNA (Corso et al., 2020) | - | 387,155 | 0.142±0.010 | - |
| Higher-order GNNs | RingGNN (Chen et al., 2019) | 178.03 | 527,283 | 0.353±0.019 | - |
| | 3WLGNN (Maron et al., 2019a) | 179.35 | 507,603 | 0.303±0.068 | - |
| Substructure-based GNNs | GSN (Bouritsas et al., 2022) | - | ~500k | 0.101±0.010 | - |
| | CIN-Small (Bodnar et al., 2021a) | - | ~100k | 0.094±0.004 | 0.044±0.003 |
| Subgraph GNNs | NGNN (Zhang & Li, 2021) | - | ~500k | 0.111±0.003 | 0.029±0.001 |
| | DSS-GNN (Bevilacqua et al., 2022) | - | 445,709 | 0.097±0.006 | - |
| | GNN-AK (Zhao et al., 2022) | - | ~500k | 0.105±0.010 | - |
| | GNN-AK+ (Zhao et al., 2022) | - | ~500k | 0.091±0.011 | - |
| | SUN (Frasca et al., 2022) | 15.04 | 526,489 | 0.083±0.003 | - |
| Graph Transformers | GT (Dwivedi & Bresson, 2021) | - | 588,929 | 0.226±0.014 | - |
| | SAN (Kreuzer et al., 2021) | - | 508,577 | 0.139±0.006 | - |
| | Graphormer (Ying et al., 2021a) | 12.26 | 489,321 | 0.122±0.006 | 0.052±0.005 |
| | URPE (Luo et al., 2022b) | 12.40 | 491,737 | 0.086±0.007 | 0.028±0.002 |
| GD-WL | Graphormer-GD (ours) | 12.52 | 502,793 | 0.081±0.009* | 0.025±0.004* |

## ACKNOWLEDGMENTS

Bohang Zhang is grateful to Ruichen Li for his great help in discussing and checking several of the main results in this paper, including Theorems 3.1, 3.2, 4.1 and C.58. In particular, after the initial submission, Ruichen Li discovered a simpler proof of Lemma C.28 and helped complete the proof of Theorem C.58. Bohang Zhang would also thank Yiheng Du, Kai Yang amd Ruichen Li for correcting some small mistakes in the proof of Lemmas C.20 and C.45. We also thank all the anonymous reviewers for the careful reviews and the valuable suggestions. Their help has further enhanced our work. This work is supported by National Key R&D Program of China (2022ZD0114900) and National Science Foundation of China (NSFC62276005).

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

# Appendix

## Table of Contents

# A    RECENT ADVANCES IN EXPRESSIVE GNNS

Since the seminal works of Xu et al. (2019); Morris et al. (2019), extensive studies have devoted to developing new GNN architectures with better expressiveness beyond the 1-WL test. These works can be broadly classified into the following categories.

**Higher-order GNNs**. One straightforward way to design provably more expressive GNNs is inspired by the higher-order WL tests (see Appendix B.2). Instead of performing node feature aggregation, these higher-order GNNs calculate a feature vector for each $k$-tuple of nodes ($k \geq 2$) and perform aggregation between features of different tuples using tensor operations (Morris et al., 2019; Maron et al., 2019b;c;a; Keriven & Peyré, 2019; Azizian & Lelarge, 2021; Geerts & Reutter, 2022). In particular, Maron et al. (2019a) leveraged equivariant matrix multiplication to design network layers that mimic the 2-FWL aggregation. Due to the huge computational cost of higher-order GNNs, several recent works considered improving efficiency by leveraging the sparse and local nature of graphs and designing a "local" version of the $k$-WL aggregation, which comes at the cost of some expressiveness (Morris et al., 2020; 2022). The work of Vignac et al. (2020) can also be seen as a local 2-order GNN and its expressive power is bounded by 3-IGN (Maron et al., 2019c).

**Substructure-based GNNs**. Another way to design more expressive GNNs is inspired by studying the failure cases of 1-WL test. In particular, Chen et al. (2020) pointed out that standard MPNNs cannot detect/count common substructures such as cycles, cliques, and paths. Based on this finding, Bouritsas et al. (2022) designed the Graph Substructure Network (GSN) by incorporating substructure counting into node features using a preprocessing step. Such an approach was later extended by Barceló et al. (2021) based on homomorphism counting. Bodnar et al. (2021b;a); Thiede et al. (2021); Horn et al. (2022) further developed novel WL aggregation schemes that take into account these substructures (e.g., cycles or cliques). Toenshoff et al. (2021) considered using random walk techniques to generate small substructures.

**Subgraph GNNs**. In fact, the graphs indistinguishable by 1-WL tend to possess a high degree of symmetry (e.g., see Figure 2). Based on this observation, a variety of recent approaches sought to break the symmetry by feeding *subgraphs* into an MPNN. To maintain equivariance, a set of subgraphs is generated *symmetrically* from the original graph using predefined policies, and the final output is aggregated across all subgraphs. There have been several subgraph generation policies in prior works, such as node deletion (Cotta et al., 2021), edge deletion (Bevilacqua et al., 2022), node marking (Papp & Wattenhofer, 2022), and ego-networks (Zhao et al., 2022; Zhang & Li, 2021; You et al., 2021). These works also slightly differ in the aggregation schemes. In particular, Bevilacqua et al. (2022) developed a unified framework, called ESAN, which includes per-layer aggregation across subgraphs and thus enjoys better expressiveness. Very recently, Frasca et al. (2022) further extended the framework based on a more relaxed symmetry analysis and proved an upper bound of its expressiveness to be 3-WL. Qian et al. (2022) provided a theoretical analysis of how subgraph GNNs relate to $k$-FWL and also designed an approach to learn policies.

**Non-equivariant GNNs**. Perhaps one of the simplest way to break the intrinsic symmetry of 1-WL aggregation is to use non-equivariant GNNs. Indeed, Loukas (2020) proved that if each node in a GNN is equipped with a unique identifier, then standard MPNNs can already be Turing universal. There have been several works that exploit this idea to build powerful GNNs, such as using port numbering (Sato et al., 2019), relational pooling (Murphy et al., 2019), random features (Sato et al., 2021; Abboud et al., 2021), or dropout techniques (Papp et al., 2021). However, since the resulting architectures cannot fully preserve equivariance, the sample complexity required for training and generalization may not be guaranteed (Garg et al., 2020). Therefore, in this paper we only focus on analyzing and designing equivariant GNNs.

**Other approaches**. Wijesinghe & Wang (2022); de Haan et al. (2020) designed novel variants of MPNNs based on more powerful neighborhood aggregation schemes that are aware of the local graph structure, rather than simply treating neighboring nodes as a set. Li et al. (2020); Velingker et al. (2022) incorporated distance encoding into node/edge features to enhance the expressive power of MPNNs. Balcilar et al. (2021); Feldman et al. (2022) utilized spectral information of graphs to achieve better expressiveness beyond 1-WL. Talak et al. (2021) proposed the Neural Tree Network that performs message passing between higher-order subgraphs instead of node-level aggregation.

Finally, for a comprehensive survey on expressive GNNs, we refer readers to Sato (2020) and Morris et al. (2021).

# B   THE WEISFEILER-LEHMAN ALGORITHMS AND RECENTLY PROPOSED VARIANTS

In this section, we give a precise description on the family of Weisfeiler-Lehman algorithms and several recently proposed variants that are studied in this paper. We first present the classic 1-WL algorithm (Weisfeiler & Leman, 1968) and the more advanced $k$-FWL (Cai et al., 1992; Morris et al., 2019). Then we present several recently proposed WL variants, including WL with Substructure Counting (SC-WL) (Bouritsas et al., 2022), Overlap Subgraph WL (OS-WL) (Wijesinghe & Wang, 2022), Equivariant Subgraph Aggregation WL (DSS-WL) (Bevilacqua et al., 2022) and Generalized Distance WL (GD-WL).

Throughout this section, we assume hash : $\mathcal{X} \to \mathcal{C}$ is an *injective* hash function that can map "arbitrary objects" to a color in $\mathcal{C}$ where $\mathcal{C}$ is an abstract set called the *color set*. Formally, the domain $\mathcal{X}$ comprises all the objects we are interested in:

- $\mathbb{R} \subset \mathcal{X}$ and $\mathcal{C} \subset \mathcal{X}$;
- For any finite multiset $\mathcal{M}$ with elements in $\mathcal{X}$, $\mathcal{M} \in \mathcal{X}$;
- For any tuple $\boldsymbol{c} \in \mathcal{X}^k$ of finite dimension $k \in \mathbb{N}_+$, $\boldsymbol{c} \in \mathcal{X}$.

## B.1   1-WL TEST

Given a graph $G = (\mathcal{V}, \mathcal{E})$, the 1-dimensional Weisfeiler-Lehman algorithm (1-WL), also called the *color refinement* algorithm, iteratively calculates a color mapping $\chi_G$ from each vertex $v \in \mathcal{V}$ to a color $\chi_G(v) \in \mathcal{C}$. The pseudo code of 1-WL is presented in Algorithm 1. Intuitively, at the beginning the color of each vertex is initialized to be the same. Then in each iteration, 1-WL algorithm updates each vertex color by combining its own color with the neighborhood color multiset using a hash function. This procedure is repeated for a sufficiently large number of iterations $T$, e.g. $T = |\mathcal{V}|$.

---

**Algorithm 1:** The 1-dimensional Weisfeiler-Lehman Algorithm

**Input** : Graph $G = (\mathcal{V}, \mathcal{E})$ and the number of iterations $T$
**Output:** Color mapping $\chi_G : \mathcal{V} \to \mathcal{C}$
1 **Initialize:** Pick a fixed (arbitrary) element $c_0 \in \mathcal{C}$, and let $\chi_G^0(v) := c_0$ for all $v \in \mathcal{V}$
2 **for** $t \leftarrow 1$ **to** $T$ **do**
3    **for each** $v \in \mathcal{V}$ **do**
4       $\chi_G^t(v) := \text{hash}\left(\chi_G^{t-1}(v), \{\!\{\chi_G^{t-1}(u) : u \in \mathcal{N}_G(v)\}\!\}\right)$
5 **Return:** $\chi_G^T$

---

At each iteration, the color mapping $\chi_G^t$ induces a *partition* of the vertex set $\mathcal{V}$ with an equivalence relation $\sim_{\chi_G^t}$ defined to be $u \sim_{\chi_G^t} v \iff \chi_G^t(u) = \chi_G^t(v)$ for $u, v \in \mathcal{V}$. We call each equivalence class a *color class* with an associated color $c \in \mathcal{C}$, denoted as $(\chi_G^t)^{-1}(c) := \{v \in \mathcal{V} : \chi_G^t(v) = c\}$. The corresponding partition is then denoted as $\mathcal{P}_G^t = \{(\chi_G^t)^{-1}(c) : c \in \mathcal{C}_G^t\}$ where $\mathcal{C}_G^t := \{\chi_G^t(v) : v \in \mathcal{V}\}$ is the color set containing all the presented colors of vertices in $G$.

An important observation is that each 1-WL iteration *refines* the partition $\mathcal{P}_G^t$ to a finer partition $\mathcal{P}_G^{t+1}$, because for any $u, v \in \mathcal{V}$, $u \sim_{\chi_G^{t+1}} v$ implies $u \sim_{\chi_G^t} v$. Since the number of vertices $|\mathcal{V}|$ is finite, there must exist an iteration $T_{\text{stable}} < |\mathcal{V}|$ such that $\mathcal{P}_G^{T_{\text{stable}}} = \mathcal{P}_G^{T_{\text{stable}}+1}$. It follows that $\mathcal{P}_G^t = \mathcal{P}_G^{T_{\text{stable}}}$ for all $t \geq T_{\text{stable}}$, i.e. the partition stabilizes. We thus denote $\mathcal{P}_G := \mathcal{P}_G^{T_{\text{stable}}}$ as the stable partition induced by the 1-WL algorithm, and denote $\chi_G$ as any stable color mapping (i.e. by picking any $\chi_G^t$ with $t \geq T_{\text{stable}}$). We can similarly define the inverse mapping $\chi_G^{-1}$. The mapping $\chi_G$ serves as a node feature extractor so that $\chi_G(v)$ is the representation of node $v \in \mathcal{V}$. Correspondingly, the multiset $\{\!\{\chi_G(v) : v \in \mathcal{V}\}\!\}$ can serve as the representation of graph $G$.

The 1-WL algorithm can be used to distinguish whether two graphs $G$ and $H$ are isomorphic, by comparing their graph representations $\{\!\{\chi_G(v) : v \in \mathcal{V}\}\!\}$ and $\{\!\{\chi_H(v) : v \in \mathcal{V}\}\!\}$. If the two multisets are not equivalent, then $G$ and $H$ are clearly non-isomorphic. Thus 1-WL is a necessary condition to test graph isomorphism. Nevertheless, the 1-WL test fails when $\{\!\{\chi_G(v) : v \in \mathcal{V}\}\!\} = \{\!\{\chi_H(v) : v \in \mathcal{V}\}\!\}$ but $G$ and $H$ are still non-isomorphic (see Figure 2 for a counterexample). This motivates the more powerful higher-order WL tests, which are illustrated in the next subsection.

## B.2  $k$-FWL TEST

In this section, we present a family of algorithms called the $k$-dimensional Folklore Weisfeiler-Lehman algorithms ($k$-FWL). Instead of calculating a node color mapping, $k$-FWL computes a color mapping on each $k$-tuple of nodes. The pseudo code of $k$-FWL ($k \geq 2$) is presented in Algorithm 2.

---

**Algorithm 2:** The $k$-dimensional Folklore Weisfeiler-Lehman Algorithm

---

**Input**  : Graph $G = (\mathcal{V}, \mathcal{E})$ and the number of iterations $T$
**Output:** Color mapping $\chi_G : \mathcal{V}^k \to \mathcal{C}$
1 **Initialize:** Pick three fixed different elements $c_0, c_1, c_{\text{node}} \in \mathcal{C}$, let $\chi_G^0(\boldsymbol{v}) := \text{hash}(\text{vec}(\mathbf{A}^{\boldsymbol{v}}))$
  for each $\boldsymbol{v} \in \mathcal{V}^k$ where $\mathbf{A}^{\boldsymbol{v}} \in \mathcal{C}^{k \times k}$ is a matrix with elements

$$A_{ij}^{\boldsymbol{v}} = \begin{cases} c_{\text{node}} & \text{if } v_i = v_j \\ c_0 & \text{if } v_i \neq v_j \text{ and } \{v_i, v_j\} \notin \mathcal{E} \\ c_1 & \text{if } v_i \neq v_j \text{ and } \{v_i, v_j\} \in \mathcal{E} \end{cases} \tag{6}$$

2 **for** $t \leftarrow 1$ **to** $T$ **do**
3     **for each** $\boldsymbol{v} \in \mathcal{V}^k$ **do**
4        $\chi_G^t(\boldsymbol{v}) := \text{hash}\left(\chi_G^{t-1}(\boldsymbol{v}), \{\!\!\{(\chi_G^{t-1}(N_1(\boldsymbol{v}, u)), \cdots, \chi_G^{t-1}(N_k(\boldsymbol{v}, u))) : u \in \mathcal{V}\}\!\!\}\right)$
5        where $N_i(\boldsymbol{v}, u) = (v_1, \cdots, v_{i-1}, u, v_{i+1}, \cdots, v_k)$
6 **Return:** $\chi_G^T$

---

Intuitively, at the beginning, the color of each vertex tuple $\boldsymbol{v}$ encodes the full structure (i.e. isomophism type) of the subgraph induced by the *ordered* vertex set $\{v_i : i \in [k]\}$, by hashing the "adjacency" matrix $\mathbf{A}^{\boldsymbol{v}}$ defined in (6). Then in each iteration, $k$-FWL algorithm updates the color of each vertex tuple by combining its own color with the "neighborhood" color using a hash function. Here, the neighborhood of a tuple $\boldsymbol{v}$ is all the tuples that differ $\boldsymbol{v}$ by exactly one element. These $k \times |\mathcal{V}|$ neighborhood colors are grouped into a multiset of size $|\mathcal{V}|$ where each element is a $k$-tuple. Finally, the update procedure is repeated for a sufficiently large number of iterations $T$, e.g. $T = |\mathcal{V}|^k$.

Simiar to 1-WL, the $k$-FWL color mapping $\chi_G^t$ induces a partition of the set of *vertex $k$-tuples* $\mathcal{V}^k$, and each $k$-FWL iteration *refines* the partition of the previous iteration. Since the number of vertex $k$-tuples $|\mathcal{V}|^k$ is finite, there must exist an iteration $T_{\text{stable}} < |\mathcal{V}|^k$ such that the partition no longer changes after $t \geq T_{\text{stable}}$. We denote the stable color mapping as $\chi_G$ by picking any $\chi_G^t$ with $t \geq T_{\text{stable}}$.

The $k$-FWL algorithm can be used to distinguish whether two graphs $G$ and $H$ are isomorphic, by comparing their graph representations $\{\!\!\{\chi_G(\boldsymbol{v}) : \boldsymbol{v} \in \mathcal{V}^k\}\!\!\}$ and $\{\!\!\{\chi_H(\boldsymbol{v}) : \boldsymbol{v} \in \mathcal{V}^k\}\!\!\}$. It has been proved that $k$-FWL is strictly more powerful than 1-WL in distinguishing non-isomorphic graphs, and $(k+1)$-FWL is strictly more powerful than $k$-FWL for all $k \geq 2$ (Cai et al., 1992).

Moreover, the $k$-FWL algorithm can also be used to extract *node* representations as with 1-WL. To do this, we can simply define $\chi_G(v) := \chi_G(v, \cdots, v)$ as the vertex color of the $k$-FWL algorithm (without abuse of notation), which induces a partition $\mathcal{P}_G$ over vertex set $\mathcal{V}$. It has been shown that this partition is *finer* than the partition induces by 1-WL, and also the vertex partition induced by $(k+1)$-FWL is finer than that of $k$-FWL (Kiefer, 2020).

## B.3  WL WITH SUBSTRUCTURE COUNTING (SC-WL)

Recently, Bouritsas et al. (2022) proposed a variant of the 1-WL algorithm by incorporating the so-called *substructure counting* into WL aggregation procedure. This yields a algorithm that is provably powerful than the original 1-WL test.

To describe the algorithm, we first need the notation of *automorphism group*. Given a graph $H = (\mathcal{V}_H, \mathcal{E}_H)$, an automorphism of $H$ is a bijective mapping $f : \mathcal{V}_H \to \mathcal{V}_H$ such that for any two vertices $u, v \in \mathcal{V}_H, \{u, v\} \in \mathcal{E}_H \iff \{f(u), f(v)\} \in \mathcal{E}_H$. It follows that all automorphisms of $H$ form a group under function composition, which is called the *automorphism group* and denoted as $\text{Aut}(H)$.

The automorphism group $\mathrm{Aut}(H)$ yields a partition of the vertex set $\mathcal{V}$, called *orbits*. Formally, given a vertex $v \in \mathcal{V}_H$, define its orbit $\mathrm{Orb}_H(v) = \{u \in \mathcal{V}_H : \exists f \in \mathrm{Aut}(H), f(u) = v\}$. The set of all orbits $H \backslash \mathrm{Aut}(H) := \{\mathrm{Orb}_H(v) : v \in \mathcal{V}_H\}$ is called the *quotient* of the automorphism. Denote $d_H = |H \backslash \mathrm{Aut}(H)|$ and denote the elements in $H \backslash \mathrm{Aut}(H)$ as $\{\mathcal{O}_{H,i}^{\mathrm{V}}\}_{i=1}^{d_H}$. We are now ready to describe the procedure of SC-WL.

**Pre-processing**. Depending on the tasks, one first specify a set of (small) connected graphs $\mathcal{H} = \{H_1, \cdots, H_k\}$, which will be used for sub-structure counting in the input graph $G$. Popular choices of these small graphs are cycles of different lengths (e.g., triangle or square) and cliques. Given a graph $G = (\mathcal{V}_G, \mathcal{E}_G)$, for each vertex $v \in \mathcal{V}_G$ and each graph $H \in \mathcal{H}$, the following quantities are calculated:

$$x_{H,i}^{\mathrm{V}}(v) := \left\{ G[\mathcal{S}] : \mathcal{S} \subset \mathcal{V}, G[\mathcal{S}] \simeq H, v \in \mathcal{S}, f_{G[\mathcal{S}] \to \mathcal{V}_H}(v) \in \mathcal{O}_{H,i}^{\mathrm{V}} \right\}, \quad i \in [d_H] \qquad (7)$$

where $f_{G[\mathcal{S}] \to \mathcal{V}_H}$ is any isomorphism that maps the vertices of graph $G[\mathcal{S}]$ to those of graph $H$. Intuitively, $x_{H,i}^{\mathrm{V}}(v)$ counts the number of induced subgraphs of $G$ that is isomorphic to $H$ and contains node $v$, such that the orbit of $v$ is similar to the orbit $\mathcal{O}_{H,i}^{\mathrm{V}}$. The counts corresponding to different orbits $\mathcal{O}_{H,i}^{\mathrm{V}}$ and different graphs $H$ are finally combined and concatenated into a vector:

$$\boldsymbol{x}^{\mathrm{V}}(v) = [\boldsymbol{x}_{H_1}^{\mathrm{V}}(v)^\top, \cdots, \boldsymbol{x}_{H_k}^{\mathrm{V}}(v)^\top]^\top \in \mathbb{N}_+^D \qquad (8)$$

where the dimension of $\boldsymbol{x}^{\mathrm{V}}(v)$ is $D = \sum_{i \in [k]} d_i$.

**Message Passing**. The message passing procedure is similar to Algorithm 1, except that the aggregation formula (Line 4) is replaced by the following update rule:

$$\chi_G^t(v) := \mathrm{hash}\left(\chi_G^{t-1}(v), \boldsymbol{x}^{\mathrm{V}}(v), \{\!\{(\chi_G^{t-1}(u), \boldsymbol{x}^{\mathrm{V}}(u)) : u \in \mathcal{N}_G(v)\}\!\}\right) \qquad (9)$$

which incorporates the substructure counts (7, 8). Note that the update rule (9) is slightly simpler than the original paper (Bouritsas et al., 2022, Section 3.2), but the expressive power of the two formulations are the same.

Finally, we note that the above procedure counts substructures and calculates features $\boldsymbol{x}^{\mathrm{V}}$ for *each vertex* of $G$. One can similarly consider calculating substructure counts for *each edge* of $G$, and the conclusion in this paper (Theorem 3.1) still holds. Please refer to Bouritsas et al. (2022) for more details on how to calculate edge features.

### B.4 EQUIVARIANT SUBGRAPH AGGREGATION WL (DSS-WL)

Recently, Bevilacqua et al. (2022) developd a new type of graph neural networks, called Equivariant Subgraph Aggregation Networks, as well as a new WL variant named DSS-WL. Given a graph $G = (\mathcal{V}, \mathcal{E})$, DSS-WL first generates a bag of graphs $\mathcal{B}_G^\pi = \{\!\{G_1, \cdots, G_m\}\!\}$ which share the vertices, i.e. $G_i = (\mathcal{V}, \mathcal{E}_i)$, but differ in the edge sets $\mathcal{E}_i$. Here $\pi$ denotes the graph generation policy which determines the edge set $\mathcal{E}_i$ for each graph $G_i$. The initial coloring $\chi_{G_i}^0(v)$ for each node $v \in \mathcal{V}$ in graph $G_i$ is also determined by $\pi$ and can be different across different nodes and graphs. In each iteration, the algorithm refines the color of each node by jointly aggregating its neighboring colors in the own graph and across different graphs. This procedure is repeated for a sufficiently large iterations $T$ to obtain the stable color mappings $\chi_{G_i}$ and $\chi_G$. The pseudo code of DSS-WL is presented in Algorithm 3.

The key component in the DSS-WL algorithm is the graph generation policy $\pi$ which must maintain *symmetry*, i.e., be equivairant under permutation of the vertex set. We list several common choices below:

- **Node marking policy** $\pi = \pi_{\mathrm{NM}}$. In this policy, we have $\mathcal{B}_G^\pi = \{\!\{G_v : v \in \mathcal{V}\}\!\}$ where $G_v = G$, i.e., there are $|\mathcal{V}|$ graphs in $\mathcal{B}_G^\pi$ whose structures are the completely the same. The difference, however, lies in the initial coloring which marks the special node $v$ in the following way: $\chi_{G_v}^0(v) = c_1$ and $\chi_{G_v}^0(u) = c_0$ for other nodes $u \neq v$, where $c_0, c_1 \in \mathcal{C}$ are two different colors.
- **Node deletion policy** $\pi = \pi_{\mathrm{ND}}$. The bag of graphs for this policy is also defined as $\mathcal{B}_G^\pi = \{\!\{G_v : v \in \mathcal{V}\}\!\}$, but each graph $G_v = (\mathcal{V}, \mathcal{E}_v)$ has a different edge set $\mathcal{E}_v := \mathcal{E} \backslash \{\{v, w\} : w \in \mathcal{N}_G(v)\}$. Intuitively, it removes all edges that connects to node $v$ and thus makes $v$ an isolated node. The initial coloring is chosen as a constant $\chi_{G_i}^0(v) = c_0$ for all $v \in \mathcal{V}$ and $G_i \in \mathcal{B}_G^\pi$ for some fixed color $c_0 \in \mathcal{C}$.

---

**Algorithm 3:** DSS Weisfeiler-Lehman Algorithm

---

**Input** : Graph $G = (\mathcal{V}, \mathcal{E})$, the number of iterations $T$, and graph selection policy $\pi$
**Output:** Color mapping $\chi_G : \mathcal{V} \to \mathcal{C}$

1 **Initialize:** Generate a bag of graphs $\mathcal{B}_G^\pi = \{\!\{G_i\}\!\}_{i=1}^m$, $G_i = (\mathcal{V}, \mathcal{E}_i)$ and initial coloring $\chi_{G_i}^0$ for $i \in [m]$ according to policy $\pi$
2 Let $\chi_G^0(v) := \mathrm{hash}\left(\{\!\{\chi_{G_i}^t(v) : i \in [m]\}\!\}\right)$ for each $v \in \mathcal{V}$
3 **for** $t \leftarrow 1$ **to** $T$ **do**
4     **for each** $v \in \mathcal{V}$ **do**
5         **for** $i \leftarrow 1$ **to** $m$ **do**
6             $\chi_{G_i}^t(v) :=$
            $\mathrm{hash}\left(\chi_{G_i}^{t-1}(v), \{\!\{\chi_{G_i}^{t-1}(u) : u \in \mathcal{N}_{G_i}(v)\}\!\}, \chi_G^{t-1}(v), \{\!\{\chi_G^{t-1}(u) : u \in \mathcal{N}_G(v)\}\!\}\right)$
7         $\chi_G^t(v) := \mathrm{hash}\left(\{\!\{\chi_{G_i}^t(v) : i \in [m]\}\!\}\right)$
8 **Return:** $\chi_G^T$

---

- **Ego network policy** $\pi = \pi_{\mathrm{EGO}(k)}$. In this policy, we also have $\mathcal{B}_G^\pi = \{\!\{G_v : v \in \mathcal{V}\}\!\}$, $G_v = (\mathcal{V}, \mathcal{E}_v)$. The edge set $\mathcal{E}_v$ is defined as $\mathcal{E}_v := \{\{u, w\} \in \mathcal{E} : \mathrm{dis}_G(u, v) \le k, \mathrm{dis}_G(w, v) \le k\}$, which corresponds to a subgraph containing all the $k$-hop neighbors of $v$ and isolating other nodes. The initial coloring is chosen as $\chi_{G_i}^0(v) = c_0$ for all $v \in \mathcal{V}$ and $G_i \in \mathcal{B}_G^\pi$ where $c_0 \in \mathcal{C}$ is a constant. One can also consider the **ego network policy with marking** $\pi = \pi_{\mathrm{EGOM}(k)}$, by marking the initial color of the special node $v$ for each $G_v$.

We note that for all the above policies, $|\mathcal{B}_G^\pi| = |\mathcal{V}|$. There are other choices such as the edge deletion policy (Bevilacqua et al., 2022), but we do not discuss them in this paper. A straightforward analysis yields that DSS-WL with any above policy is strictly powerful than the classic 1-WL algorithm. Also, node marking policy has been shown to be not less powerful than the node deletion policy (Papp & Wattenhofer, 2022).

Finally, we highlight that Bevilacqua et al. (2022); Cotta et al. (2021) also proposed a weaker version of DSS-WL, called the DS-WL algorithm. The difference is that for DS-WL, Lines 6 and 7 in Algorithm 3 are replaced by a simple 1-WL aggregation:

$$\chi_{G_i}^t(v) := \mathrm{hash}\left(\chi_{G_i}^{t-1}(v), \{\!\{\chi_{G_i}^{t-1}(u) : u \in \mathcal{N}_G(v)\}\!\}\right). \tag{10}$$

However, the original formulation of DS-WL (Bevilacqua et al., 2022) only outputs a graph representation $\{\!\{\{\!\{\chi_{G_i}(v) : v \in \mathcal{V}\}\!\} : G_i \in \mathcal{B}_G^\pi\}\!\}$ rather than outputs each node color, which does not suit the node-level tasks (e.g., finding cut vertices). Nevertheless, there are simple adaptations that makes DS-WL output a color mapping $\chi_G$. We will study these adaptations in Appendix C.2 (see the paragraph above Proposition C.16) and discuss their limitations compared with DSS-WL.

## B.5 Generalized Distance WL (GD-WL)

In this paper, we study a new variant of the color refinement algorithm, called the Generalized Distance WL (GD-WL). The complete algorithm is described below. As a special case, when choosing $d_G = \mathrm{dis}_G$, the resulting algorithm is called the Shortest Path Distance WL (SPD-WL), which is strictly powerful than the classic 1-WL.

---

**Algorithm 4:** The Genealized Distance Weisfeiler-Lehman Algorithm

---

**Input** : Graph $G = (\mathcal{V}, \mathcal{E})$, distance metric $d_G : \mathcal{V} \times \mathcal{V} \to \mathbb{R}_+$, and the number of iterations $T$
**Output:** Color mapping $\chi_G : \mathcal{V} \to \mathcal{C}$

1 **Initialize:** Pick a fixed (arbitrary) element $c_0 \in \mathcal{C}$, and let $\chi_G^0(v) := c_0$ for all $v \in \mathcal{V}$
2 **for** $t \leftarrow 1$ **to** $T$ **do**
3     **for each** $v \in \mathcal{V}$ **do**
4         $\chi_G^t(v) := \mathrm{hash}\left(\{\!\{(d_G(v, u), \chi_G^{t-1}(u)) : u \in \mathcal{V}\}\!\}\right)$
5 **Return:** $\chi_G^T$

---

## C   PROOF OF THEOREMS

This section provides all the missing proofs in this paper. For the convenience of reading, we will restate each theorem before giving a proof.

### C.1   PROPERTIES OF COLOR REFINEMENT ALGORITHMS

In this subsection, we first derive several important properties that are shared by a general class of color refinement algorithms. They will serve as key lemmas in our subsequent proofs. Here, a general color refinement algorithm takes a graph $G = (\mathcal{V}_G, \mathcal{E}_G)$ as input and calculates a color mapping $\chi_G : \mathcal{V}_G \to \mathcal{C}$. We first define a concept called the *WL-condition*.

**Definition C.1.** A color mapping $\chi_G : \mathcal{V}_G \to \mathcal{C}$ is said to satisfy the WL-condition if for any two vertices $u, v$ with the same color (i.e. $\chi_G(u) = \chi_G(v)$) and any color $c \in \mathcal{C}$,

$$|\mathcal{N}_G(u) \cap \chi_G^{-1}(c)| = |\mathcal{N}_G(v) \cap \chi_G^{-1}(c)|,$$

where $\chi_G^{-1}$ is the inverse mapping of $\chi_G$.

**Remark C.2.** The WL-condition can be further generalized to handle two graphs. Let $\chi_G : \mathcal{V}_G \to \mathcal{C}$ and $\chi_H : \mathcal{V}_H \to \mathcal{C}$ be two color mappings obtained by applying the same color refinement algorithm for graphs $G$ and $H$, respectively. $\chi_G$ and $\chi_H$ are said to *jointly satisfy the WL-condition*, if for any two vertices $u \in \mathcal{V}_G$ and $v \in \mathcal{V}_H$ with the same color ($\chi_G(u) = \chi_H(v)$) and any color $c \in \mathcal{C}$,

$$|\mathcal{N}_G(u) \cap \chi_G^{-1}(c)| = |\mathcal{N}_H(v) \cap \chi_H^{-1}(c)|.$$

It clearly implies Definition C.1 by choosing $G = H$.

It is easy to see that the classic 1-WL algorithm (Algorithm 1) satisfies the WL-condition. In fact, many of the presented algorithms in this paper satisfy such a condition as we will show below, such as DSS-WL (Algorithm 3), SPD-WL (Algorithm 4 with $d_G = \text{dis}_G$), and $k$-FWL (Algorithm 2).

**Proposition C.3.** *Consider the DSS-WL algorithm (Algorithm 4) with arbitrary graph selection policy $\pi$. Let $\chi_G$ and $\chi_H$ be the color mappings for graphs $G$ and $H$, and let $\{\!\{\chi_{G_i} : i \in [m_G]\}\!\}$ and $\{\!\{\chi_{H_i} : i \in [m_H]\}\!\}$ be the color mapping for subgraphs generated by $\pi$. Then,*

- *$\chi_G$ and $\chi_H$ jointly satisfy the WL-condition;*
- *$\chi_{G_i}$ and $\chi_{H_j}$ jointly satisfy the WL-condition for any $i \in [m_G]$ and $j \in [m_H]$.*

*Proof.* We first prove the second bullet of Proposition C.3. By definition of the DSS-WL aggregation procedure (Line 6 in Algorithm 3), $\chi_{G_i}(u) = \chi_{H_i}(v)$ already implies $\{\!\{\chi_{G_i}(w) : w \in \mathcal{N}_{G_i}(u)\}\!\} = \{\!\{\chi_{H_j}(w) : w \in \mathcal{N}_{H_j}(v)\}\!\}$. Namely, $|\{w : w \in \mathcal{N}_{G_i}(u) \cap \chi_{G_i}^{-1}(c)\}| = |\{w : w \in \mathcal{N}_{H_j}(v) \cap \chi_{H_j}^{-1}(c)\}|$ holds for any $c \in \mathcal{C}$.

We then turn to the first bullet. If $\chi_G(u) = \chi_H(v)$, then $\{\!\{\chi_{G_i}(u) : i \in [m_G]\}\!\} = \{\!\{\chi_{H_j}(v) : j \in [m_H]\}\!\}$ (Line 7 in Algorithm 3). Then there exists a pair of indices $i \in [m_G]$ and $j \in [m_H]$ such that $\chi_{G_i}(u) = \chi_{H_j}(v)$. By definition of the DSS-WL aggregation, it implies $\{\!\{\chi_G(w) : w \in \mathcal{N}_G(u)\}\!\} = \{\!\{\chi_H(w) : w \in \mathcal{N}_H(v)\}\!\}$ and concludes the proof. $\qquad\square$

**Proposition C.4.** *Let $\chi_G$ and $\chi_H$ be two mappings returned by SPD-WL (Algorithm 4 with $d_G = \text{dis}_G$) for graphs $G$ and $H$, respectively. Then $\chi_G$ and $\chi_H$ jointly satisfy the WL-condition.*

*Proof.* If $\chi_G(u) = \chi_H(v)$ for some nodes $u, v$, then by the update rule (Line 4 in Algorithm 4)

$$\{\!\{(\text{dis}_G(u, w), \chi_G(w)) : w \in \mathcal{V}\}\!\} = \{\!\{(\text{dis}_G(v, w), \chi_G(w)) : w \in \mathcal{V}\}\!\}.$$

Since $w \in \mathcal{N}_G(u)$ if and only if $\text{dis}_G(u, w) = 1$, we have

$$\{\!\{\chi_G(w) : w \in \mathcal{N}_G(u)\}\!\} = \{\!\{\chi_G(w) : w \in \mathcal{N}_G(v)\}\!\}.$$

Therefore, for any $c \in \mathcal{C}$, $|\{w : w \in \mathcal{N}_G(u) \cap \chi_G^{-1}(c)\}| = |\{w : w \in \mathcal{N}_G(v) \cap \chi_G^{-1}(c)\}|$. $\qquad\square$

**Proposition C.5.** *Let $\chi_G$ and $\chi_H$ be two vertex color mappings returned by the $k$-FWL algorithm ($k \geq 2$). Then $\chi_G$ and $\chi_H$ jointly satisfy the WL-condition.*

*Proof.* Let $\chi_G(u) = \chi_H(v)$ for some $u \in \mathcal{V}_G$ and $v \in \mathcal{V}_H$. By the update formula (Line 4 in Algorithm 2), $\{\!\{\chi_G(u, \cdots, u, w) : w \in \mathcal{V}_G\}\!\} = \{\!\{\chi_H(v, \cdots, v, w) : w \in \mathcal{V}_H\}\!\}$. Note that for any nodes $w_1 \in \mathcal{V}_G, w_2 \in \mathcal{V}_H$ and any $x_1 \in \mathcal{N}_G(w_1), x_2 \notin \mathcal{N}_H(w_2)$, one has $\chi_G(w_1, \cdots, w_1, x_1) \neq \chi_H(w_2, \cdots, w_2, x_2)$. This is obtained by the definition of the initialization mapping $\chi_G^0$ and the fact that $\chi_G$ refines $\chi_G^0$. Consequently, $\{\!\{\chi_G(u, \cdots, u, w) : w \in \mathcal{N}_G(u)\}\!\} = \{\!\{\chi_G(v, \cdots, v, w) : w \in \mathcal{N}_H(v)\}\!\}$. Next, we can use the fact that if $\chi_G(u, \cdots, u, w_1) = \chi_G(v, \cdots, v, w_2)$ for some $w_1, w_2 \in \mathcal{V}$, then $\chi_G(w_1) = \chi_G(w_2)$ (see Lemma C.6). Therefore, $\{\!\{\chi_G(w) : w \in \mathcal{N}_G(u)\}\!\} = \{\!\{\chi_G(w) : w \in \mathcal{N}_H(v)\}\!\}$, which concludes the proof. $\qquad\square$

To complete the proof of Proposition C.5, it remains to prove the following lemma:

**Lemma C.6.** *Let $\chi_G$ and $\chi_H$ be color mappings for graphs $G$ and $H$ in the $k$-FWL algorithm ($k \geq 2$). Denote*

$$\mathrm{cat}_{i,j}(w, x) := (\underbrace{w, \cdots, w}_{i \text{ times}}, \underbrace{x, \cdots, x}_{j \text{ times}}).$$

*Then for any $i \in [k-1]$ and any nodes $u, w \in \mathcal{V}_G$, $v, x \in \mathcal{V}_H$, if $\chi_G(\mathrm{cat}_{k-i,i}(u, w)) = \chi_H(\mathrm{cat}_{k-i,i}(v, x))$, then $\chi_G(\mathrm{cat}_{k-i-1,i+1}(u, w)) = \chi_H(\mathrm{cat}_{k-i-1,i+1}(v, x))$. Consequently, $\chi_G(w) = \chi_H(x)$.*

*Proof.* By the update formula (Line 4 in Algorithm 2), $\chi_G(\mathrm{cat}_{k-i,i}(u, w)) = \chi_H(\mathrm{cat}_{k-i,i}(v, x))$ implies that $\{\!\{\chi_G(\mathrm{cat}_{k-i-1,1,i}(u, y, w)) : y \in \mathcal{V}_G\}\!\} = \{\!\{\chi_H(\mathrm{cat}_{k-i-1,1,i}(v, y, x)) : y \in \mathcal{V}_H\}\!\}$. Note that for any $j \in [k-1]$ and any $\boldsymbol{z} \in \mathcal{V}_G^k$, $\boldsymbol{z}' \in \mathcal{V}_H^k$ with $z_j = z_{j+1}$ but $z'_j \neq z'_{j+1}$, one has $\chi_G(\boldsymbol{z}) \neq \chi_H(\boldsymbol{z}')$. This is obtained by the definition of the initialization mapping $\chi_G^0$ and the fact that $\chi_G$ refines $\chi_G^0$. Therefore, we have $\chi_G(\mathrm{cat}_{k-i-1,i+1}(u, w)) = \chi_H(\mathrm{cat}_{k-i-1,i+1}(v, x))$, as desired. $\qquad\square$

Equipped with the concept of WL-condition, we now present several key results. In the following, let $\chi_G : \mathcal{V}_G \to \mathcal{C}$ and $\chi_H : \mathcal{V}_H \to \mathcal{C}$ be two color mappings jointly satisfying the WL-condition.

**Lemma C.7.** *Let $(v_0, \cdots, v_d)$ be any path (not necessarily simple) of length $d$ in graph $G$. Then for any node $u_0 \in \chi_H^{-1}(\chi_G(v_0))$ in graph $H$, there exists a path $(u_0, \cdots, u_d)$ of the same length $d$ starting at $u_0$, such that $\chi_H(u_i) = \chi_G(v_i)$ holds for all $i \in [d]$.*

*Proof.* The proof is based on induction over the path length $d$. For the base case of $d = 1$, if the conclusion does not hold, then there exists two vertices $u \in \mathcal{V}_G$, $v \in \mathcal{V}_H$ with the same color (i.e. $\chi_G(u) = \chi_H(v)$) and a color $c = \chi_G(v_1)$ such that $\mathcal{N}_G(u) \cap \chi_G^{-1}(c) \neq \emptyset$ but $\mathcal{N}_H(v) \cap \chi_H^{-1}(c) = \emptyset$. This obviously contradicts the WL-condition. For the induction step on the path length $d$, one can just split it by two parts $(v_0, \cdots, v_{d-1})$ and $(v_{d-1}, v_d)$. Separately using induction yields two paths $(u_0, \cdots, u_{d-1})$ and $(u_{d-1}, u_d)$ such that $\chi_H(u_i) = \chi_G(v_i)$ for all $i \in [d]$. By linking the two paths we have completed the proof. $\qquad\square$

Finally, let us define the shortest path distance between node $u$ and vertex set $\mathcal{S}$ as $\mathrm{dis}_G(u, \mathcal{S}) := \min_{v \in \mathcal{S}} \mathrm{dis}_G(u, v)$. The above lemma directly yields the following corollary:

**Corollary C.8.** *For any color $c \in \{\chi_G(w) : w \in \mathcal{V}_G\}$ and any two vertices $u \in \mathcal{V}_G$, $v \in \mathcal{V}_H$ with the same color (i.e. $\chi_G(u) = \chi_H(v)$), $\mathrm{dis}_G(u, \chi_G^{-1}(c)) = \mathrm{dis}_H(v, \chi_H^{-1}(c))$.*

## C.2 COUNTEREXAMPLES

We provides the following two families of counterexamples, which most prior works cannot distinguish.

**Example C.9.** Let $G_1 = (\mathcal{V}, \mathcal{E}_1)$ and $G_2 = (\mathcal{V}, \mathcal{E}_2)$ be a pair of graphs with $n = 2km + 1$ nodes where $m, k$ are two positive integers satisfying $mk \geq 3$. Denote $\mathcal{V} = [n]$ and define the edge sets as follows:

$$\begin{aligned}
\mathcal{E}_1 &= \{\{i, (i \bmod 2km) + 1\} : i \in [2km]\} \cup \{\{n, i\} : i \in [2km], i \bmod m = 0\}, \\
\mathcal{E}_2 &= \{\{i, (i \bmod km) + 1\} : i \in [km]\} \cup \{\{i + km, (i \bmod km) + km + 1\} : i \in [km]\} \cup \\
&\quad \{\{n, i\} : i \in [2km], i \bmod m = 0\}.
\end{aligned}$$

See Figure 2(a-c) for an illustration of three cases: (i) $m = 2, k = 2$; (ii) $m = 4, k = 1$; (iii) $m = 1, k = 4$. It is easy to see that regardless of the chosen of $m$ and $k$, $G_1$ always has no cut vertex but $G_2$ do always have a cut vertex with node number $n$. The case of $k = 1$ is more special, for which $G_2$ actually has three cut vertices with node number $m$, $2m$, and $n$, respectively, and it even has two cut edges $\{m, n\}$ and $\{2m, n\}$ (Figure 2(b)).

**Example C.10.** Let $G_1 = (\mathcal{V}, \mathcal{E}_1)$ and $G_2 = (\mathcal{V}, \mathcal{E}_2)$ be a pair of graphs with $n = 2m$ nodes where $m \geq 3$ is an arbitrary integer. Denote $\mathcal{V} = [n]$ and define the edge sets as follows:

$$\mathcal{E}_1 = \{\{i, (i \bmod n) + 1\} : i \in [n]\} \cup \{\{m, 2m\}\},$$

$$\mathcal{E}_2 = \{\{i, (i \bmod m) + 1\} : i \in [m]\} \cup \{\{i + m, (i \bmod m) + m + 1\} : i \in [m]\} \cup \{\{m, 2m\}\}.$$

See Figure 2(d) for an illustration of the case $n = 8$. It is easy to see that $G_1$ does not have any cut vertex or cut edge, but $G_2$ do have two cut vertices with node number $m$ and $2m$, and has a cut edge $\{m, 2m\}$.

**Theorem C.11.** *Let $\mathcal{H} = \{H_1, \cdots, H_k\}$, $H_i = (\mathcal{V}_i, \mathcal{E}_i)$ be any set of connected graphs and denote $n_{\mathrm{V}} = \max_{i \in [k]} |\mathcal{V}_i|$. Then SC-WL (Appendix B.3) using the substructure set $\mathcal{H}$ can neither distinguish whether a given graph has cut vertices nor distinguish whether it has cut edges. Moreover, there exist counterexample graphs whose size (both in terms of vertices and edges) is $O(n_{\mathrm{V}})$.*

*Proof.* We would like to prove that SC-WL cannot distinguish both Examples C.9 and C.10 when $n_{\mathrm{V}} < m$ ($m$ is defined in these examples). First note that for both examples, any cycle in both $G_1$ and $G_2$ has a length of at least $m$. Since the number of nodes in $H_i$ is $O(n_{\mathrm{V}})$, if $H_i$ contains cycles, it will not occur in both $G_1$ and $G_2$, thus taking no effect in distinguishing the two graphs. As a result, we can simply assume all graphs in $\mathcal{H}$ are trees (connected graphs with no cycles). Below, we provide a complete proof for Example C.9, which already yields the conclusion that SC-WL can neither distinguish cut vertices nor cut edges. We omit the proof for Example C.10 since the proof technique is similar.

*Proof for Example C.9.* Let $H_i$ be a tree with less than $m$ vertices where $m$ is defined in Example C.9. By symmetry of the two graphs $G_1$ and $G_2$, it suffices to prove the following two types of equations: $\boldsymbol{x}_{G_1}^{\mathrm{V}}(n) = \boldsymbol{x}_{G_2}^{\mathrm{V}}(n)$ and $\boldsymbol{x}_{G_1}^{\mathrm{V}}(i) = \boldsymbol{x}_{G_2}^{\mathrm{V}}(i)$ for all $m < i \leq 2m$, where $\boldsymbol{x}^{\mathrm{V}}$ is defined in (8). We first aim to prove that $\boldsymbol{x}_{G_1}^{\mathrm{V}}(v) = \boldsymbol{x}_{G_2}^{\mathrm{V}}(v)$ for $v \in \{m + 1, \cdots, 2m\}$. Consider an induced subgraph $G_1[\mathcal{S}]$ which is isomorphic to $H_i$ and contains node $v$. Define the set $\mathcal{T} := \{jm : j \in [k]\} \cap \mathcal{S}$. For ease of presentation, we define an operation $\mathrm{cir}(x, a, b)$ that outputs an integer $y$ in the range of $(a, b]$ such that $y$ has the same remainder as $x \pmod{b - a}$. Formally, $\mathrm{cir}(x, a, b) = y$ if $a < y \leq b$ and $x \equiv y \pmod{b - a}$.

- If $n \notin \mathcal{S}$, then it is easy to see that $G_1[\mathcal{S}]$ is a chain, i.e., no vertices have a degree larger than 2. We define the following mapping $g_{\mathcal{S}} : \mathcal{S} \to [n]$, such that

$$g_{\mathcal{S}}(u) = \begin{cases} \mathrm{cir}(u, m, 2m) & \text{if } k = 1, \\ \mathrm{cir}(u, 0, km) & \text{if } k \geq 2. \end{cases}$$

  In this way, the chain $G_1[\mathcal{S}]$ is mapped to a chain of $G_2$ that contains $v$. Concretely, denote $g_{\mathcal{S}}(\mathcal{S}) = \{g_{\mathcal{S}}(u) : u \in \mathcal{S}\}$, then $G_2[g_{\mathcal{S}}(\mathcal{S})] \simeq G_1[\mathcal{S}] \simeq H_i$, and obviously the orbit of $v$ in $G_2[g_{\mathcal{S}}(\mathcal{S})]$ matches the orbit of $v$ in $G_1[\mathcal{S}]$. See Figure 3(a,b) for an illustration of this case.

- If $n \in \mathcal{S}$, then it is easy to see that the set $\mathcal{T} \neq \emptyset$. We will similarly construct a mapping $g_{\mathcal{S}} : \mathcal{S} \to [n]$ that maps $\mathcal{S}$ to $g_{\mathcal{S}}(\mathcal{S})$ satisfying $g_{\mathcal{S}}(v) = v$, which is defined as follows. For each $u \in \mathcal{S} \backslash \{n\}$, we find a unique vertex $w_u$ in $\mathcal{T}$ such that $\mathrm{dis}_{G_1[\mathcal{S}]}(u, w_u)$ is the minimum. Note that the node $w_u$ is well-defined since $\mathcal{T} \neq \emptyset$ and any path in $G_1[\mathcal{S}]$ from $u$ to a node in $\mathcal{T}$ goes through $w_u$. Define

$$g_{\mathcal{S}}(u) = \begin{cases} \mathrm{cir}(u, m, 2m) & \text{if } k = 1 \text{ and } w_u = w_v, \\ \mathrm{cir}(u, 0, m) & \text{if } k = 1 \text{ and } w_u \neq w_v, \\ \mathrm{cir}(u, 0, km) & \text{if } k > 1 \text{ and } w_u \leq km, \\ \mathrm{cir}(u, km, 2km) & \text{if } k > 1 \text{ and } w_u > km. \end{cases}$$

  We also define $g_{\mathcal{S}}(n) = n$. Such a definition guarantees that for any $x_1, x_2 \in \mathcal{S}$, $\{x_1, x_2\} \in \mathcal{E}_{G_1} \iff \{g_{\mathcal{S}}(x_1), g_{\mathcal{S}}(x_2)\} \in \mathcal{E}_{G_2}$. Therefore, $G_2[g_{\mathcal{S}}(\mathcal{S})] \simeq G_1[\mathcal{S}] \simeq H_i$. Moreover, observe that $g_{\mathcal{S}}(u) \equiv u \pmod{m}$ always holds, and thus it is easy to see that the orbit of $v$ in $G_2[g_{\mathcal{S}}(\mathcal{S})]$ matches the orbit of $v$ in $G_1[\mathcal{S}]$. See Figure 3(c,d) for an illustration of this case.

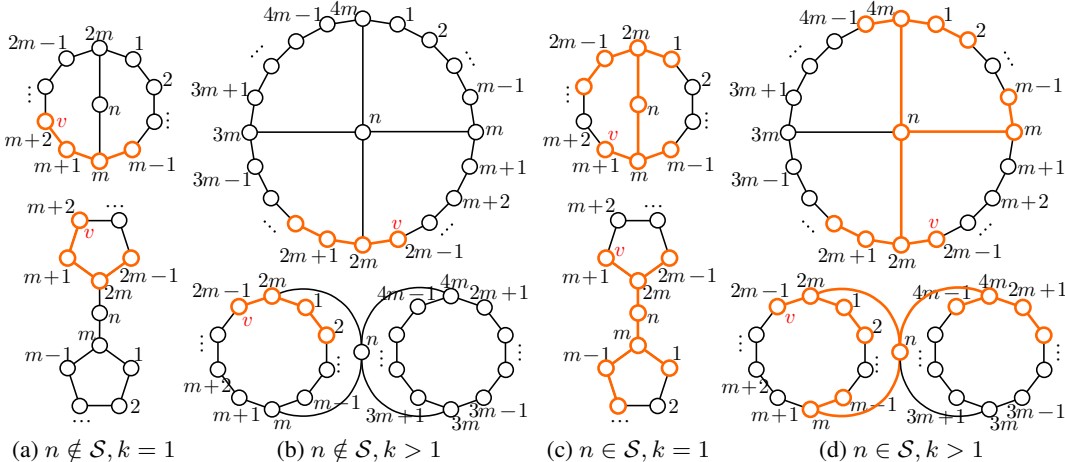

(a) $n \notin \mathcal{S}, k = 1$      (b) $n \notin \mathcal{S}, k > 1$      (c) $n \in \mathcal{S}, k = 1$      (d) $n \in \mathcal{S}, k > 1$

Figure 3: Illustration of the proof of Theorem 3.1. The trees $G_1[\mathcal{S}], G_2[g(\mathcal{S})]$ are outlined by orange.

Finally, note that for any two different sets $\mathcal{S}_1$ and $\mathcal{S}_2$ such that $G_1[\mathcal{S}_1] \simeq G_1[\mathcal{S}_2] \simeq H_i$, we have $g_{\mathcal{S}_1}(\mathcal{S}_1) \neq g_{\mathcal{S}_2}(\mathcal{S}_2)$, which guarantees that the mapping $g : \{\mathcal{S} \subset [n] : G_1[\mathcal{S}] \simeq H_i, v \in \mathcal{S}\} \to \{\mathcal{S} \subset [n] : G_2[\mathcal{S}] \simeq H_i, v \in \mathcal{S}\}$ defined to be $g(\mathcal{S}) = g_{\mathcal{S}}(\mathcal{S})$ is injective. One can further check that the mapping $g$ is also surjective, and thus it is bijective. This means $\boldsymbol{x}_{G_1}^{\mathrm{V}}(v) = \boldsymbol{x}_{G_2}^{\mathrm{V}}(v)$ for $v \in \{m, \cdots, 2m - 1\}$. The proof for $\boldsymbol{x}_{G_1}^{\mathrm{V}}(n) = \boldsymbol{x}_{G_2}^{\mathrm{V}}(n)$ is almost the same, so we omit it here. Noting that under classic 1-WL, the colors $\chi_{G_1}(v) = \chi_{G_2}(v)$ are also the same. Therefore, adding the features $\boldsymbol{x}^{\mathrm{V}}(v)$ does not help distinguish the two graphs. We have finished the proof for Example C.9. □

Using a similar cycle analysis as the above proof, we have the following negative result for Simplicial WL (Bodnar et al., 2021b) and Cellular WL (Bodnar et al., 2021a):

**Proposition C.12.** *Consider the SWL algorithm (Bodnar et al., 2021b), or more generally, the CWL algorithms with either $k$-CL, $k$-IC, or $k$-C as lifting maps ($k \geq 3$ is an integer) (Bodnar et al., 2021a, Definition 14). These algorithms can neither distinguish whether a given graph has cut vertices nor distinguish whether it has cut edges.*

*Proof.* Observe that the counterexample graphs in both Examples C.9 and C.10 do not have cliques. Therefore, SWL (or CWL with $k$-CL) reduces to the classic 1-WL and thus fails to distinguish them. Since the lengths of any cycles in these counterexample graphs are at least $m$ ($m$ is defined in Examples C.9 and C.10), we have that CWL with $k$-IC or $k$-C also reduces to 1-WL when $m > k$. Therefore, there exists graphs whose size is $O(k)$ such that CWL can neither distinguish cut vertices nor cut edges.

Finally, we point out that even if $k$ is not a constant (i.e., can scale to the graph size), CWL with $k$-IC still fails to distinguish whether a given graph has cut vertices. This is because for Example C.9 with $k \geq 2$ (e.g. Figure 2(b,c)), CWL with IC still outputs the same graph representation for both $G_1$ and $G_2$. This happens because all the 2-dimensional *cells* in these examples are cycles of an equal length of $m + 2$ and one can easily check that they have the same CWL color. □

We finally turn to the case of subgraph-based WL variants.

**Proposition C.13.** *The Overlap Subgraph WL (Wijesinghe & Wang, 2022) using any subgraph mapping $\omega$ can neither distinguish whether a given graph has cut vertices nor distinguish whether it has cut edges.*

*Proof.* An important limitation of OS-WL is that if a graph does not contain triangles, then any overlap subgraph $S_{uv}$ between two adjacent nodes $u, v$ will only have one edge $\{u, v\}$. Consequently, the subgraph mapping $\omega$ does not take effect can OS-WL reduces to the classic 1-WL. Therefore, Example C.9 with $m > 1$ and Example C.10 with $m > 3$ still apply here since the graphs $G_1$ and $G_2$ do not contain triangles (see Figure 2(a,b,d)). Moreover, Example C.9 with $m = 1$ (see Figure 2(c)) is also a counterexample as discussed in Wijesinghe & Wang (2022, Figure 2(a)). □

**Proposition C.14.** *The DSS-WL with ego network policy without marking cannot distinguish the graphs in Example C.9 with $m = 1$ (Figure 2(c)).*

*Proof.* First note that for any two vertices $u, v$ in either $G_1$ or $G_2$ defined in Example C.9, their shortest path distance does not exceed 2. Thus we only need to consider the ego network policy $\pi_{\mathrm{EGO}(1)}$ and $\pi_{\mathrm{EGO}(2)}$.

- For $\pi_{\mathrm{EGO}(2)}$, the ego graphs of all nodes are simply the original graph and thus all graphs in the bag $\mathcal{B}^\pi$ and equal. Thus DSS-WL reduces to the classic 1-WL and cannot distinguish $G_1$ and $G_2$.

- For $\pi_{\mathrm{EGO}(1)}$, the ego graph of each node $v \neq n$ is a graph with 5 edges, having a shape of two triangles sharing one edge. These ego graphs are clearly isomorphic. The ego graph of the special node $n$ is the original graph containing all edges. It is easy to see that the vertex partition of DSS-WL becomes stable only after one iteration, and the color mapping of $G_1$ and $G_2$ is the same. Therefore, DSS-WL cannot distinguish $G_1$ and $G_2$.

We thus conclude the proof. □

**Proposition C.15.** *The GNN-AK architecture proposed in Zhao et al. (2022) cannot distinguish whether a given graph has cut vertices.*

*Proof.* The GNN-AK architecture is quite similar to DSS-WL using the ego network policy but is weaker. There is also a subtle difference: GNN-AK adds the so-called centroid encoding. However, unlike node marking that is performed before the WL procedure, centroid encoding is performed after the WL procedure. The subtle difference causes GNN-AK to be unable to distinguish between the two graphs $G_1$ and $G_2$. □

We finally consider the DS-WL algorithm proposed in Cotta et al. (2021); Bevilacqua et al. (2022). As discussed in Appendix B.4, the original DS-WL formulation only outputs a graph representation rather than node colors. There are two simple ways to define nodes colors for DS-WL:

- If the graph generation policy $\pi$ is node-based, then each subgraph in $\mathcal{B}_G^\pi = \{\!\{G_i\}\!\}_{i=1}^{|\mathcal{V}|}$ is uniquely associated to a specific node $v \in \mathcal{V}$. We can thus use the graph representation of each subgraph $G_i$ as the color of each node. This strategy has appeared in prior works, e.g. Zhao et al. (2022).

- For a general graph generation policy $\pi$, there no longer exists an explicit bijective mapping between nodes and subgraphs. In this case, another possible way is to define $\chi_G(v) := \{\!\{\chi_{G_i}(v) : G_i \in \mathcal{B}_G^\pi\}\!\}$, similar to DSS-WL. This approach is recently introduced by Qian et al. (2022). However, such a strategy loses the memory advantage of DS-WL (i.e., needing $\Theta(|\mathcal{V}||\mathcal{B}_G^\pi|)$ memory complexity rather than $\Theta(|\mathcal{V}| + |\mathcal{B}_G^\pi|)$), and is less expressive than DSS-WL. We thus do not study this variant in the present work.

**Proposition C.16.** *The DS-WL algorithm with node marking/deletion policy cannot distinguish cut vertices when each node's color is defined as its associated subgraph representation.*

*Proof.* One can similarly check that for Example C.9 with $m = 1$ (see Figure 2(c)), the color of node $n$ will be the same for both graphs $G_1$ and $G_2$. Therefore, DS-WL cannot identify cut vertices. □

Finally, using a similar proof technique, the NGNN architecture proposed in Zhang & Li (2021) (with shortest path distance encoding) cannot identify cut vertices.

## C.3    PROOF OF THEOREM 3.2

**Theorem C.17.** *Let $G = (\mathcal{V}, \mathcal{E}_G)$ and $H = (\mathcal{V}, \mathcal{E}_H)$ be two graphs, and let $\chi_G$ and $\chi_H$ be the corresponding DSS-WL color mapping with node marking policy. Then the following holds:*

- *For any two nodes $w \in \mathcal{V}$ in $G$ and $x \in \mathcal{V}$ in $H$, if $\chi_G(w) = \chi_H(x)$, then $w$ is a cut vertex in graph $G$ if and only if $x$ is a cut vertex in graph $H$.*

- *For any two edges $\{w_1, w_2\} \in \mathcal{E}_G$ and $\{x_1, x_2\} \in \mathcal{E}_H$, if $\{\!\{\chi_G(w_1), \chi_G(w_2)\}\!\} = \{\!\{\chi_H(x_1), \chi_H(x_2)\}\!\}$, then $\{w_1, w_2\}$ is a cut edge if and only if $\{x_1, x_2\}$ is a cut edge.*

*Proof.* We divide the proof into two parts in Appendices C.3.1 and C.3.2, separately focusing on proving each bullet of Theorem 3.2. Before going into the proof, we first define several notations. Denote $\chi_G^u(v)$ as the color of node $v$ under the DSS-WL algorithm when marking $u$ as a special node. By definition of DSS-WL (Line 7 in Algorithm 3), $\chi_G(v) = \mathrm{hash}\left(\{\!\{\chi_G^u(v) : u \in \mathcal{V}\}\!\}\right)$. We can similarly define the inverse mappings $(\chi_G^u)^{-1}$.

We first present a lemma which can help us exclude the case of disconnected graphs.

**Lemma C.18.** *Given a node $w$, let $\mathcal{S}_G(w) \subset \mathcal{V}$ be the connected component in graph $G$ that comprises node $w$. For any two nodes $w \in \mathcal{V}$ in $G$ and $x \in \mathcal{V}$ in $H$, if $\chi_G(w) = \chi_H(x)$, then $\chi_{G[\mathcal{S}_G(w)]}(w) = \chi_{H[\mathcal{S}_H(x)]}(x)$.*

*Proof.* We first prove that if $\chi_G(w) = \chi_H(x)$, then $\{\!\{\chi_G^u(w) : u \in \mathcal{S}_G(w)\}\!\} = \{\!\{\chi_H^u(x) : u \in \mathcal{S}_H(x)\}\!\}$. First note that for any nodes $u, w$ in $G$ and $v, x$ in $H$, if $u \in \mathcal{S}_G(w)$ but $v \notin \mathcal{S}_H(x)$, then $\chi_G^u(w) \neq \chi_H^v(x)$. This is because DSS-WL only performs neighborhood aggregation, and the marking $v$ cannot propagate to node $x$ while the marking $u$ can propagate to node $w$. By definition we have

$$\chi_G(w) = \mathrm{hash}\left(\{\!\{\chi_G^u(w) : u \in \mathcal{S}_G(w)\}\!\} \cup \{\!\{\chi_G^v(w) : v \notin \mathcal{S}_G(w)\}\!\}\right).$$

Similarly,

$$\chi_H(x) = \mathrm{hash}\left(\{\!\{\chi_H^u(x) : u \in \mathcal{S}_H(x)\}\!\} \cup \{\!\{\chi_H^v(x) : v \notin \mathcal{S}_H(x)\}\!\}\right).$$

Since $\chi_G(w) = \chi_H(x)$, we have $\{\!\{\chi_G^u : u \in \mathcal{S}_G(w)\}\!\} = \{\!\{\chi_H^u : u \in \mathcal{S}_H(x)\}\!\}$. This clearly implies $\{\!\{\chi_{G[\mathcal{S}_G(w)]}^u : u \in \mathcal{S}_G(w)\}\!\} = \{\!\{\chi_{H[\mathcal{S}_H(x)]}^u : u \in \mathcal{S}_H(x)\}\!\}$, and thus $\chi_{G[\mathcal{S}_G(w)]}(w) = \chi_{H[\mathcal{S}_H(x)]}(x)$. $\square$

Note that $w$ is a cut vertex in $G$ implies $w$ is a cut vertex in $G[\mathcal{S}_G(w)]$. Therefore, based on Lemma C.18, we can restrict our attention to subgraphs $G[\mathcal{S}_G(w)]$ and $H[\mathcal{S}_H(x)]$ instead of the original (potentially disconnected) graphs. In other words, in the subsequent proof we can simply assume that *both graphs $G$ and $H$ are connected.*

We next present several simple but important properties regrading the DSS-WL color mapping as well as the subgraph color mappings.

**Lemma C.19.** *Let $u, w$ be two nodes in connected graph $G$ and $v, x$ be two nodes in connected graph $H$. Then the following holds:*

(a) *If $w = u$ and $x \neq v$, then $\chi_G^u(w) \neq \chi_H^v(x)$;*

(b) *If $\chi_G^u(w) = \chi_H^v(x)$, then $\chi_G(w) = \chi_H(x)$;*

(c) *If $\chi_G^u(w) = \chi_H^v(x)$, then $\chi_G(u) = \chi_H(v)$;*

(d) *$\chi_G(w) = \chi_H(x)$ if and only if $\chi_G^w(w) = \chi_H^x(x)$;*

(e) *If $\chi_G^u(w) = \chi_H^v(x)$, then $\mathrm{dis}_G(u, w) = \mathrm{dis}_H(v, x)$.*

*Proof.* Item (a) holds because in DSS-WL, the node with marking cannot have the same color as a node without marking. This can be rigorously proved by induction over the iteration $t$ in the DSS-WL algorithm (Line 6 in Algorithm 3).

Item (b) simply follows by definition of the DSS-WL aggregation procedure since the color $\chi_G^u(w)$ encodes the color of $\chi_G(w)$.

We next prove item (c), which follows by using the WL-condition of DSS-WL algorithm (Proposition C.3). Since $G$ is connected, there is a path from $w$ to $u$. Therefore, in graph $H$ there is also a path from $x$ to some node $v'$ satisfying $\chi_G^u(u) = \chi_H^v(v')$ (Lemma C.7). Now using item (a), it can only be the case $v' = v$ and thus $\chi_G^u(u) = \chi_H^v(v)$. Finally, by item (b) we obtain the desired result.

We next prove item (d). On the one hand, item (b) already shows that $\chi_G^w(w) = \chi_G^x(x) \implies \chi_G(w) = \chi_H(x)$. On the other hand, by definition of the DSS-WL algorithm,

$$\chi_G(w) = \mathrm{hash}\left(\{\!\{\chi_G^w(w)\}\!\} \cup \{\!\{\chi_G^u(w) : u \in \mathcal{V} \backslash \{w\}\}\!\}\right),$$
$$\chi_H(x) = \mathrm{hash}\left(\{\!\{\chi_H^x(x)\}\!\} \cup \{\!\{\chi_H^v(x) : v \in \mathcal{V} \backslash \{x\}\}\!\}\right).$$

Since $\chi_G(w) = \chi_H(x)$ and $\chi_G^w(w) \neq \chi_H^v(x)$ holds for all $v \in \mathcal{V} \backslash \{x\}$ (by item (a)), we obtain $\chi_G^w(w) = \chi_G^x(x)$.

We finally prove item (e), which again can be derived from the WL-condition of DSS-WL algorithm. If $\chi_G^u(w) = \chi_H^v(x)$, then by Corollary C.8 we have $\mathrm{dis}_G(w, (\chi_G^u)^{-1}(\chi_G^u(u))) = \mathrm{dis}_H(x, (\chi_H^v)^{-1}(\chi_G^u(u)))$. Using item (a), we have $(\chi_G^u)^{-1}(\chi_G^u(u)) = \{u\}$ and for any $v' \neq v$, $\chi_H^v(v') \neq \chi_H^v(v)$. Therefore, it can only be the case that $(\chi_H^v)^{-1}(\chi_G^u(u)) = \{v\}$ and $\chi_H^v(v) = \chi_G^u(u)$. This yields $\mathrm{dis}_G(u, w) = \mathrm{dis}_G(v, x)$ and concludes the proof. □

### C.3.1 Proof for the first part of Theorem 3.2

The following technical lemma is useful in the subsequent proof:

**Lemma C.20.** *Let $u, v \in \mathcal{V}$ be two nodes in connected graphs $G$ and $H$, respectively. If $\chi_G^u(u) = \chi_H^v(v)$, then $\{\!\{\chi_G^u(w) : w \in \mathcal{V}\}\!\} = \{\!\{\chi_H^v(w) : w \in \mathcal{V}\}\!\}$.*

*Proof.* Let $\mathcal{N}_G^d(u) := \{w \in \mathcal{V} : \mathrm{dis}_G(u, w) = d\}$ be the $d$-hop neighbors of node $u$ in graph $G$, and denote $\mathcal{C}_G^d := \{\!\{\chi_G^u(w) : w \in \mathcal{N}_G^d(u)\}\!\}$ as the multiset containing the color of all nodes $w$ with distance $d$ to node $u$. We can similarly denote $\mathcal{N}_H^d(v) := \{w : \mathrm{dis}_H(v, w) = d\}$ and $\mathcal{C}_H^d = \{\!\{\chi_H^v(w) : w \in \mathcal{N}_H^d(v)\}\!\}$. It suffices to prove that for all $d \in \mathbb{N}_+$, $\mathcal{C}_G^d = \mathcal{C}_H^d$.

We will prove the above result by induction. The case of $d = 0$ is trivial. Now suppose the case of $d$ is true (i.e., $\mathcal{C}_G^d = \mathcal{C}_H^d$) and we want to prove $\mathcal{C}_G^{d+1} = \mathcal{C}_H^{d+1}$. Note that for any nodes $x_1, x_2$ satisfying $\chi_G^u(x_1) = \chi_H^u(x_2)$, $\{\!\{\chi_G^u(w) : w \in \mathcal{N}_G(x_1)\}\!\} = \{\!\{\chi_H^u(w) : w \in \mathcal{N}_H(x_2)\}\!\}$. Therefore, by the induction assumption $\mathcal{C}_G^d = \mathcal{C}_H^d$,

$$\bigcup_{x \in \mathcal{N}_G^d(u)} \{\!\{\chi_G^u(w) : w \in \mathcal{N}_G(x)\}\!\} = \bigcup_{x \in \mathcal{N}_H^d(v)} \{\!\{\chi_H^v(w) : w \in \mathcal{N}_H(x)\}\!\}.$$

We next claim that $\mathcal{C}_G^d \cap \mathcal{C}_G^{d'} = \emptyset$ for any $d \neq d'$. This is because for any nodes $w_1$ and $w_2$ with the same color $\chi_G^u(w_1) = \chi_G^u(w_2)$, by Lemma C.19(e) we have $\mathrm{dis}_G(w_1, u) = \mathrm{dis}_G(w_2, u)$. Using this property, we obtain

$$\bigcup_{x \in \mathcal{N}_G^d(u)} \{\!\{\chi_G^u(w) : w \in \mathcal{N}_G(x) \cap \mathcal{N}_G^{d+1}(u)\}\!\} = \bigcup_{x \in \mathcal{N}_H^d(v)} \{\!\{\chi_H^v(w) : w \in \mathcal{N}_H(x) \cap \mathcal{N}_H^{d+1}(v)\}\!\}.$$

It is equivalent to the following equation:

$$\bigcup_{w \in \mathcal{N}_G^{d+1}(u)} \{\!\{\chi_G^u(w)\}\!\} \times |\mathcal{N}_G(w) \cap \mathcal{N}_G^d(u)| = \bigcup_{w \in \mathcal{N}_H^{d+1}(v)} \{\!\{\chi_H^v(w)\}\!\} \times |\mathcal{N}_H(w) \cap \mathcal{N}_H^d(v)|.$$

where $\{\!\{c\}\!\} \times m$ is a multiset containing $m$ repeated elements $c$. Finally, observe that if $\chi_G^u(w_1) = \chi_H^v(w_2)$ for some nodes $w_1$ and $w_2$, then $|\mathcal{N}_G(w_1) \cap \mathcal{N}_G^d(u)| = |\mathcal{N}_H(w_2) \cap \mathcal{N}_H^d(v)|$ (because $\mathcal{C}_G^d \cap \mathcal{C}_G^{d'} = \emptyset$ for any $d \neq d'$). Consequently, $\{\!\{\chi_G^u(w) : w \in \mathcal{N}_G^{d+1}(u)\}\!\} = \{\!\{\chi_H^v(w) : w \in \mathcal{N}_H^{d+1}(v)\}\!\}$, namely $\mathcal{C}_G^{d+1} = \mathcal{C}_H^{d+1}$. We have thus completed the proof of the induction step. □

We now present the following key result, which shows an important property of the color mapping for DSS-WL:

**Corollary C.21.** *Let $u, v \in \mathcal{V}$ be two nodes in connected graph $G$ with the same DSS-WL color, i.e. $\chi_G(u) = \chi_G(v)$. Then for any color $c \in \mathcal{C}$, $\{\!\{\chi_G^u(w) : w \in \chi_G^{-1}(c)\}\!\} = \{\!\{\chi_G^v(w) : w \in \chi_G^{-1}(c)\}\!\}$.*

*Proof.* First observe that if $\chi_G(u) = \chi_G(v)$, then $\chi_G^u(u) = \chi_G^v(v)$ (by Lemma C.19(d)). Consequently, $\{\!\{\chi_G^u(w) : w \in \mathcal{V}\}\!\} = \{\!\{\chi_G^v(w) : w \in \mathcal{V}\}\!\}$ holds by Lemma C.20. If $\{\!\{\chi_G^u(w) : w \in \chi_G^{-1}(c)\}\!\} \neq \{\!\{\chi_G^v(w) : w \in \chi_G^{-1}(c)\}\!\}$, then there must exist two nodes $w_1 \in \chi_G^{-1}(c)$ and $w_2 \notin \chi_G^{-1}(c)$, such that $\chi_G^u(w_1) = \chi_G^v(w_2)$. Therefore, by Lemma C.19(b) we have $\chi_G(w_1) = \chi_G(w_2)$, yielding a contradiction. □

In the subsequent proof, we assume the connected graph $G$ is not vertex-biconnected and let $u \in \mathcal{V}$ be a cut vertex in $G$. Let $\{\mathcal{S}_i\}_{i=1}^m$ ($m \geq 2$) be the partition of the vertex set $\mathcal{V} \backslash \{u\}$, representing each connected component after removing node $u$.

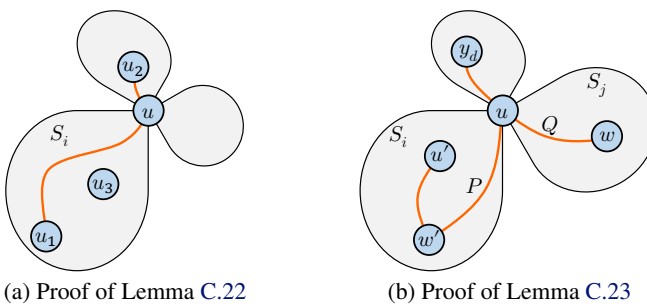

| (a) Proof of Lemma C.22 | (b) Proof of Lemma C.23 |

Figure 4: Several illustrations to help understand the lemmas.

**Lemma C.22.** *There is at most one set $\mathcal{S}_i$ satisfying $\mathcal{S}_i \cap \chi_G^{-1}(\chi_G(u)) \neq \emptyset$. In other words, if $\mathcal{S}_i \cap \chi_G^{-1}(\chi_G(u)) \neq \emptyset$ for some $i \in [m]$, then for any $j \in [m]$ and $j \neq i$, $\mathcal{S}_j \cap \chi_G^{-1}(\chi_G(u)) = \emptyset$.*

*Proof.* When $|\chi_G^{-1}(\chi_G(u))| = 1$, the conclusion clearly holds. If $|\chi_G^{-1}(\chi_G(u))| > 1$, then we can pick a node $u_1 \in \chi_G^{-1}(\chi_G(u))$ that maximizes the shortest path distance $\mathrm{dis}_G(u_1, u)$. Let $u_1 \in \mathcal{S}_i$ for some $i \in [m]$. If the lemma does not hold, then we can pick another node $u_2 \in \chi_G^{-1}(\chi_G(u))$ and $u_2 \notin \mathcal{S}_i$. Since $u_1$ and $u_2$ are in different connected component after removing $u$, $\mathrm{dis}_G(u_1, u_2) = \mathrm{dis}_G(u_1, u) + \mathrm{dis}_G(u_2, u)$. See Figure 4(a) for an illustration of this paragraph.

By Corollary C.21, $\{\!\{\chi_G^{u_1}(w) : w \in \chi_G^{-1}(\chi_G(u))\}\!\} = \{\!\{\chi_G^u(w) : w \in \chi_G^{-1}(\chi_G(u))\}\!\}$. Therefore, there must exist a node $u_3 \in \chi_G^{-1}(\chi_G(u))$ satisfying $\chi_G^{u_1}(u_2) = \chi_G^u(u_3)$. We thus have $\mathrm{dis}_G(u_2, u_1) = \mathrm{dis}_G(u_3, u)$ by Lemma C.19(e). On the other hand, by definition of the node $u_1$, $\mathrm{dis}_G(u_1, u) \geq \mathrm{dis}_G(u_3, u)$. Therefore, $\mathrm{dis}_G(u_2, u_1) = \mathrm{dis}_G(u_1, u) + \mathrm{dis}_G(u_2, u) > \mathrm{dis}_G(u_3, u)$. This yields a contradiction and concludes the proof. $\square$

**Lemma C.23.** *For all $u' \in \chi_G^{-1}(\chi_G(u))$, $u'$ it is a cut vertex of $G$.*

*Proof.* When $|\chi_G^{-1}(\chi_G(u))| = 1$, the conclusion clearly holds. Now assume $|\chi_G^{-1}(\chi_G(u))| > 1$. Since $u$ is a cut vertex in $G$, by Lemma C.22, there exists a set $\mathcal{S}_j$ such that $\mathcal{S}_j \cap \chi_G^{-1}(\chi_G(u)) = \emptyset$. Pick any node $w \in \mathcal{S}_j$, then $\chi_G(w) \neq \chi_G(u)$. Let $u' \neq u$ be any node with color $\chi_G(u) = \chi_G(u')$. It follows that $\chi_G^u(u) = \chi_G^{u'}(u')$ by Lemma C.19(d). Based on the WL-condition of the mappings $\chi_G^u$ and $\chi_G^{u'}$, by Lemma C.7 there exists a node $w'$ with color $\chi_G^{u'}(w') = \chi_G^u(w)$ (because there is a path from node $u$ to $w$). See Figure 4(b) for an illustration of this paragraph.

Suppose $u'$ is not a cut vertex. Then there is a path $P$ from $w'$ to $u$ without going through node $u'$. Denote $P = (x_0, \cdots, x_d)$ where $x_0 = w'$ and $x_d = u$. It follows that $\chi_G^{u'}(x_i) \neq \chi_G^{u'}(u')$ for all $i \in [d]$ (by Lemma C.19(a)). Again by using the WL-condition, there exists a path $Q = (y_0, \cdots, y_d)$ satisfying $y_0 = w$ and $\chi_G^u(y_i) = \chi_G^{u'}(x_i)$ for all $i \in [d]$. In particular, $\chi_G^u(y_d) = \chi_G^{u'}(u)$, which implies $\chi_G(y_d) = \chi_G(u)$ by using Lemma C.19(b). By the definition of $w$ and Lemma C.22, any path from $w$ to $y_d \in \chi_G^{-1}(\chi_G(u))$ must go through node $u$, implying that $\chi_G^u(y_i) = \chi_G^u(u)$ for some $i \in [d]$. However, we have proved that $\chi_G^u(y_i) = \chi_G^{u'}(x_i) \neq \chi_G^{u'}(u') = \chi_G^u(u)$, yielding a contradiction. Therefore, $u'$ is a cut vertex. $\square$

Using a similar proof technique as the one in Lemma C.23, we can prove the first part of Theorem 3.2. Suppose $u' \in \chi_H^{-1}(\chi_G(u))$ and we want to prove that $u'$ is a cut vertex of graph $H$. Observe that $|\chi_G^{-1}(\chi_G(u))| = |\chi_H^{-1}(\chi_H(u))|$. (A simple proof is as follows: $\chi_G(u) = \chi_H(u')$ implies $\chi_G^u(u) = \chi_H^{u'}(u')$ by Lemma C.19(d), and thus using Lemma C.20 we have $\{\!\{\chi_G^u(w) : w \in \mathcal{V}\}\!\} = \{\!\{\chi_H^{u'}(w) : w \in \mathcal{V}\}\!\}$ and finally obtain $\{\!\{\chi_G(w) : w \in \mathcal{V}\}\!\} = \{\!\{\chi_H(w) : w \in \mathcal{V}\}\!\}$ by Lemma C.19(b).)

We first consider the case when $|\chi_G^{-1}(\chi_G(u))| = |\chi_H^{-1}(\chi_H(u))| > 1$. Following the above proof, we can similarly pick $w \in \mathcal{S}_j$ in $G$ and $w'$ in $H$ satisfying $\chi_G(w) \neq \chi_G(u)$ and $\chi_H^{u'}(w') = \chi_G^u(w)$. Since $|\chi_G^{-1}(\chi_G(u))| > 1$, we can pick a node $u_H \in \chi_H^{-1}(\chi_G(u))$ in $H$ such that $u_H \neq u'$. If $u'$ is

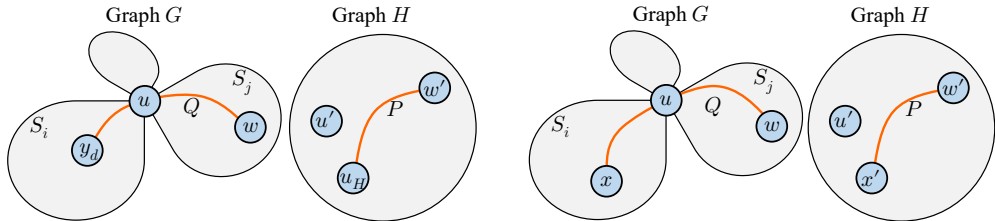

(a) Proof of the main theorem ($|\chi_G^{-1}(\chi_G(u))| > 1$)    (b) Proof of the main theorem ($|\chi_G^{-1}(\chi_G(u))| = 1$)

Figure 5: Several illustrations to help understand the main proof of Theorem 3.2.

not a cut vertex, then there is a path $P = (x_0, \cdots, x_d)$ in $H$ where $x_0 = w'$ and $x_d = u_H$, such that $\chi_H^{u'}(x_i) \neq \chi_H^{u'}(u')$ for all $i \in [d]$ (by Lemma C.19(a)). Using the WL-condition, there exists a path $Q = (y_0, \cdots, y_d)$ in $G$ satisfying $y_0 = w$ and $\chi_G^u(y_i) = \chi_H^{u'}(x_i)$ for all $i \in [d]$. In particular, $\chi_G^u(y_d) = \chi_H^{u'}(u_H)$, which implies $\chi_G(y_d) = \chi_G(u_H)$ by using Lemma C.19(b). However, any path from $w$ to $y_d \in \chi_G^{-1}(\chi_G(u))$ must go through node $u$, implying that $\chi_G^u(y_i) = \chi_G^u(u)$ for some $i \in [d]$. This yields a contradiction because $\chi_G^u(y_i) = \chi_H^{u'}(x_i) \neq \chi_H^{u'}(u') = \chi_G^u(u)$. See Figure 5(a) for an illustration of this paragraph.

We finally consider the case when $|\chi_G^{-1}(\chi_G(u))| = |\chi_H^{-1}(\chi_H(u))| = 1$. Let $w \in \mathcal{S}_1$ and $x \in \mathcal{S}_2$ be two nodes in $G$ that belongs to different connected components when removing node $u$, then $\chi_G(w) \neq \chi_G(u)$ and $\chi_G(x) \neq \chi_G(u)$. Since $\chi_G(u) = \chi_H(u')$, by the WL-condition (Lemma C.7) there is a node $w' \in \chi_H^{-1}(\chi_G(w))$ in $H$. Consequently, $\chi_G^w(w) = \chi_H^{w'}(w')$ (Lemma C.19(d)). Again by the WL-condition, there is a node $x' \in (\chi_H^{w'})^{-1}(\chi_G^w(x))$ in $H$. Clearly, $w' \neq u'$ and $x' \neq u'$ (because they have different colors). If $u'$ is not a cut vertex, then there is path $P = (y_0, \cdots, y_d)$ in $H$ such that $y_0 = x'$, $y_d = w'$ and $y_i \neq u'$ for all $i \in [d]$. It follows that for all $i \in [d]$, $\chi_H(y_i) \neq \chi_H(u')$ by our assumption $|\chi_H^{-1}(\chi_H(u))| = 1$, and thus $\chi_H^{w'}(y_i) \neq \chi_H^{w'}(u')$ (by Lemma C.19(b)). Since $\chi_G^w(x) = \chi_H^{w'}(x')$, by the WL-condition (Lemma C.7), there is a path $Q = (z_0, \cdots, z_d)$ in $G$ satisfying $z_0 = x$ and $z_i \in (\chi_G^w)^{-1}(\chi_H^{w'}(y_i))$ for $i \in [d]$. See Figure 5(b) for an illustration of this paragraph.

Clearly, we have $z_d = w$ using $\chi_G^w(z_d) = \chi_H^{w'}(w')$ and Lemma C.19(a). On the other hand, by Lemma C.19(b), $\chi_G^w(z_i) = \chi_H^{w'}(y_i)$ implies $\chi_G(z_i) = \chi_H(y_i)$ and thus $\chi_G(z_i) \neq \chi_H(u') = \chi_G(u)$ holds for all $i \in [d]$ and thus $z_i \neq u$. In other words, we have found a path from $x$ to $w$ without going through node $u$, which yields a contradiction as $u$ is a cut vertex. We have thus finished the proof.

### C.3.2 PROOF FOR THE SECOND PART OF THEOREM 3.2

The proof is based on the following key result:

**Corollary C.24.** *Let $w$ and $x$ be two nodes in connected graph $G$ with the same DSS-WL color, i.e. $\chi_G(w) = \chi_G(x)$. Then for any color $c \in \mathcal{C}$,*

$$\{\!\{\mathrm{dis}_G(w,v) : v \in \chi_G^{-1}(c)\}\!\} = \{\!\{\mathrm{dis}_G(x,v) : v \in \chi_G^{-1}(c)\}\!\}.$$

*Proof.* By Corollary C.21, we have $\{\!\{\chi_G^w(v) : v \in \chi_G^{-1}(c)\}\!\} = \{\!\{\chi_G^x(v) : v \in \chi_G^{-1}(c)\}\!\}$. Since for any nodes $u, v$, $\chi_G^w(u) = \chi_G^x(v)$ implies $\mathrm{dis}_G(u,w) = \mathrm{dis}_G(v,x)$ (by Lemma C.19(e)), we have obtained the desired conclusion. $\square$

Equivalently, the above corollary says that if $\chi_G(w) = \chi_G(x)$, then the following two multisets are equivalent:

$$\{\!\{(\mathrm{dis}_G(w,v), \chi_G(v)) : v \in \mathcal{V}\}\!\} = \{\!\{(\mathrm{dis}_G(x,v), \chi_G(v)) : v \in \mathcal{V}\}\!\}.$$

Therefore, it guarantees that the vertex partition induced by the DSS-WL color mapping is *finer* than that of the SPD-WL (Algorithm 4 with $d_G = \mathrm{dis}_G$). We can thus invoke Theorem 4.1, which directly concludes the proof (due to Proposition C.56).

## C.4 PROOF OF THEOREM 4.1

**Theorem C.25.** *Let $G = (\mathcal{V}, \mathcal{E}_G)$ and $H = (\mathcal{V}, \mathcal{E}_H)$ be two graphs, and let $\chi_G$ and $\chi_H$ be the corresponding SPD-WL color mapping. Then the following holds:*

- *For any two edges $\{w_1, w_2\} \in \mathcal{E}_G$ and $\{x_1, x_2\} \in \mathcal{E}_H$, if $\{\!\{\chi_G(w_1), \chi_G(w_2)\}\!\} = \{\!\{\chi_H(x_1), \chi_H(x_2)\}\!\}$, then $\{w_1, w_2\}$ is a cut edge if and only if $\{x_1, x_2\}$ is a cut edge.*

- *If the graph representations of $G$ and $H$ are the same under SPD-WL, then their block cut-edge trees (Definition 2.3) are isomorphic. Mathematically, $\{\!\{\chi_G(w) : w \in \mathcal{V}\}\!\} = \{\!\{\chi_H(w) : w \in \mathcal{V}\}\!\}$ implies that $\mathrm{BCETree}(G) \simeq \mathrm{BCETree}(H)$.*

*Proof Sketch.* The proof of Theorem 4.1 is highly non-trivial and is divided into three parts (presented in Appendices C.4.1 to C.4.3, respectively). We first consider the special setting when both $G$ and $H$ are connected and $\{\!\{\chi_G(w) : w \in \mathcal{V}\}\!\} = \{\!\{\chi_H(w) : w \in \mathcal{V}\}\!\}$. Assume $G$ is not edge-biconnected, and let $\{u, v\} \in \mathcal{E}_G$ be a cut edge in $G$. We separately consider two cases: $\chi_G(u) \neq \chi_G(v)$ (Appendix C.4.1) and $\chi_G(u) = \chi_G(v)$ (Appendix C.4.2), and prove that any edge $\{u', v'\} \in \mathcal{E}_H$ satisfying $\{\!\{\chi_G(u), \chi_G(v)\}\!\} = \{\!\{\chi_H(u'), \chi_H(v')\}\!\}$ is also a cut edge of $H$. This basically finishes the proof of the first bullet in the theorem. Finally, we consider the general setting where graphs $G, H$ can be disconnected and their representation is not the same in Appendix C.4.3, and complete the proof of Theorem 4.1.

Without abuse of notation, throughout Appendices C.4.1 and C.4.2 we redefine the color set $\mathcal{C} := \{\chi_G(w) : w \in \mathcal{V}\} = \{\chi_H(w) : w \in \mathcal{V}\}$ to focus only on colors that are present in $G$ (or $H$), rather than all (irrelevant) colors in the range of a hash function.

### C.4.1 THE CASE OF $\chi_G(u) \neq \chi_G(v)$ FOR CONNECTED GRAPHS

We first define several notations. Throughout this case, denote $\{\mathcal{S}_u, \mathcal{S}_v\}$ as the partition of $\mathcal{V}$, representing the two connected components after removing the edge $\{u, v\}$ such that $u \in \mathcal{S}_u$, $v \in \mathcal{S}_v$, $\mathcal{S}_u \cap \mathcal{S}_v = \emptyset$ and $\mathcal{S}_u \cup \mathcal{S}_v = \mathcal{V}$. We then define an important concept called the color graph.

**Definition C.26.** (Color graph) Define the auxiliary color graph $G^{\mathrm{C}} = (\mathcal{C}, \mathcal{E}_{G^{\mathrm{C}}})$ where $\mathcal{E}_{G^{\mathrm{C}}} = \{\{\!\{\chi_G(w), \chi_G(x)\}\!\} : \{w, x\} \in E_G\}$. Note that $G^{\mathrm{C}}$ can have self loops, so each edge is denoted as a multiset with two elements.

**Lemma C.27.** *Let $\mathcal{S} = \chi_G^{-1}(\chi_G(u)) \cup \chi_G^{-1}(\chi_G(v))$ be the set containing vertices with color $\chi_G(u)$ or $\chi_G(v)$. Then either $\mathcal{S} \cap \mathcal{S}_u = \{u\}$ or $\mathcal{S} \cap \mathcal{S}_v = \{v\}$.*

*Proof.* Assume the lemma does not hold, i.e. $|\mathcal{S} \cap \mathcal{S}_u| > 1$ and $|\mathcal{S} \cap \mathcal{S}_v| > 1$. We first prove that $\chi_G^{-1}(\chi_G(u)) \cap \mathcal{S}_v \neq \emptyset$ and $\chi_G^{-1}(\chi_G(v)) \cap \mathcal{S}_u \neq \emptyset$. By symmetry, we only need to prove the former. Suppose $\chi_G^{-1}(\chi_G(u)) \cap \mathcal{S}_v = \emptyset$, then $(\chi_G^{-1}(\chi_G(v)) \cap \mathcal{S}_v) \backslash \{v\} \neq \emptyset$ (because $|\mathcal{S} \cap \mathcal{S}_v| > 1$), and thus there exists $v' \in \mathcal{S}_v$, $v' \neq v$ such that $\chi_G(v') = \chi_G(v)$. Note that $v'$ must connect to a node $u'$ with $\chi_G(u') = \chi_G(u)$. Since $\{u, v\}$ is a cut edge in $G$, $u' \in \mathcal{S}_v$. Therefore, $\chi_G^{-1}(\chi_G(u)) \cap \mathcal{S}_v \neq \emptyset$, yielding a contradiction. This paragraph is illustrated in Figure 6(a).

We next prove that at least one of the following two conditions holds (which are symmetric): (i) $(\chi_G^{-1}(\chi_G(u)) \cap \mathcal{S}_u) \backslash \{u\} \neq \emptyset$; (ii) $(\chi_G^{-1}(\chi_G(v)) \cap \mathcal{S}_v) \backslash \{v\} \neq \emptyset$. Based on the above paragraph, there exists $v' \in \mathcal{S}_u$ satisfying $\chi_G(v') = \chi_G(v)$. Note that $v'$ must connect to a node with color $\chi_G(u)$. If condition (i) does not hold, i.e. $\chi_G^{-1}(\chi_G(u)) \cap \mathcal{S}_u = \{u\}$, then $v'$ must connect to $u$. This means $|\mathcal{N}_G(u) \cap \chi_G^{-1}(\chi_G(v))| \geq 2$. Again using $\chi_G^{-1}(\chi_G(u)) \cap \mathcal{S}_v \neq \emptyset$ (the above paragraph), we can pick such a node $u' \in \chi_G^{-1}(\chi_G(u)) \cap \mathcal{S}_v$. By the WL-condition (Proposition C.4), $|\mathcal{N}_G(u') \cap \chi_G^{-1}(\chi_G(v))| \geq 2$, which implies $|\mathcal{S}_v \cap \chi_G^{-1}(\chi_G(v))| \geq 2$. Thus $(\chi_G^{-1}(\chi_G(v)) \cap \mathcal{S}_v) \backslash \{v\} \neq \emptyset$ holds, which is exactly the condition (ii). This paragraph is illustrated in Figure 6(b).

Based on the above two paragraphs, by symmetry we can without loss of generality assume $\chi_G^{-1}(\chi_G(u)) \cap \mathcal{S}_v \neq \emptyset$ and $(\chi_G^{-1}(\chi_G(u)) \cap \mathcal{S}_u) \backslash \{u\} \neq \emptyset$. We are now ready to derive a contradiction. To do this, pick $\tilde{u} = \arg\max_{w \in \chi_G^{-1}(\chi_G(u))} \mathrm{dis}_G(u, w)$ and separately consider the following two cases:

- $\tilde{u} \in \mathcal{S}_u$. Then by picking a node $x \in \mathcal{S}_v \cap \chi_G^{-1}(\chi_G(u))$, it follows that $\mathrm{dis}_G(x, \tilde{u}) = \mathrm{dis}_G(x, v) + \mathrm{dis}_G(u, \tilde{u}) + 1 > \mathrm{dis}_G(u, \tilde{u})$.

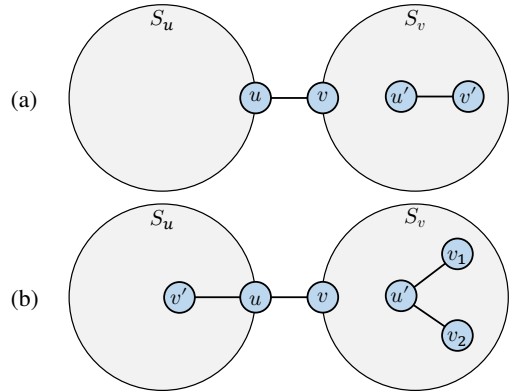

(a)

(b)

Figure 6: Illustration of the proof of Lemma C.27.

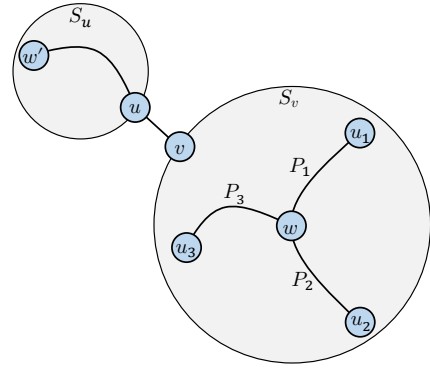

Figure 7: Illustration of the proof of Lemma C.28.

- $\tilde{u} \in \mathcal{S}_v$. Then by picking a node $x \in (\mathcal{S}_u \cap \chi_G^{-1}(\chi_G(u))) \backslash \{u\}$, it follows that $\operatorname{dis}_G(x, \tilde{u}) \geq \operatorname{dis}_G(x, u) + \operatorname{dis}_G(u, \tilde{u}) > \operatorname{dis}_G(u, \tilde{u})$.

In both cases, $x$ and $u$ cannot have the same color under SPD-WL because

$$\max_{w \in \chi_G^{-1}(\chi_G(u))} \operatorname{dis}_G(u, w) = \operatorname{dis}_G(u, \tilde{u}) < \operatorname{dis}_G(x, \tilde{u}) \leq \max_{w \in \chi_G^{-1}(\chi_G(u))} \operatorname{dis}_G(x, w).$$

This yields a contradiction and concludes the proof. $\qquad\square$

Based on Lemma C.27, in the subsequent proof we can without loss of generality assume $\chi_G^{-1}(\chi_G(u)) \cap \mathcal{S}_u = \{u\}$ and $\chi_G^{-1}(\chi_G(v)) \cap \mathcal{S}_u = \emptyset$. This leads to the following lemma:

**Lemma C.28.** *For any $u_1, u_2 \in \chi_G^{-1}(\chi_G(u))$, $u_1 \neq u_2$, any path from $u_1$ to $u_2$ goes through a node $v' \in \chi_G^{-1}(\chi_G(v))$.*

*Proof.* Note that $\chi_G^{-1}(\chi_G(u)) \cap \mathcal{S}_u = \{u\}$. If $|\chi_G^{-1}(\chi_G(u)) \cap \mathcal{S}_v| \leq 1$, the conclusion is clear since any path from $u_1$ to $u_2$ goes through $v$. Now suppose $|\chi_G^{-1}(\chi_G(u)) \cap \mathcal{S}_v| > 1$ and the lemma does not hold. Then there exist two different nodes $u_1', u_2' \in \chi_G^{-1}(\chi_G(u)) \cap \mathcal{S}_v$ and a path $P$ from $u_1'$ to $u_2'$ without going through any node in the set $\chi_G^{-1}(\chi_G(v))$. Pick $u_1$, $u_2$ and $P$ such that the length $|P|$ is minimal. Split $P$ into two parts $P_1$ and $P_2$ with endpoints $\{u_1, w\}$ and $\{w, u_2\}$ such that $|P_1| \leq |P_2| \leq |P_1| + 1$ and $|P_1| + |P_2| = |P|$. Note that $|P| \geq 2$ since $\{u_1, u_2\} \notin \mathcal{E}_G$ (otherwise $u$ cannot have the same color as $u_1$ because $\chi_G^{-1}(u) \cap \mathcal{S}_u = \{u\}$). Therefore, $w \neq u_1$ and $w \neq u_2$. Also note that $\chi_G(w) \neq \chi_G(u)$ since $|P|$ is minimal. Since SPD-WL satisfies the WL-condition (Proposition C.4), there is a path (not necessarily simple) from $u$ to some $w' \in \chi_G^{-1}(\chi_G(w))$ of length $|P_1|$ without going through nodes in the set $\chi_G^{-1}(\chi_G(v))$ (according to Lemma C.7). Therefore, $w' \in \mathcal{S}_u$. See Figure 6 for an illustration of this paragraph.

We next prove that $\operatorname{dis}_G(u, w') = |P_1|$. First, we obviously have $\operatorname{dis}_G(u, w') \leq |P_1|$. Moreover, since $w', u \in \mathcal{S}_u$ and $\chi_G^{-1}(\chi_G(v)) \cap \mathcal{S}_u = \emptyset$ (Lemma C.27), any shortest path from $w'$ to $u$ does not go through nodes in the set $\chi_G^{-1}(\chi_G(v))$. Again using the WL-condition, there exists a path $P_3$ (not necessarily simple) from $w$ to some $u_3 \in \chi_G^{-1}(\chi_G(u))$ of length $|P_3| = \operatorname{dis}_G(u, w')$ without going through nodes in the set $\chi_G^{-1}(\chi_G(v))$ (according to Lemma C.7). It follows that $u_3 \in \mathcal{S}_v$. Consider the following two cases:

- If $u_3 = u_1$, by the minimal length of $P$ we have $|P_1| \leq |P_3| = \operatorname{dis}_G(u, w') \leq |P_1|$ and thus $\operatorname{dis}_G(u, w') = |P_1|$.

- If $u_3 \neq u_1$, by linking the path $P_1$ and $P_3$, there will be a path of length $|P_1| + |P_3|$ from $u_1$ to $u_3$ without going through nodes in $\chi_G^{-1}(\chi_G(v))$. Since $P$ has the minimal length, $|P_1| + |P_2| \leq |P_1| + |P_3|$. Therefore, $|P_2| \leq |P_3| = \operatorname{dis}_G(u, w')$ and thus by definition $|P_1| \leq |P_2| \leq \operatorname{dis}_G(u, w') \leq |P_1|$. Therefore, $|P_1| = |P_2| = \operatorname{dis}_G(u, w')$.

Now define the set $\mathcal{D}(x) := \{u' : u' \in \chi_G^{-1}(\chi_G(u)), \mathrm{dis}_G(x, u') \le |P_2|\}$. Let us focus on the cardinality of the sets $\mathcal{D}(w)$ and $\mathcal{D}(w')$. It follows that $\mathcal{D}(w') = \{u\}$, because for any other node $u' \in \chi_G^{-1}(\chi_G(u))$, $u' \ne u$, we have $u' \in \mathcal{S}_v$ and thus

$$\mathrm{dis}_G(w', u') > \mathrm{dis}_G(w', v) = \mathrm{dis}_G(w', u) + 1 = |P_1| + 1 \ge |P_2|.$$

Therefore, $|\mathcal{D}(w')| = 1$. On the other hand, we clearly have $|\mathcal{D}(w)| \ge 2$ since both $u_1, u_2 \in \mathcal{D}(w)$. Consequently, $w$ and $w'$ cannot have the same color under the SPD-WL algorithm because $|\mathcal{D}(w')| \ne |\mathcal{D}(w')|$. This yields a contradiction and completes the proof. $\square$

The next lemma presents an important property of the color graph $G^{\mathrm{C}}$ (defined in Definition C.26).

**Lemma C.29.** $G^{\mathrm{C}}$ *has a cut edge* $\{\{\chi_G(u), \chi_G(v)\}\}$.

*Proof.* Suppose $\{\{\chi_G(u), \chi_G(v)\}\}$ is not a cut edge of $G^{\mathrm{C}}$. Then there is a simple cycle $(c_1, \cdots, c_m)$ where $c_1 = \chi_G(u)$, $c_m = \chi_G(v)$ and $m > 2$. Namely, there exists a simple path from $c_1$ to $c_m$ with length $\ge 2$. By the definition of $G^{\mathrm{C}}$ and the WL-condition, there exists a sequence of nodes of $G$ $\{w_i\}_{i=1}^m$ where $w_1 = u$ and $\chi(w_i) = c_i$ such that $\{w_i, w_{i+1}\} \in \mathcal{E}_G$, $i \in [m-1]$. Note that $w_i \ne u$ for $i = \{2, \cdots, m\}$ and $w_2 \ne v$ because $(c_1, \cdots, c_m)$ is a simple path. Therefore, $w_i \in \mathcal{S}_u$ for all $i \in [m]$. However, it contradicts $|\mathcal{S} \cap \mathcal{S}_u| = 1$ (Lemma C.27) since $\chi_G(w_m) = \chi_G(v)$. $\square$

Combining Lemmas C.27 to C.29, we arrived at the following corollary:

**Corollary C.30.** *For all* $u' \in \chi_G^{-1}(\chi_G(u))$ *and* $v' \in \chi_G^{-1}(\chi_G(v))$, *if* $\{u', v'\} \in \mathcal{E}_G$, *then it is a cut edge of* $G$.

*Proof.* If $\{u', v'\}$ is not a cut edge, there is a simple cycle going through $\{u', v'\}$. Denote it as $(w_1, \cdots, w_m)$ where $w_1 = u', w_m = v', m > 2$. By Lemma C.27, $w_2 \notin \chi_G(v)$, otherwise $u'$ will connect to at least two different nodes $w_2, w_m \in \chi_G^{-1}(\chi_G(v))$ and thus $u'$ and $u$ cannot have the same color under SPD-WL. Let $j$ be the index such that $j = \min\{j \in [m] : \chi_G(w_j) = \chi_G(v)\}$, then $j > 2$. Consider the path $(w_1, \cdots, w_j)$. It follows that $\chi_G(w_k) \ne \chi_G(u)$ for all $k \in \{2, \cdots, j\}$ by Lemma C.28 (otherwise there is a path from node $w_1$ to some node $w_i \in \chi_G^{-1}(\chi_G(u))$ ($i \in \{2, \cdots, j\}$) that does not go through nodes in the set $\chi_G^{-1}(\chi_G(v))$, a contradiction). Therefore, $(\chi_G(w_1), \cdots, \chi_G(w_j))$ is a path of length $\ge 2$ in $G^{\mathrm{C}}$ from $\chi_G(u)$ to $\chi_G(v)$ (not necessarily simple), without going through the edge $\{\{\chi_G(u), \chi_G(v)\}\}$. This contradicts Lemma C.29, which says that $\{\{\chi_G(u), \chi_G(v)\}\}$ is a cut edge in $G^{\mathrm{C}}$. $\square$

Based on Lemma C.29, the cut edge $\{\{\chi_G(u), \chi_G(v)\}\}$ partitions the vertices $\mathcal{C}$ of the color graph $G^{\mathrm{C}}$ into two classes. Denote them as $\{\mathcal{C}_u, \mathcal{C}_v\}$ where $\chi_G(u) \in \mathcal{C}_u$ and $\chi_G(v) \in \mathcal{C}_v$. The next corollary characterizes the structure of the node colors calculated in SPD-WL.

**Corollary C.31.** *For any* $w$ *satisfying* $\chi_G(w) \in \mathcal{C}_u$, *there exists a cut edge* $\{u', v'\}$, $u' \in \chi_G^{-1}(\chi_G(u))$, $v' \in \chi_G^{-1}(\chi_G(v))$, *that partitions* $\mathcal{V}$ *into two classes* $\mathcal{S}_{u'} \cup \mathcal{S}_{v'}$, $u', w \in \mathcal{S}_{u'}$, $v' \in \mathcal{S}_{v'}$, *such that* $\chi_G^{-1}(\chi_G(u')) \cup \mathcal{S}_{u'} = \{u'\}$ *and* $\chi_G^{-1}(\chi_G(v')) \cup \mathcal{S}_{u'} = \emptyset$.

**Remark C.32.** Corollary C.31 can be seen as a generalized version of Lemma C.27. Indeed, when $w \in \mathcal{S}_u$, one can pick $u' = u$ and $v' = v$. Then $\chi_G^{-1}(\chi_G(u')) \cup \mathcal{S}_{u'} = \{u'\}$ and $\chi_G^{-1}(\chi_G(v')) \cup \mathcal{S}_{u'} = \emptyset$ hold due to Lemma C.27. In general, Corollary C.31 says that all the cut edges with color $\{\chi_G(u), \chi_G(v)\}$ play an equal role: Lemma C.27 applies for any chosen cut edge $\{u', v'\}$. An illustration of Corollary C.31 is given in Figure 8(a).

*Proof.* By the definition of $\mathcal{C}_u$, any node $c \in \mathcal{C}_u$ in the *color graph* can reach the node $\chi_G(u)$ without going through $\chi_G(v)$. Therefore, there exists some $u' \in \chi_G^{-1}(\chi_G(u))$ such that there exists a path $P_1$ from $w$ to $u'$ without going through nodes in the set $\chi_G^{-1}(\chi_G(v))$. Also, there exists a node $v' \in \mathcal{N}_G(u')$ with $\chi_G(v') = \chi_G(v)$ due to the color of $u'$. By Corollary C.30, $\{u', v'\}$ is a cut edge of $G$. Clearly, $w \in \mathcal{S}_{u'}$.

We next prove the following fact: for any $x \in \mathcal{S}_{u'}$, $\chi_G(x) \in \mathcal{C}_u$. Otherwise, one can pick a node $x \in \mathcal{S}_{u'}$ with color $\chi_G(x) \in \mathcal{C}_v$. Consider the *shortest* path between nodes $x$ and $u'$, denoted as $(y_1, \cdots, y_m)$ where $y_1 = x$ and $y_m = u'$. It follows that $y_i \in \mathcal{S}_u$ for all $i \in [m]$. Denote

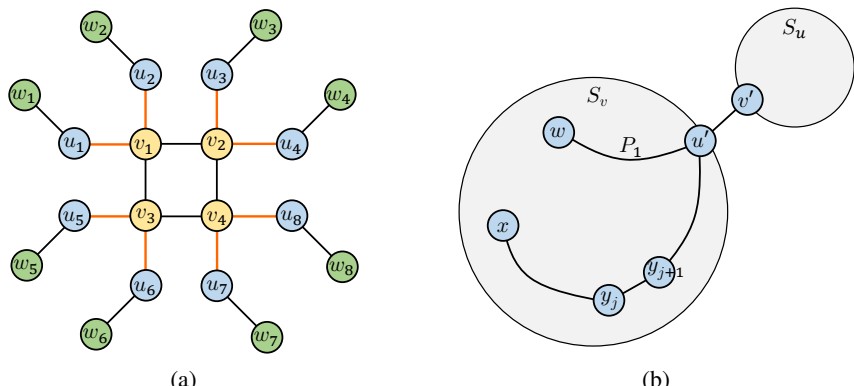

Figure 8: Illustration of Corollary C.31 and its proof.

$c_i = \chi_G(y_i), i \in [m]$. Then $(c_1, \cdots, c_m)$ is a path (not necessarily simple) in the color graph $G^C$. Now pick the index $j = \max\{j \in [m] : c_j \in \mathcal{C}_v\}$ (which is well-defined because $c_1 \in \mathcal{C}_v$). It follows that $j < m$ (since $y_m \in \mathcal{C}_u$), $c_j = \chi_G(v)$ and $c_{j+1} = \chi_G(u)$ (because $\{\{\chi_G(u), \chi_G(v)\}\}$ is a cut edge that partitions the color graph $G^C$ into $\mathcal{C}_u$ and $\mathcal{C}_v$). Consider the following two cases (see Figure 8(b) for an illustration):

- $j = m-1$. Then $u'$ connects to both nodes $y_j$ and $v'$ with color $\chi_G(y_j) = \chi_G(v') = \chi_G(v)$. This contradicts Lemma C.27 since $u$ only connects to one node $v$ with color $\chi_G(v)$.

- $j < m - 1$. Then $y_{j+1} \neq u'$ because the path $(y_1, \cdots, y_m)$ is simple. Howover, one has $\chi_G(y_i) \neq \chi_G(v)$ for all $i \in \{j+1, \cdots, m\}$ by definition of $j$. This contradicts Lemma C.28.

This completes the proof that for any $x \in \mathcal{S}_{u'}$, $\chi_G(x) \in \mathcal{C}_u$. Therefore, $\chi_G^{-1}(\chi_G(v')) \cup \mathcal{S}_{u'} = \emptyset$.

We finally prove that $\chi_G^{-1}(\chi_G(u)) \cup \mathcal{S}_{u'} = \{u'\}$. If not, pick $u'' \in \chi_G^{-1}(\chi_G(u)) \cup \mathcal{S}_{u'}$ and $u'' \neq u'$. By Lemma C.28, the *shortest* path between $u'$ and $u''$ goes through some node $v''$ with color $\chi_G(v)$. Clearly, $v'' \in \mathcal{S}_u$, which contradicts the above paragraph and concludes the proof. $\square$

We have already fully characterized the properties of cut edges $\{u', v'\}$ with color $\{\chi_G(u), \chi_G(v)\}$. Now we switch our focus to the graph $H$. We first prove a general result that holds for arbitrary $H$.

**Lemma C.33.** *Let $\{w_1, w_2\} \in \mathcal{E}_H$ and $P$ is a path with the minimum length from $w_1$ to $w_2$ without going through edge $\{w_1, w_2\}$. In other words, linking path $P$ with the edge $\{w_1, w_2\}$ forms a simple cycle $Q$. Then for any two nodes $x_1, x_2$ in $Q$, $\mathrm{dis}_H(x_1, x_2) = \mathrm{dis}_Q(x_1, x_2)$.*

*Proof.* Split the cycle $Q$ into two paths $Q_1$ and $Q_2$ with endpoints $\{x_1, x_2\}$ where $Q_1$ contains the edge $\{w_1, w_2\}$ and $Q_2$ does not contain $\{w_1, w_2\}$. Assume the above lemma does not hold and $\mathrm{dis}_H(w, x) < \mathrm{dis}_Q(w, x)$. It means that there exists a path $R$ in $H$ from $x_1$ to $x_2$ for which $|R| < \min(|Q_1|, |Q_2|)$. Note that the edge $\{u, v\}$ occurs at most once in $R$. Separately consider two cases:

- $\{w_1, w_2\}$ occurs in $R$. Then linking $R$ with $Q_2$ forms a cycle that contains $\{w_1, w_2\}$ exactly once;

- $\{w_1, w_2\}$ does not occur in $R$. Then linking $R$ with $Q_1$ forms a cycle that contains $\{w_1, w_2\}$ exactly once.

In both cases, the cycle has a length less than $|Q|$. This contradicts the condition that $P$ is a path with minimum length from $w_1$ to $w_2$ without passing edge $\{w_1, w_2\}$. $\square$

We can similarly consider the color graph $H^C = (\mathcal{C}, \mathcal{E}_{H^C})$ defined in Definition C.26. Note that we have assumed that the graph representations of $G$ and $H$ are the same, i.e. $\{\{\chi_G(w) : w \in \mathcal{V}\}\} = \{\{\chi_H(w) : w \in \mathcal{V}\}\}$. It follows that $H^C$ is isomorphic to $G^C$ and the identity vertex mapping is an isomorphism, i.e., $\{\{c_1, c_2\}\} \in \mathcal{E}_{G^C} \iff \{\{c_1, c_2\}\} \in \mathcal{E}_{H^C}$. Therefore, $\{\{\chi_G(u), \chi_G(v)\}\}$ is a cut edge of $H^C$ (Lemma C.29) that splits the vertices $\mathcal{C}$ into two classes $\mathcal{C}_u, \mathcal{C}_v$. Since the vertex labels of $H$ are not important, we can without abuse of notation let $u, v$ be two nodes such that

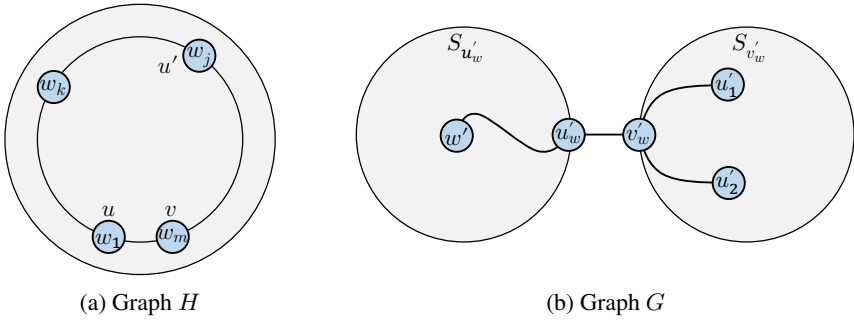

| (a) Graph $H$ | (b) Graph $G$ |

Figure 9: Illustrations to help understand the proof of the main result.

$\{u, v\} \in \mathcal{E}_H$, $\chi_H(u) = \chi_G(u)$, $\chi_H(v) = \chi_G(v)$, and $\chi_H(u) \in \mathcal{C}_u$, $\chi_H(v) \in \mathcal{C}_v$. We similarly define $\chi_H^{-1}(c) = \{w \in \mathcal{V} : \chi_H(w) = c\}$. Define a mapping $h : \mathcal{C} \to \{\chi_H(u), \chi_H(v)\}$ where

$$h(c) = \begin{cases} \chi_H(u) & \text{if } \mathrm{dis}_{H^C}(c, \chi_H(u)) < \mathrm{dis}_{H^C}(c, \chi_H(v)), \\ \chi_H(v) & \text{if } \mathrm{dis}_{H^C}(c, \chi_H(u)) > \mathrm{dis}_{H^C}(c, \chi_H(v)). \end{cases} \tag{11}$$

Note that it never happens that $\mathrm{dis}_{H^C}(c, \chi_H(u)) = \mathrm{dis}_{H^C}(c, \chi_H(v))$ because $\{\!\!\{\chi_H(u), \chi_H(v)\}\!\!\}$ is a cut edge of $H^C$.

Assume $\{u, v\}$ is not a cut edge in $H$. Then there exists a path $(w_1, \cdots, w_m)$ in $H$ with $w_1 = u$ and $w_m = v$ without going through $\{u, v\}$. We pick such a path with the minimum length, then the path is simple. Since $h(\chi_H(u)) \in \mathcal{C}_u$ and $h(\chi_H(v)) \in \mathcal{C}_v$, there is a minimum index $j \in [m-1]$ such that $h(\chi_H(w_j)) \in \mathcal{C}_u$ and $h(\chi_H(w_{j+1})) \in \mathcal{C}_v$. By definition of $\mathcal{C}_u, \mathcal{C}_v$ and the cut edge $\{\!\!\{\chi_H(u), \chi_H(v)\}\!\!\}$, it follows that $\chi_H(w_j) = \chi_H(u)$ and $\chi_H(w_{j+1}) = \chi_H(v)$. Denote $u' := w_j$. Note that $j \neq 1$ and $j \neq 2$, otherwise $u$ either connects to two nodes $w_2$ and $w_m$ with color $\chi_H(w_2) = \chi_H(w_m) = \chi_H(v)$, or connects to the node $u'$ with color $\chi_H(u') = \chi_H(u)$, contradicting $\chi_H(u) = \chi_G(u)$. Pick $k = \lceil j/2 \rceil$. By Lemma C.33, $(w_1, \cdots, w_k)$ is the shortest path between $u$ and $w_k$, and $(w_k, \cdots, w_j)$ is the shortest path between $w_k$ and $u'$. We give an illustration of the structure of $H$ in Figure 9(a) based on this paragraph.

Since the graph representations of $G$ and $H$ are the same under SPD-WL, there exists a node $w'$ with color $\chi_G(w') = \chi_H(w_k)$ and two different nodes $u_1', u_2'$ with color $\chi_G(u_1') = \chi_G(u_2') = \chi_G(u)$, such that $\mathrm{dis}_G(w', u_1') = \mathrm{dis}_H(w_k, u_1)$ and $\mathrm{dis}_G(w', u_2') = \mathrm{dis}_H(w_k, u_2)$. In particular, $|\mathrm{dis}_G(w', u_1') - \mathrm{dis}_G(w', u_2')| \leq 1$. Note that by the definition of indices $j$ and $k$, in the color graph $H^C$ there is a path from $\chi_H(w_k)$ to $\chi_H(u)$ without going through nodes in the set $\chi_H^{-1}(\chi_H(v))$, so $\chi_H(w_k) \in \mathcal{C}_u$, namely $\chi_G(w') \in \mathcal{C}_u$. By Corollary C.31, there is a cut edge $\{u_w', v_w'\}$ that partitions $G$ into two vertex sets $\mathcal{S}_{u_w'}, \mathcal{S}_{v_w'}$, with $w', u_w' \in \mathcal{S}_{u_w'}$, $v_w' \in \mathcal{S}_{v_w'}$. Note that $u_w' \neq u_1'$ and $u_w' \neq u_2'$ (otherwise by Corollary C.31 any path from $w'$ to a node $u' \neq u_w'$ with color $\chi_G(u') = \chi_G(u)$ must first go through $u_w'$ and then go through $v_w'$, implying that $|\mathrm{dis}_G(w', u_1') - \mathrm{dis}_G(w', u_2')| \geq 2$ and yielding a contradiction). Therefore, $\mathrm{dis}_G(w', u_1') > \mathrm{dis}_G(w', u_w')$ and $\mathrm{dis}_G(w', u_2') > \mathrm{dis}_G(w', u_w')$. We give an illustration of the structure of $G$ in Figure 9(b) based on this paragraph.

Pick any $v_w \in \chi_H^{-1}(\chi_H(v))$ satisfying $\mathrm{dis}_H(v_w, w_k) = \mathrm{dis}_G(v_w', w')$. Denote the operation $\mathrm{dropmin}(\mathcal{S}) := \mathcal{S} \backslash \{\!\!\{\min \mathcal{S}\}\!\!\}$ that takes a multiset $\mathcal{S}$ and removes one of the minimum elements in $\mathcal{S}$. We have

$$\mathrm{dropmin}(\{\!\!\{\mathrm{dis}_G(w', u_G) : u_G \in \chi_G^{-1}(\chi_G(u)))$$
$$= \mathrm{dropmin}(\{\!\!\{\mathrm{dis}_G(w', v_w') + \mathrm{dis}_G(v_w', u_G) : u_G \in \chi_G^{-1}(\chi_G(u))\}\!\!\}) \qquad \text{(by Corollary C.31)}$$
$$= \mathrm{dropmin}(\{\!\!\{\mathrm{dis}_H(w_k, v_w) + \mathrm{dis}_H(v_w, u_H) : u_H \in \chi_H^{-1}(\chi_H(u))\}\!\!\})$$

and also

$$\mathrm{dropmin}(\{\!\!\{\mathrm{dis}_G(w', u_G) : u_G \in \chi_G^{-1}(\chi_G(u))) = \mathrm{dropmin}(\{\!\!\{\mathrm{dis}_H(w_k, u_H) : u_H \in \chi_H^{-1}(\chi_H(u))\}\!\!\})$$

due to the same color $\chi_G(w') = \chi_H(w_k)$. Combining the above two equations and noting that $\mathrm{dis}_H(w_k, v_w) + \mathrm{dis}_H(v_w, u_H) \geq \mathrm{dis}_H(w_k, u_H)$, we obtain the following result: for any

$u_H \in \chi_H^{-1}(\chi_H(u))$ for which $\mathrm{dis}_H(w_k, v_w) + \mathrm{dis}_H(v_w, u_H) > \mathrm{dis}_G(w', u'_w)$, $\mathrm{dis}_H(w_k, v_w) + \mathrm{dis}_H(v_w, u_H) = \mathrm{dis}_H(w_k, u_H)$. In particular,

$$\mathrm{dis}_H(w_k, w_1) = \mathrm{dis}_H(w_k, v_w) + \mathrm{dis}_H(v_w, w_1),$$
$$\mathrm{dis}_H(w_k, w_j) = \mathrm{dis}_H(w_k, v_w) + \mathrm{dis}_H(v_w, w_j).$$

Therefore,

$$\begin{aligned} \mathrm{dis}_H(w_1, w_j) &= \mathrm{dis}_H(w_1, w_k) + \mathrm{dis}_H(w_k, w_j) \\ &= 2\mathrm{dis}_H(w_k, v_w) + \mathrm{dis}_H(v_w, w_1) + \mathrm{dis}_H(v_w, w_j) \\ &\geq 2\mathrm{dis}_H(w_k, v_w) + \mathrm{dis}_H(w_1, w_j), \end{aligned}$$

implying $w_k = v_w$. However, $\chi_H(w_k) \in \mathcal{C}_u$ while $\chi_H(v_w) \in \mathcal{C}_v$, yielding a contradiction.

### C.4.2  THE CASE OF $\chi_G(u) = \chi_G(v)$ FOR CONNECTED GRAPHS

We first define several notations. Define the mapping $f_G : \mathcal{V} \to \{u, v\} \times \mathcal{C}$ as follows: $f_G(w) = (h_G(w), \chi_G(w))$ where

$$h_G(w) = \begin{cases} u & \text{if } \mathrm{dis}_G(w, v) = \mathrm{dis}_G(w, u) + 1, \\ v & \text{if } \mathrm{dis}_G(w, u) = \mathrm{dis}_G(w, v) + 1. \end{cases} \tag{12}$$

It is easy to see that $h_G$ is well-defined for all $w \in \mathcal{V}$ because $\{u, v\}$ is a cut edge of $G$. We further define the following auxiliary graph:

**Definition C.34.** (Auxiliary graph) Define the auxiliary graph $G^{\mathrm{A}} = (\mathcal{V}_{G^{\mathrm{A}}}, \mathcal{E}_{G^{\mathrm{A}}})$ where $\mathcal{V}_{G^{\mathrm{A}}} := \{u, v\} \times \mathcal{C}$ and $\mathcal{E}_{G^{\mathrm{A}}} := \{\{f_G(w_1), f_G(w_2)\} : \{w_1, w_2\} \in \mathcal{E}_G\}$. Note that $G^{\mathrm{A}}$ can have self loops, so each edge is denoted as a multiset with two elements.

It is straightforward to see that there is only one edge in $G^{\mathrm{A}}$ with the form $\{\{(u, c_1), (v, c_2)\}\} \in \mathcal{E}_{G^{\mathrm{A}}}$ for some $c_1, c_2 \in \mathcal{C}$ since $\{u, v\}$ is a cut edge of $G$. Therefore, the only edge is $\{\{(u, \chi_G(u)), (v, \chi_G(v))\}\}$ and is a cut edge in $G^{\mathrm{A}}$.

We also define $f_G^{-1}$ as the inverse mapping of $f_G$, i.e. $f_G^{-1}(z, c) = \{w \in \mathcal{V} : f_G(w) = (z, c)\}$. We first prove that $f_G^{-1}$ is well-defined on the domain $\mathcal{V}_{G^{\mathrm{A}}}$.

**Lemma C.35.** $f_G$ is a surjection.

*Proof.* Suppose that $f_G$ is not a surjection. Then there exists a color $c \in \mathcal{C}$ such that either $f_G^{-1}(u, c)$ or $f_G^{-1}(v, c)$ is an empty set. Without loss of generality, assume $f_G^{-1}(v, c) = \emptyset$, then $f_G^{-1}(u, c) \neq \emptyset$. Pick any $w \in f_G^{-1}(u, c)$. Obviously, $w \neq u$ (otherwise $f_G^{-1}(v, \chi_G(v)) = \emptyset$, a contradiction). Then we claim that for any $x \in \mathcal{N}_G(w)$, $f_G^{-1}(v, \chi_G(x))$ is empty. Note that $x \in f_G^{-1}(u, \chi_G(x))$. If the claim does not hold, take $x' \in f_G^{-1}(v, \chi_G(x))$. Since $x$ connects to a node with color $c$ and $\chi_G(x) = \chi_G(x')$, $x'$ must also connect to a node with color $c$. Denote the node that connects to $x'$ with color $c$ as $w'$. Then $w' \in f_G^{-1}(v, c)$, yielding a contradiction.

By induction, for any $x$ such that there exists a path from $x$ to $w$ without going through the edge $\{u, v\}$, we have $f_G^{-1}(v, \chi_G(x)) = \emptyset$. This finally implies $f_G^{-1}(v, \chi_G(v)) = \emptyset$, leading to a contradiction. Therefore, $f$ is a surjection. $\square$

**Lemma C.36.** $|f_G^{-1}(u, \chi_G(u))| = |f_G^{-1}(v, \chi_G(v))| = 1$.

*Proof.* Pick $u' = \arg\max_{u' \in f_G^{-1}(u, \chi(u))} \mathrm{dis}_G(u, u')$ and similarly pick $v'$. It follows that any path between $u'$ and $v'$ goes through edge $\{u, v\}$. Therefore, $\mathrm{dis}_G(u', v') = \mathrm{dis}_G(u, u') + \mathrm{dis}_G(v, v') + 1$. Since all nodes $u, u', v, v'$ have the same color under SPD-WL, there exists a node $w \in \chi_G^{-1}(\chi_G(u))$ satisfying $\mathrm{dis}_G(u, w) = \mathrm{dis}_G(u', v')$ and thus $\mathrm{dis}_G(u, w) > \mathrm{dis}_G(u, u')$. By definition of the node $u'$, $f_G(w) \neq (u, \chi(u))$ and thus $f_G(w) = (v, \chi(u))$. Therefore, $\mathrm{dis}_G(u, w) = \mathrm{dis}_G(v, w) + 1$, which implies that

$$\mathrm{dis}_G(v, w) = \mathrm{dis}_G(v, v') + \mathrm{dis}_G(u, u').$$

Since $\mathrm{dis}_G(v, w) \leq \mathrm{dis}_G(v, v')$, we have $\mathrm{dis}_G(v, w) = \mathrm{dis}_G(v, v')$ and $u = u'$. A similar argument yields $v = v'$, finishing the proof. $\square$

We can now prove some useful properties of the auxiliary graph $G^A$ based on Lemmas C.35 and C.36.

**Corollary C.37.** *For any $c_1, c_2 \in \mathcal{C}$, $\{\{(u, c_1), (u, c_2)\}\} \in \mathcal{E}_{G^A}$ if and only if $\{\{(v, c_1), (v, c_2)\}\} \in \mathcal{E}_{G^A}$.*

*Proof.* By definition of $\mathcal{E}_G^A$, if $\{\{(u, c_1), (u, c_2)\}\} \in \mathcal{E}_{G^A}$, then there exists two vertices $w_1 \in f_G^{-1}(u, c_1)$ and $w_2 \in f_G^{-1}(u, c_2)$ such that $\{w_1, w_2\} \in \mathcal{E}_G$. By Lemma C.36, either $\chi_G(w_1) \neq \chi_G(u)$ or $\chi_G(w_2) \neq \chi_G(u)$. Without loss of generality, assume $c_1 \neq \chi_G(u)$. By Lemma C.35, there exists $x_1 \in f_G^{-1}(v, c_1)$. Since $\chi_G(x_1) = \chi_G(w_1)$, $x_1$ must also connect to a node $x_2$ with $\chi_G(x_2) = c_2$. The edge $\{x_1, x_2\} \neq \{u, v\}$ because $\chi_G(x_1) = c_1 \neq \chi_G(u)$. Therefore, $f(x_2) = (v, c_2)$, namely $\{\{(v, c_1), (v, c_2)\}\} \in \mathcal{E}_G^A$. $\square$

The following lemma establishes the distance relationship between graphs $G$ and $G^A$.

**Lemma C.38.** *The following holds:*

- *For any $w, w' \in \mathcal{V}$, $\mathrm{dis}_G(w, w') \geq \mathrm{dis}_{G^A}(f(w), f(w'))$.*

- *For any $\xi, \xi' \in \mathcal{V}^A$ and any node $w \in f_G^{-1}(\xi)$, there exists a node $w' \in f_G^{-1}(\xi')$ such that $\mathrm{dis}_G(w, w') = \mathrm{dis}_{G^A}(\xi, \xi')$.*

*Proof.* The first bullet is trivial since for all $\{w, w'\} \in \mathcal{E}_G$, $\{\{f(w), f(w')\}\} \in \mathcal{E}_{G^A}$ by Definition C.34. We prove the second bullet in the following. Note that $G^A$ can have self-loops, but for any $\xi, \xi' \in \mathcal{V}^A$, the shortest path between $\xi$ and $\xi'$ will not go through self-loops. We only need to prove that for all $\{\{\xi, \xi'\}\} \in \mathcal{E}^A$, $\xi \neq \xi'$ and all $w \in f_G^{-1}(\xi)$, there exists $w' \in f_G^{-1}(\xi')$ such that $\{w, w'\} \in \mathcal{E}_G$. This will imply that $\mathrm{dis}_G(w, w') \leq \mathrm{dis}_{G^A}(\xi, \xi')$ and completes the proof by combining the first bullet in Lemma C.38.

The case of $\{\{\xi, \xi'\}\} = \{\{(u, \chi_G(u)), (v, \chi_G(v))\}\}$ is trivial. Now assume that $\{\{\xi, \xi'\}\} \neq \{\{(u, \chi_G(u)), (v, \chi_G(v))\}\}$. By Definition C.34, there exists $x \in f_G^{-1}(\xi)$ and $x' \in f_G^{-1}(\xi')$, such that $\{x, x'\} \in \mathcal{E}_G$. Note that $h_G(x) = h_G(x')$ because $\{x, x'\} \neq \{u, v\}$. Since $\chi_G(x) = \chi_G(w)$, there exists $w' \in \chi_G^{-1}(\chi_G(x'))$ such that $\{w, w'\} \in \mathcal{E}_G$. It must hold that $h_G(w) = h_G(w')$ (otherwise $\{w, w'\} = \{u, v\}$ and thus $\{\{\xi, \xi'\}\} = \{\{(u, \chi_G(u)), (v, \chi_G(v))\}\}$). Therefore, $h_G(w') = h_G(w) = h_G(x) = h_G(x')$ and thus $f_G(w') = f_G(x')$, namely $w' \in f_G^{-1}(\xi')$. $\square$

Lemma C.38 leads to the following corollary:

**Corollary C.39.** *The following holds:*

- *For any $w, w' \in \mathcal{V}$ satisfying $\chi_G(w) = \chi_G(w')$ and $h_G(w) = h_G(w')$ (i.e. $f_G(w) = f_G(w')$), $\mathrm{dis}_G(u, w) = \mathrm{dis}_G(u, w')$ and $\mathrm{dis}_G(v, w) = \mathrm{dis}_G(v, w')$;*

- *For any $w, w' \in \mathcal{V}$ satisfying $\chi_G(w) = \chi_G(w')$ and $h_G(w) \neq h_G(w')$, $\mathrm{dis}_G(u, w) = \mathrm{dis}_G(v, w')$ and $\mathrm{dis}_G(v, w) = \mathrm{dis}_G(u, w')$.*

*Proof.* Proof of the first bullet: by Lemma C.38, there exists two nodes $u_1, u_2 \in f_G^{-1}(f_G(u))$ such that $\mathrm{dis}_G(u_1, w) = \mathrm{dis}_{G^A}(f_G(u), f_G(w))$ and $\mathrm{dis}_G(u_2, w') = \mathrm{dis}_{G^A}(f_G(u), f_G(w'))$. Therefore, $\mathrm{dis}_G(u_1, w) = \mathrm{dis}_G(u_2, w')$. However, by Lemma C.36 and the condition $h_G(w) = h_G(w')$, it must be $u_1 = u_2 = u$, namely $\mathrm{dis}_G(u, w) = \mathrm{dis}_G(u, w')$. The proof of $\mathrm{dis}_G(v, w) = \mathrm{dis}_G(v', w')$ is similar.

Proof of the second bullet: Let $\chi_G(w) = \chi_G(w') = c$. Without loss of generality, assume $f_G(w) = (u, c)$ and $f(w') = (v, c)$. By Lemma C.38, it suffices to prove that $\mathrm{dis}_{G^A}((u, \chi_G(u)), (u, c)) = \mathrm{dis}_{G^A}((v, \chi_G(v)), (v, c))$. By the definition of $G^A$ and its cut edge $\{\{(u, \chi_G(u), (v, \chi_G(v))\}\}$, the shortest path between $(u, \chi_G(u))$ and $(u, c)$ must only go through nodes in the set $\{(u, c_1) : c_1 \in \mathcal{C}\}$, and similarly the shortest path between $(v, \chi_G(v))$ and $(v, c)$ must only go through nodes in $\{(v, c_2) : c_2 \in \mathcal{C}\}$. Finally, Corollary C.37 says that for $c_1, c_2 \in \mathcal{C}$, $\{\{(u, c_1), (u, c_2)\}\} \in G^A$ if and only if $\{\{(v, c_1), (v, c_2)\}\} \in G^A$. We thus conclude that $\mathrm{dis}_{G^A}((u, \chi_G(u)), (u, c)) = \mathrm{dis}_{G^A}((v, \chi_G(v)), (v, c))$ and $\mathrm{dis}_G(u, w) = \mathrm{dis}_G(v, w')$. $\square$

Finally, we can prove the following important corollary:

**Corollary C.40.** *For any $c \in \mathcal{C}$, $|f_G^{-1}(u,c)| = |f_G^{-1}(v,c)|$.*

*Proof.* Pick any $w \in f_G^{-1}(u,c)$ and $x \in f_G^{-1}(v,c)$. By Corollary C.39, we have

$$\mathrm{dis}_G(w,u) = \mathrm{dis}_G(x,v) := d,$$
$$\mathrm{dis}_G(w,v) = \mathrm{dis}_G(x,u) = d+1.$$

The multiset $\{\!\{\mathrm{dis}_G(u,w') : \chi_G(w') = c\}\!\}$ contains $|f_G^{-1}(u,c)|$ elements of value $d$ and $|f_G^{-1}(v,c)|$ elements of value $d+1$. The multiset $\{\!\{\mathrm{dis}_G(v,w') : \chi_G(w') = c\}\!\}$ has $|f_G^{-1}(v,c)|$ elements of value $d$ and $|f_G^{-1}(u,c)|$ elements of value $d+1$. Since $u$ and $v$ has the same color under SPD-WL, the two multiset must be equivalent. Therefore, $|f_G^{-1}(u,c)| = |f_G^{-1}(v,c)|$. $\qquad\square$

Next, we switch our focus to the graph $H$. Since we have assumed that the graph representations of $G$ and $H$ are the same, i.e. $\{\!\{\chi_G(w) : w \in \mathcal{V}\}\!\} = \{\!\{\chi_H(w) : w \in \mathcal{V}\}\!\}$, the size of the set $\{w \in \mathcal{V} : \chi_H(w) = \chi_G(u)\}$ must be 2. We may denote the elements as $u$ and $v$ without abuse of notation and thus $\{u,v\} \in \mathcal{E}_H$. Also for any $w \in \mathcal{V}$, we have $\mathrm{dis}_H(w,u) \neq \mathrm{dis}_H(w,v)$. Therefore, we can similarly define the mapping $f_H : \mathcal{V} \to \{u,v\} \times \mathcal{V}$ and the mapping $h_H : \mathcal{V} \to \{u,v\}$ as in (12). The auxiliary graph $H^{\mathrm{A}}$ is defined analogous to Definition C.34.

**Lemma C.41.** *For any $c \in \mathcal{C}$, $|f_H^{-1}(u,c)| = |f_H^{-1}(v,c)| = |f_G^{-1}(u,c)| = |f_G^{-1}(v,c)|$.*

*Proof.* If $|f_H^{-1}(u,c)| \neq |f_H^{-1}(v,c)|$, we have $\{\!\{\mathrm{dis}_H(u,w) : \chi_H(w) = c\}\!\} \neq \{\!\{\mathrm{dis}_H(v,w) : \chi_H(w) = c\}\!\}$, implying that $u$ and $v$ cannot have the same color under SPD-WL. This already concludes the proof by using Corollary C.40 as

$$|f_H^{-1}(u,c)| + |f_H^{-1}(v,c)| = |f_G^{-1}(u,c)| + |f_G^{-1}(v,c)|. \qquad\square$$

We finally present a technical lemma which will be used in the subsequent proof.

**Lemma C.42.** *Given node $w \in \mathcal{V}$ and color $c \in \mathcal{C}$, define multisets*

$$\mathcal{D}_{G,=}(w,c) := \{\!\{\mathrm{dis}_G(w,x) : x \in \chi_G^{-1}(c), h_G(x) = h_G(w)\}\!\},$$
$$\mathcal{D}_{G,\neq}(w,c) := \{\!\{\mathrm{dis}_G(w,x) : x \in \chi_G^{-1}(c), h_G(x) \neq h_G(w)\}\!\}.$$

*For any two nodes $w, w' \in \mathcal{V}$ in graphs $G$ and $H$ satisfying $\chi_G(w) = \chi_H(w')$, pick any $d \in \mathcal{D}_{G,\neq}(w,c)$ and $d' \in \mathcal{D}_{H,=}(w',c)$. Then $d' < d$.*

*Proof.* Without loss of generality, assume $h_G(w) = h_H(w') = u$ and let $f_G(w) = f_H(w') = (u,c_w)$. Pick $x \in f_G^{-1}(v,c)$ and $x' \in f_H^{-1}(u,c)$, then $\mathrm{dis}_H(x',u) = \min(\mathrm{dis}_G(x,u), \mathrm{dis}_G(x,v))$ and $\mathrm{dis}_H(w',u) = \min(\mathrm{dis}_G(w,u), \mathrm{dis}_G(w,v))$. Thus

$$\begin{aligned}
\mathrm{dis}_H(w',x') &\leq \mathrm{dis}_H(w',u) + \mathrm{dis}_H(u,x') \\
&= \min(\mathrm{dis}_G(w,u), \mathrm{dis}_G(w,v)) + \min(\mathrm{dis}_G(x,u), \mathrm{dis}_G(x,v)) \\
&< \min(\mathrm{dis}_G(w,u) + \mathrm{dis}_G(x,v), \mathrm{dis}_G(w,v) + \mathrm{dis}_G(x,u)) + 1 \\
&= \mathrm{dis}_G(w,x),
\end{aligned}$$

which concludes the proof. $\qquad\square$

In the following, we will prove that $\{u,v\}$ is a cut edge in graph $H$. Consider an edge $\{\!\{(u,c_1),(v,c_2)\}\!\} \in \mathcal{E}_{H^{\mathrm{A}}}$ (such an edge exists because $\{\!\{(u,\chi_H(u)),(v,\chi_H(v))\}\!\} \in \mathcal{E}_H^{\mathrm{A}}$). We will prove that this is the only case, i.e. it must be $c_1 = \chi_H(u) = \chi_H(v) = c_2$.

By Definition C.34, $\{\!\{(u,c_1),(v,c_2)\}\!\} \in \mathcal{E}_{H^{\mathrm{A}}}$ implies that there exists two nodes $x' \in f_H^{-1}(u,c_1)$ and $w' \in f_H^{-1}(v,c_2)$, such that $\{w',x'\} \in \mathcal{E}_H$. Pick $w \in \chi_G^{-1}(c_2)$. By Lemma C.42, $\mathcal{D}_{H,=}(w',c_1) \cap \mathcal{D}_{G,\neq}(w,c_1) = \emptyset$. Since $w'$ and $w$ have the same color under SPD-WL,

$$\mathcal{D}_{H,=}(w',c_1) \cup \mathcal{D}_{H,\neq}(w',c_1) = \mathcal{D}_{G,=}(w,c_1) \cup \mathcal{D}_{G,\neq}(w,c_1).$$

By Lemma C.41, $|\mathcal{D}_{H,=}(w',c_1)| = |\mathcal{D}_{H,\neq}(w',c_1)| = |\mathcal{D}_{G,=}(w,c_1)| = |\mathcal{D}_{G,\neq}(w,c_1)|$. Therefore, $\mathcal{D}_{G,\neq}(w,c_1) = \mathcal{D}_{H,\neq}(w',c_1)$. We claim that all elements in the set $\mathcal{D}_{G,\neq}(w,c_1)$ are the same. This is because for any $x \in \chi_G^{-1}(c_1), h_G(x) \neq h_G(w)$, we have

$$\mathrm{dis}_G(w,x) = \mathrm{dis}_G(w,h(w)) + 1 + \mathrm{dis}_G(h(x),x),$$

and by Corollary C.39 $\mathrm{dis}_G(w, h(w))$ (or $\mathrm{dis}_G(h(x), x)$) has an equal value for different $x$. Since $\{w', x'\} \in \mathcal{E}_H$, we have $1 \in \mathcal{D}_{H,\neq}(w', c_1)$ and thus all elements in $\mathcal{D}_{G,\neq}(w, c_1)$ equals 1. Therefore, $c_1 = \chi_G(u)$. Analogously, $c_2 = \chi_G(u)$. Therefore, $c_1 = \chi_H(u) = \chi_H(v) = c_2$.

Let $\mathcal{S}_u = \{w \in \mathcal{V} : h_H(w) = u\}$ and $\mathcal{S}_v = \{w \in \mathcal{V} : h_H(w) = v\}$. Then the above argument implies that if $w \in \mathcal{S}_u$, $x \in \mathcal{S}_v$ and $\{w, x\} \in \mathcal{E}_G$, then $\{w, x\} = \{u, v\}$. Therefore $\{u, v\}$ is a cut edge of graph $H$.

### C.4.3 THE GENERAL CASE

The above proof assumes that the graphs $G$ and $H$ are both connected, and their graph representations are euqal, i.e. $\{\!\{\chi_G(w) : w \in \mathcal{V}\}\!\} = \{\!\{\chi_H(w) : w \in \mathcal{V}\}\!\}$. In the subsequent proof we remove these assumptions and prove the general setting.

**Lemma C.43.** *Either of the following two properties holds:*

- $\{\!\{\chi_G(w) : w \in \mathcal{V}\}\!\} = \{\!\{\chi_H(w) : w \in \mathcal{V}\}\!\}$;

- $\{\!\{\chi_G(w) : w \in \mathcal{V}\}\!\} \cap \{\!\{\chi_H(w) : w \in \mathcal{V}\}\!\} = \emptyset$.

*Proof.* Consider the GD-WL procedure defined in Algorithm 4 with arbitrary distance function $d_G$. Suppose at iteration $t \geq T$, $\{\!\{\chi_G^t(w) : w \in \mathcal{V}\}\!\} \neq \{\!\{\chi_H^t(w) : w \in \mathcal{V}\}\!\}$. Then at iteration $t+1$, we have for each $v \in \mathcal{V}$,

$$\chi_G^{t+1}(v) = \mathrm{hash}\left(\{\!\{\mathrm{hash}(d_G(v, u), \chi_G^t(u)) : u \in \mathcal{V}\}\!\}\right).$$

Therefore, $\chi_G^{t+1}(v) \neq \chi_G^{t+1}(u)$ for all $u, v \in \mathcal{V}$, namely

$$\{\!\{\chi_G^{t+1}(w) : w \in \mathcal{V}\}\!\} \cap \{\!\{\chi_H^{t+1}(w) : w \in \mathcal{V}\}\!\} = \emptyset.$$

Finally, by the injective property of the hash function, for any $t \geq T+1$, the above equation always holds. Therefore, the stable color mappings $\chi_G$ and $\chi_H$ satisfy Lemma C.43. $\qquad\square$

The above lemma implies that if there exists edges $\{w_1, w_2\} \in \mathcal{E}_G$, $\{x_1, x_2\} \in \mathcal{E}_H$ satisfying $\{\!\{\chi_G(w_1), \chi_G(w_2)\}\!\} = \{\!\{\chi_H(x_1), \chi_H(x_2)\}\!\}$, then $\{\!\{\chi_G(w) : w \in \mathcal{V}\}\!\} = \{\!\{\chi_H(w) : w \in \mathcal{V}\}\!\}$. Also, SPD-WL ensures that both graphs are either connected or disconnected. If they are both connected, the previous proof (Appendices C.4.1 and C.4.2) ensures that $\{w_1, w_2\}$ is a cut edge of $G$ if and only if $\{x_1, x_2\}$ is a cut edge of $H$. For the disconnected case, let $\mathcal{S}_G \subset \mathcal{V}$ be the largest connected component containing nodes $w_1, w_2$, and similarly denote $\mathcal{S}_H \subset \mathcal{V}$ as the largest connected component containing nodes $x_1, x_2$. Obviously, $|\mathcal{S}_G| = |\mathcal{S}_H|$ due to the facts that $\mathrm{dis}_G(w_1, y) = \infty \neq \mathrm{dis}_G(w_1, y')$ for all $y \notin \mathcal{S}_G, y \in \mathcal{S}_G$ and that the two edges $\{w_1, w_2\} \in \mathcal{E}_G$, $\{x_1, x_2\}$ have the same color under SPD-WL. Moreover, $\{\!\{\chi_G(w) : w \in \mathcal{S}_G\}\!\} = \{\!\{\chi_H(w) : w \in \mathcal{S}_H\}\!\}$. Now consider re-execute the SPD-WL algorithm on subgraphs $G[\mathcal{S}_G]$ and $H[\mathcal{S}_H]$ induced by the vertices in set $\mathcal{S}_G$ and $\mathcal{S}_H$, respectively. It follows that for any $u_G \in \mathcal{S}_G$ and $u_H \in \mathcal{S}_H$, $\chi_G(u_G) = \chi_H(u_H)$ implies that $\chi_{G[\mathcal{S}_G]}(u_G) = \chi_{H[\mathcal{S}_H]}(u_H)$. Therefore, $\{w_1, w_2\}$ is a cut edge of $G[\mathcal{S}_G]$ if and only if $\{x_1, x_2\}$ is a cut edge of $H[\mathcal{S}_H]$. By the dinifition of $\mathcal{S}_G$ and $\mathcal{S}_H$, $\{w_1, w_2\}$ is a cut edge of $G$ if and only if $\{x_1, x_2\}$ is a cut edge of $H$.

It remains to prove that $\{\!\{\chi_G(w) : w \in \mathcal{V}\}\!\} = \{\!\{\chi_H(w) : w \in \mathcal{V}\}\!\}$ implies $\mathrm{BCETree}(G) \simeq \mathrm{BCETree}(H)$. By definition of the block cut-edge tree, each cut edge of $G$ corresponds to a tree edge in $\mathrm{BCETree}(G)$ and each biconnected component of $G$ corresponds to a node of $\mathrm{BCETree}(G)$. We still only focus on the case of connected graphs $G, H$, and it is straightforward to extend the proof to the general (disconnected) case using a similar technique as the previous paragraph.

Given a fixed SPD-WL graph representation $\mathcal{R}$, consider any graphs $G = (\mathcal{V}, \mathcal{E}_G)$ satisfying $\{\!\{\chi_G(w) : w \in \mathcal{V}\}\!\} = \mathcal{R}$. Since we have proved that the SPD-WL node feature $\chi_G(v)$, $v \in \mathcal{V}$ precisely locates all the cut edges, the multiset

$$\mathcal{C}^{\mathrm{E}} := \{\!\{\{\chi_G(u), \chi_G(v)\} : \{u, v\} \in \mathcal{E}_G \text{ is a cut edge}\}\!\}$$

is fixed (fully determined by $\mathcal{R}$, not $G$). Denote $\mathcal{C}^{\mathrm{V}} := \bigcup_{\{c_1, c_2\} \in \mathcal{C}^{\mathrm{E}}} \{c_1, c_2\}$ as the set that contains the color of endpoints of all cut edges. For each cut edge $\{u, v\} \in \mathcal{E}_G$, denote $\mathcal{S}_{G,u}$ and $\mathcal{S}_{G,v}$ be

the vertex partition corresponding to the two connected components after removing the edge $\{u, v\}$, satisfying $u \in \mathcal{S}_{G,u}$, $v \in \mathcal{S}_{G,v}$, $\mathcal{S}_{G,u} \cap \mathcal{S}_{G,v} = \emptyset$, $\mathcal{S}_{G,u} \cup \mathcal{S}_{G,v} = \mathcal{V}$. It suffices to prove that given a cut edge $\{u, v\} \in \mathcal{E}_G$ with color $\{\chi_G(u), \chi_G(v)\}$, the multiset $\{\!\{\chi_G(w) : w \in \mathcal{S}_{G,u}, \chi_G(w) \in \mathcal{C}^V\}\!\}$ can be determined purely based on $\mathcal{R}$ and the edge color $\{\!\{\chi_G(u), \chi_G(v)\}\!\}$, rather than the specific graph $G$ or edge $\{u, v\}$. This basically concludes the proof, since the BCETree can be uniquely constructed as follows: if $\{\!\{\chi_G(w) : w \in \mathcal{S}_{G,u}, \chi_G(w) \in \mathcal{C}^V\}\!\} = \{\!\{\chi_G(u)\}\!\}$ (i.e. with only one element), then $\{\!\{\chi_G(u), \chi_G(v)\}\!\}$ is a leaf edge of the BCETree such that $\chi_G(u)$ connects to a biconnected component that is a leaf of the BCETree. After finding all the leaf edges, we can then find the BCETree edges that connect to leaf edges and determine which leaf edges they connect. The procedure can be recursively executed until the full BCETree is constructed. The whole procedure does not depend on the specific graph $G$ and only depends on $\mathcal{R}$.

We now show how to determine $\{\!\{\chi_G(w) : w \in \mathcal{S}_{G,u}, \chi_G(w) \in \mathcal{C}^V\}\!\}$ given a cut edge $\{u, v\} \in \mathcal{E}_G$ with color $\{\chi_G(u), \chi_G(v)\}$. Define the multiset

$$\mathcal{D}(c_1, c_2) := \{\!\{\mathrm{dis}_G(w, x) : x \in \chi_G^{-1}(c_2)\}\!\} \qquad (w \in \chi_G^{-1}(c_1) \text{ can be picked arbitrarily})$$

Note that $\mathcal{D}(c_1, c_2)$ is well-defined (does not depend on $w$) by definition of the SPD-WL color. For any $c_u, c_v \in \mathcal{C}^E$, pick arbitrary cut edge $\{u, v\}$ with color $\chi_G(u) = c_u, \chi_G(v) = c_v$. Define

$$\mathcal{T}(c_u, c_v) = \bigcup_{c \in \mathcal{C}^V} \{\!\{c\}\!\} \times |(\mathcal{D}(c_u, c) + 1) \cap \mathcal{D}(c_v, c)| \tag{13}$$

where $\{\!\{c\}\!\} \times m$ denotes a multiset with $m$ repeated elements $c$, and $\mathcal{D}(c_u, c) + 1 := \{\!\{d + 1 : d \in \mathcal{D}(c_u, c)\}\!\}$. Intuitively speaking, $\mathcal{T}(c_u, c_v)$ corresponds to the color of all nodes $w \in \mathcal{V}$ such that $\mathrm{dis}_G(u, w) + 1 = \mathrm{dis}_G(v, w)$ and $\chi_G(w) \in \mathcal{C}^V$. Therefore, $\mathcal{T}(c_u, c_v)$ is exactly the multiset $\{\!\{\chi_G(w) : w \in \mathcal{S}_{G,u}, \chi_G(w) \in \mathcal{C}^V\}\!\}$ and we have completed the proof.

## C.5 Proof of Theorem 4.2

**Theorem C.44.** *Let $G = (\mathcal{V}, \mathcal{E}_G)$ and $H = (\mathcal{V}, \mathcal{E}_H)$ be two graphs, and let $\chi_G$ and $\chi_H$ be the corresponding RD-WL color mapping. Then the following holds:*

- *For any two nodes $w \in \mathcal{V}$ in $G$ and $x \in \mathcal{V}$ in $H$, if $\chi_G(w) = \chi_H(x)$, then $w$ is a cut vertex of $G$ if and only if $x$ is a cut vertex of $H$.*

- *If the graph representations of $G$ and $H$ are the same under RD-WL, then their block cut-vertex trees (Definition 2.4) are isomorphic. Mathematically, $\{\!\{\chi_G(w) : w \in \mathcal{V}\}\!\} = \{\!\{\chi_H(w) : w \in \mathcal{V}\}\!\}$ implies that $\mathrm{BCVTree}(G) \simeq \mathrm{BCVTree}(H)$.*

*Proof Sketch.* First observe that Lemma C.43 holds for general distances and thus applies here. Therefore, if $\chi_G(w) = \chi_H(x)$, the graph representations will be the same, i.e. $\{\!\{\chi_G(w) : w \in \mathcal{V}\}\!\} = \{\!\{\chi_H(w) : w \in \mathcal{V}\}\!\}$. By a similar analysis as SPD-WL (Appendix C.4.3), we can only focus on the case that both graphs are connected. We prove the first bullet of Theorem 4.2 in Appendix C.5.1 and prove the second bullet in Appendix C.5.2, both assuming that $G$ and $H$ are connected and their graph representations are the same. $\qquad \square$

### C.5.1 Proof of the first part

We first present a key property of the Resistance Distance, which surprisingly relates to the cut vertices in a graph.

**Lemma C.45.** *Let $G = (\mathcal{V}, \mathcal{E})$ be a connected graph and $v \in \mathcal{V}$. Then $v$ is a cut vertex of $G$ if and only if there exists two nodes $u, w \in \mathcal{V}$, $u \neq v$, $w \neq v$, such that $\mathrm{dis}_G^R(u, v) + \mathrm{dis}_G^R(v, w) = \mathrm{dis}_G^R(u, w)$.*

*Proof.* We use the key finding that the Resistance Distance is equivalent to the Commute Time Distance multiplied by a constant (Chandra et al., 1996, see also Appendix E.2), i.e. $\mathrm{dis}_G^C(u, w) = 2|\mathcal{E}| \mathrm{dis}_G^R(u, w)$. Here, the Commute Time Distance is defined as $\mathrm{dis}_G^C(u, w) := h_G(u, w) + h_G(w, u)$ where $h_G(u, w)$ is the average hitting time from $u$ to $w$ in a random walk (Appendix E.2).

- If $v$ is not a cut vertex, given any nodes $u, w$, $u \neq v$, $w \neq v$, we can partition the set of all *hitting* paths $\mathcal{P}_{uw}$ from $u$ to $w$ (not necessarily simple) into two sets $\mathcal{P}^v_{uw}$ and $\overline{\mathcal{P}}^v_{uw}$ such that all paths $P \in \mathcal{P}^v_{uw}$ contain $v$ and no path $P \in \overline{\mathcal{P}}^v_{uw}$ contains $v$. Clearly, $\mathcal{P}^v_{uw} \neq \emptyset$ and $\overline{\mathcal{P}}^v_{uw} \neq \emptyset$. Given a path $P = (x_0, \cdots, x_m)$, define the probability function $q(P) := 1/\prod_{i=0}^{m-1} \deg_G(x_i)$. Then by definitions of the average hitting time $h$,

$$
\begin{aligned}
h_G(u, w) &= \sum_{P \in \mathcal{P}_{uw}} q(P) \cdot |P| = \sum_{P \in \mathcal{P}^v_{uw}} q(P) \cdot |P| + \sum_{P \in \overline{\mathcal{P}}^v_{uw}} q(P) \cdot |P| \\
&= \sum_{P_1 \in \overline{\mathcal{P}}^w_{uv}, P_2 \in \mathcal{P}_{vw}} q(P_1) q(P_2)(|P_1| + |P_2|) + \sum_{P \in \overline{\mathcal{P}}^v_{uw}} q(P) \cdot |P| \\
&= \sum_{P_1 \in \overline{\mathcal{P}}^w_{uv}} q(P_1)|P_1| \left( \sum_{P_2 \in \mathcal{P}_{vw}} q(P_2) \right) + \sum_{P_2 \in \mathcal{P}_{vw}} q(P_2)|P_2| \left( \sum_{P_1 \in \overline{\mathcal{P}}^w_{uv}} q(P_1) \right) \\
&\quad + \sum_{P \in \overline{\mathcal{P}}^v_{uw}} q(P)|P| \\
&\leq \sum_{P \in \overline{\mathcal{P}}^w_{uv}} q(P)|P| + \sum_{P \in \overline{\mathcal{P}}^v_{uw}} q(P)|P| + \sum_{P \in \mathcal{P}_{vw}} q(P)|P| \\
&< \sum_{P \in \overline{\mathcal{P}}^w_{uv}} q(P)|P| + \sum_{P \in \mathcal{P}^w_{uv}} q(P)|P| + \sum_{P \in \mathcal{P}_{vw}} q(P)|P| \\
&= h_G(u, v) + h_G(v, w).
\end{aligned}
$$

  We can similarly prove that $h_G(w, u) < h_G(w, v) + h_G(v, u)$.

- If $v$ is a cut vertex, then there exists two different nodes $u, w \in \mathcal{V}$, $u \neq v$, $w \neq v$, such that any path from $u$ to $w$ goes through $v$. A similar analysis yields the conclusion that $h_G(u, w) = h_G(u, v) + h_G(v, w)$ and $h_G(w, u) = h_G(w, v) + h_G(v, u)$.

This completes the proof of Lemma C.45. $\qquad \square$

In the subsequent proof, assume $u \in \mathcal{V}$ is a cut vertex of $G$, and let $\{\mathcal{S}_i\}_{i=1}^m$ ($m \geq 2$) be the partition of the vertex set $\mathcal{V} \backslash \{u\}$, representing each connected component after removing node $u$. We have the following lemma (which has a similar form as Lemma C.27):

**Lemma C.46.** *There is at most one set $\mathcal{S}_i$ satisfying $\mathcal{S}_i \cap \chi_G^{-1}(\chi_G(u)) \neq \emptyset$. In other words, if $\mathcal{S}_i \cap \chi_G^{-1}(\chi_G(u)) \neq \emptyset$ for some $i \in [m]$, then for any $j \in [m]$ and $j \neq i$, $\mathcal{S}_j \cap \chi_G^{-1}(\chi_G(u)) = \emptyset$.*

*Proof.* Let $u_i = \arg\max_{u' \in \chi_G^{-1}(\chi_G(u))} \mathrm{dis}_G^{\mathrm{R}}(u, u')$. If $u_i = u$, then $\mathcal{S}_i \cap \chi_G^{-1}(\chi_G(u)) = \emptyset$ for all $i \in [m]$ and thus Lemma C.46 clearly holds. Otherwise, $u_i \in \mathcal{S}_i$ for some $i$. We will prove that for any $j \neq i$, $\mathcal{S}_j \cap \chi_G^{-1}(\chi_G(u)) = \emptyset$.

If the above conclusion does not holds, then we can pick a set $\mathcal{S}_j$ and a vertex $u_j \in \mathcal{S}_j \cap \chi_G^{-1}(\chi_G(u))$. Since $u$ is a cut vertex and $\mathcal{S}_i$, $\mathcal{S}_j$ are different connected components, by Lemma C.45 we have $\mathrm{dis}_G^{\mathrm{R}}(u_i, u_j) = \mathrm{dis}_G^{\mathrm{R}}(u_i, u) + \mathrm{dis}_G^{\mathrm{R}}(u, u_j) > \mathrm{dis}_G^{\mathrm{R}}(u_i, u)$. This yields a contradiction because $\max_{u' \in \chi_G^{-1}(\chi_G(u))} \mathrm{dis}_G^{\mathrm{R}}(u, u') \neq \max_{u' \in \chi_G^{-1}(\chi_G(u_i))} \mathrm{dis}_G^{\mathrm{R}}(u_i, u')$, which means that $u$ and $u_i$ cannot have the same RD-WL color. $\qquad \square$

The next lemma presents a key result which is similar to Corollary C.30.

**Lemma C.47.** *For all $u' \in \chi_G^{-1}(\chi_G(u))$, $u'$ it is a cut vertex of $G$.*

*Proof.* If $|\chi_G^{-1}(\chi_G(u))| = 1$, then Lemma C.47 clearly holds. Otherwise, by Lemma C.46 there exists two sets $\mathcal{S}_i$ and $\mathcal{S}_j$ satisfying $\mathcal{S}_i \cap \chi_G^{-1}(\chi_G(u)) \neq \emptyset$, $\mathcal{S}_j \cap \chi_G^{-1}(\chi_G(u)) = \emptyset$. Since $\mathcal{S}_j \neq \emptyset$, we

can pick $w \in \mathcal{S}_j$ with color $\chi_G(w) \neq \chi_G(u)$. Pick $u' \in \mathcal{S}_i \cap \chi_G^{-1}(\chi_G(u))$. Since $\chi_G(u) = \chi_G(u')$, there exists a node $w' \in \chi_G^{-1}(\chi_G(w))$ such that $\mathrm{dis}_G^{\mathrm{R}}(u, w) = \mathrm{dis}_G^{\mathrm{R}}(u', w')$. Then we have

$$\{\!\{\mathrm{dis}_G^{\mathrm{R}}(w, u'') : u'' \in \chi_G^{-1}(\chi_G(u))\}\!\} = \{\!\{\mathrm{dis}_G^{\mathrm{R}}(w, u) + \mathrm{dis}_G^{\mathrm{R}}(u, u'') : u'' \in \chi_G^{-1}(\chi_G(u))\}\!\} \quad (14)$$
$$= \{\!\{\mathrm{dis}_G^{\mathrm{R}}(w', u') + \mathrm{dis}_G^{\mathrm{R}}(u', u'') : u'' \in \chi_G^{-1}(\chi_G(u))\}\!\} \,(15)$$

where (14) holds because $u$ is a cut vertex and all $u'' \neq u$ are in the set $\mathcal{S}_i$ but $w \in \mathcal{S}_j$ (Lemma C.46), and (15) holds because $\chi_G(u) = \chi_G(u')$. On the other hands,

$$\{\!\{\mathrm{dis}_G^{\mathrm{R}}(w, u'') : u'' \in \chi_G^{-1}(\chi_G(u))\}\!\} = \{\!\{\mathrm{dis}_G^{\mathrm{R}}(w', u'') : u'' \in \chi_G^{-1}(\chi_G(u))\}\!\}.$$

Therefore, $\mathrm{dis}_G^{\mathrm{R}}(w', u'') = \mathrm{dis}_G^{\mathrm{R}}(w', u') + \mathrm{dis}_G^{\mathrm{R}}(u', u'')$ for all $u'' \in \chi_G^{-1}(\chi_G(u))$. Pick $u'' = u$, then clearly $u'' \neq u'$ and $u'' \neq w$. Lemma C.45 shows that $u'$ is a cut vertex, which concludes the proof. See Figure 10 for an illustration of the above proof. $\qquad \square$

Using a similar proof technique as the one in Lemma C.47, we can prove the first bullet of Theorem 4.2. Note that we have assumed $\{\!\{\chi_G(w) : w \in \mathcal{V}\}\!\} = \{\!\{\chi_H(w) : w \in \mathcal{V}\}\!\}$. First consider the case when $|\chi_G^{-1}(\chi_G(u))| > 1$. Pick $w_H \in \chi_H^{-1}(\chi_G(w))$ where $w$ is defined in the proof of Lemma C.47. Then, there exists $u_H \in \chi_H^{-1}(\chi_G(u))$ satisfying $\mathrm{dis}_H^{\mathrm{R}}(w_H, u_H) = \mathrm{dis}_G^{\mathrm{R}}(w, u)$. Pick another node $u'_H \in \chi_H^{-1}(\chi_G(u))$, $u'_H \neq u_H$ (this is feasible as $|\chi_H^{-1}(\chi_G(u))| > 1$). Following the procedure of the above proof, we can obtain that $\mathrm{dis}_H^{\mathrm{R}}(w_H, u'') = \mathrm{dis}_H^{\mathrm{R}}(w_H, u_H) + \mathrm{dis}_H^{\mathrm{R}}(u_H, u'')$ for all $u'' \in \chi_H^{-1}(\chi_G(u))$. Therefore, $\mathrm{dis}_H^{\mathrm{R}}(w_H, u'_H) = \mathrm{dis}_H^{\mathrm{R}}(w_H, u_H) + \mathrm{dis}_H^{\mathrm{R}}(u_H, u'_H)$, implying $u_H$ is a cut vertex of $H$ by Lemma C.45.

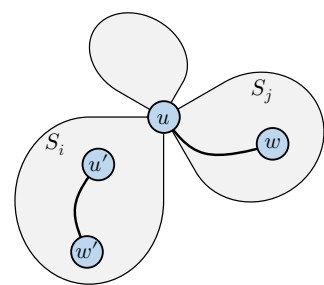

Figure 10: Illustration of the proof of Lemma C.47.

Now consider the case when $|\chi_G^{-1}(\chi_G(u))| = 1$. Then $|\chi_H^{-1}(\chi_G(u))| = 1$ and we can denote the node in $\chi_H^{-1}(\chi_G(u))$ as $u$ without abuse of notation. Choose arbitrary two nodes $w_1 \in \mathcal{S}_1$ and $w_2 \in \mathcal{S}_2$, then $\mathrm{dis}_G^{\mathrm{R}}(w_1, u) + \mathrm{dis}_G^{\mathrm{R}}(u, w_2) = \mathrm{dis}_G^{\mathrm{R}}(w_1, w_2)$ (Lemma C.45). Pick any $w'_1 \in \chi_H^{-1}(\chi_G(w_1))$ in $H$, then there exists a node $w'_2 \in \chi_H^{-1}(\chi_G(w_2))$ satisfying $\mathrm{dis}_G^{\mathrm{R}}(w_1, w_2) = \mathrm{dis}_H^{\mathrm{R}}(w'_1, w'_2)$. We also have $\mathrm{dis}_H^{\mathrm{R}}(w'_1, u) = \mathrm{dis}_G^{\mathrm{R}}(w_1, u)$ and $\mathrm{dis}_H^{\mathrm{R}}(w'_2, u) = \mathrm{dis}_G^{\mathrm{R}}(w_2, u)$ because $u$ is the unique node with color $\chi_G(u)$ in $H$. Therefore, $\mathrm{dis}_H^{\mathrm{R}}(w'_1, u) + \mathrm{dis}_H^{\mathrm{R}}(u, w'_2) = \mathrm{dis}_H^{\mathrm{R}}(w'_1, w'_2)$ and $u$ is a cut vertex in $H$ (Lemma C.45).

### C.5.2 PROOF OF THE SECOND PART

We first introduce some notations. As before, we assume $G$ and $H$ are connected and $\{\!\{\chi_G(w) : w \in \mathcal{V}\}\!\} = \{\!\{\chi_H(w) : w \in \mathcal{V}\}\!\}$. As we will consider multiple cut vertices in the following proof, we adopt the notation $\{\mathcal{S}_{G,i}(u)\}_{i=1}^{m_G(u)}$, which represents the set of connected components of graph $G$ after removing node $u$. Here, $m_G(u)$ is the number of connected components after removing node $u$, which is greater than 1 if $u$ is a cut vertex. It follows that $\bigcup_{i=1}^{m_G(u)} \mathcal{S}_{G,i}(u) = \mathcal{V} \backslash \{u\}$. We further define the index set $\mathcal{M}_G(u) := \{i \in [m_G(u)] : \mathcal{S}_{G,i}(u) \cap \chi_G^{-1}(\chi_G(u)) = \emptyset\}$. By Lemma C.46, either $|\mathcal{M}_G(u)| = m_G(u) - 1$ or $|\mathcal{M}_G(u)| = m_G(u)$.

**Lemma C.48.** *Let $u \in \mathcal{V}$ be a cut vertex of $G$. Let $u' \in \chi_H^{-1}(\chi_G(u))$, then $u'$ is also a cut vertex of $H$. Let $i \in [m_G(u)]$ and $j \in [m_H(u')]$ be two indices and pick nodes $w \in \mathcal{S}_{G,i}(u)$ and $w' \in \mathcal{S}_{H,j}(u')$. Assume $w$ and $w'$ have the same color, i.e. $\chi_G(w) = \chi_H(w')$. Then the following holds:*

- *If $i \in \mathcal{M}_G(u)$ and $j \in \mathcal{M}_H(u')$, then $\mathrm{dis}_G^{\mathrm{R}}(w, u) = \mathrm{dis}_H^{\mathrm{R}}(w', u')$.*

- *If $i \in \mathcal{M}_G(u)$ and $j \notin \mathcal{M}_H(u')$, then $\mathrm{dis}_G^{\mathrm{R}}(w, u) < \mathrm{dis}_H^{\mathrm{R}}(w', u')$.*

*Proof.* Proof of the first bullet: since $i \in \mathcal{M}_G(u)$, any path from $w$ to a node $u_G \in \chi_G^{-1}(\chi_G(u))$ goes through the cut vertex $u$, implying $\min_{u_G \in \chi_G^{-1}(\chi_G(u))} \mathrm{dis}_G^{\mathrm{R}}(w, u_G) = \mathrm{dis}_G^{\mathrm{R}}(w, u)$. Similarly,

since $j \in \mathcal{M}_H(u')$, $\min_{u_H \in \chi_H^{-1}(\chi_H(u'))} \mathrm{dis}_H^{\mathrm{R}}(w', u_H) = \mathrm{dis}_H^{\mathrm{R}}(w', u')$. Since the color of nodes $w$ and $w'$ are the same under RD-WL, we have

$$\min_{u_H \in \chi_H^{-1}(\chi_H(u'))} \mathrm{dis}_H^{\mathrm{R}}(w', u_H) = \min_{u_G \in \chi_G^{-1}(\chi_G(u))} \mathrm{dis}_G^{\mathrm{R}}(w, u_G)$$

and thus $\mathrm{dis}_H^{\mathrm{R}}(w', u') = \mathrm{dis}_G^{\mathrm{R}}(w, u)$.

Proof of the second bullet: first note that $\mathrm{dis}_H^{\mathrm{R}}(w', u') \geq \mathrm{dis}_G^{\mathrm{R}}(w, u)$ because

$$\mathrm{dis}_H^{\mathrm{R}}(w', u') \geq \min_{u_H \in \chi_H^{-1}(\chi_H(u'))} \mathrm{dis}_H^{\mathrm{R}}(w', u_H) = \min_{u_G \in \chi_G^{-1}(\chi_G(u))} \mathrm{dis}_G^{\mathrm{R}}(w, u_G) = \mathrm{dis}_G^{\mathrm{R}}(w, u).$$

If the lemma does not hold, then $\mathrm{dis}_H^{\mathrm{R}}(w', u') = \mathrm{dis}_G^{\mathrm{R}}(w, u)$. Consequently,

$$\{\!\{\mathrm{dis}_G^{\mathrm{R}}(w, u_G) : u_G \in \chi_G^{-1}(\chi_G(u))\}\!\} = \{\!\{\mathrm{dis}_G^{\mathrm{R}}(w, u) + \mathrm{dis}_G^{\mathrm{R}}(u, u_G) : u_G \in \chi_G^{-1}(\chi_G(u))\}\!\}$$
$$= \{\!\{\mathrm{dis}_H^{\mathrm{R}}(w', u') + \mathrm{dis}_H^{\mathrm{R}}(u', u_H) : u_H \in \chi_H^{-1}(\chi_H(u'))\}\!\}.$$

On the other hands,

$$\{\!\{\mathrm{dis}_G^{\mathrm{R}}(w, u_G) : u_G \in \chi_G^{-1}(\chi_G(u))\}\!\} = \{\!\{\mathrm{dis}_H^{\mathrm{R}}(w', u_H) : u_H \in \chi_H^{-1}(\chi_H(u'))\}\!\}.$$

Therefore, $\mathrm{dis}_H^{\mathrm{R}}(w', u_H) = \mathrm{dis}_H^{\mathrm{R}}(w', u') + \mathrm{dis}_H^{\mathrm{R}}(u', u_H)$ for all $u_H \in \chi_H^{-1}(\chi_H(u'))$. However, we can choose $u'' \in \chi_H^{-1}(\chi_H(u')) \cap \mathcal{S}_{H,j}(u')$ by definition of $j$, and clearly $\mathrm{dis}_H^{\mathrm{R}}(w', u'') < \mathrm{dis}_H^{\mathrm{R}}(w', u') + \mathrm{dis}_H^{\mathrm{R}}(u', u'')$ because $w'$ and $u''$ are in the same connected component (Lemma C.45). This yields a contradiction and concludes the proof. $\qquad\square$

**Corollary C.49.** *Let $u \in \mathcal{V}$ be a cut vertex of $G$. Let $u' \in \chi_H^{-1}(\chi_G(u))$, then $u'$ is also a cut vertex of $H$. Pick any $\mathcal{S}_{G,i}(u)$ and $\mathcal{S}_{H,j}(u')$ with indices $i \in \mathcal{M}_G(u)$ and $j \in \mathcal{M}_H(u')$. Then either of the following holds:*

- $\{\!\{\chi_G(w) : w \in \mathcal{S}_{G,i}(u)\}\!\} = \{\!\{\chi_H(w) : w \in \mathcal{S}_{H,j}(u')\}\!\}$.

- $\{\!\{\chi_G(w) : w \in \mathcal{S}_{G,i}(u)\}\!\} \cap \{\!\{\chi_H(w) : w \in \mathcal{S}_{H,j}(u')\}\!\} = \emptyset$.

*Proof.* Assume $\{\!\{\chi_G(w) : w \in \mathcal{S}_{G,i}(u)\}\!\} \cap \{\!\{\chi_H(w) : w \in \mathcal{S}_{H,j}(u')\}\!\} \neq \emptyset$. Then there exists nodes $w \in \mathcal{S}_{G,i}(u)$ in $G$ and $w' \in \mathcal{S}_{H,j}(u')$ in $H$, satisfying $\chi_G(w) = \chi_H(w')$. Our goal is to prove that $\{\!\{\chi_G(w) : w \in \mathcal{S}_{G,i}(u)\}\!\} = \{\!\{\chi_H(w) : w \in \mathcal{S}_{H,j}(u')\}\!\}$. It thus suffices to prove that for any color $c \in \mathcal{C}$, $|\chi_G^{-1}(c) \cap \mathcal{S}_{G,i}(u)| = |\chi_H^{-1}(c) \cap \mathcal{S}_{H,j}(u')|$.

Define $\mathcal{D}_G(w, c) = \{\!\{\mathrm{dis}_G^{\mathrm{R}}(w, x) : x \in \chi_G^{-1}(c)\}\!\}$ and define $\mathcal{D}_G(w, c) + d := \{\!\{d + d' : d' \in \mathcal{D}_G(w, c)\}\!\}$. We next claim that

$$|\chi_G^{-1}(c) \cap \mathcal{S}_{G,i}(u)| = |\chi_G^{-1}(c)| - |\mathcal{D}_G(w, c) \cap (\mathcal{D}_G(u, c) + \mathrm{dis}_G^{\mathrm{R}}(w, u))|.$$

This is simply because for any $x \in \chi_G^{-1}(c)$, either $x \in \mathcal{S}_{G,i}(u)$ or $x \notin \mathcal{S}_{G,i}(u)$. If $x \notin \mathcal{S}_{G,i}(u)$, then $\mathrm{dis}_G^{\mathrm{R}}(w, x) = \mathrm{dis}_G^{\mathrm{R}}(w, u) + \mathrm{dis}_G^{\mathrm{R}}(u, x)$ (Lemma C.45); otherwise, $\mathrm{dis}_G^{\mathrm{R}}(w, x) \neq \mathrm{dis}_G^{\mathrm{R}}(w, u) + \mathrm{dis}_G^{\mathrm{R}}(u, x)$. Similarly,

$$|\chi_H^{-1}(c) \cap \mathcal{S}_{H,j}(u')| = |\chi_H^{-1}(c)| - |\mathcal{D}_H(w', c) \cap (\mathcal{D}_H(u', c) + \mathrm{dis}_H^{\mathrm{R}}(w', u'))|.$$

Noting that $|\chi_G^{-1}(c)| = |\chi_H^{-1}(c)|$, $\mathcal{D}_G(w, c) = \mathcal{D}_H(w', c)$, and $\mathrm{dis}_G^{\mathrm{R}}(w, u) = \mathrm{dis}_H^{\mathrm{R}}(w', u')$ (Lemma C.48), we obtain $|\chi_G^{-1}(c) \cap \mathcal{S}_{G,i}(u)| = |\chi_H^{-1}(c) \cap \mathcal{S}_{H,j}(u')|$ and conclude the proof. $\qquad\square$

**Remark C.50.** *As a special case, Lemma C.48 and Corollary C.49 also hold when $G = H$. For example, Corollary C.49 implies that for any $\mathcal{S}_{G,i}(u)$ and $\mathcal{S}_{G,j}(u)$ such that $\mathcal{S}_{G,i}(u) \cap \chi_G^{-1}(\chi_G(u)) = \mathcal{S}_{G,j}(u) \cap \chi_G^{-1}(\chi_G(u)) = \emptyset$, either of the two items in Corollary C.49 holds.*

Lemma C.48 and Corollary C.49 leads to the following key corollary:

**Corollary C.51.** *Let $u \in \mathcal{V}$ be a vertex in $G$ and $u' \in \mathcal{V}$ be a vertex in $H$. If $\chi_G(u) = \chi_H(u')$, then $m_G(u) = m_H(u')$ and*

$$\{\!\{\{\!\{\chi_G(w) : w \in \mathcal{S}_{G,i}(u)\}\!\}\}\!\}_{i=1}^{m_G(u)} = \{\!\{\{\!\{\chi_H(w) : w \in \mathcal{S}_{H,i}(u')\}\!\}\}\!\}_{i=1}^{m_H(u')}.$$

*Proof.* If both $u$ and $u'$ are not cut vertices, Corollary C.51 trivially holds since $m_G(u) = m_H(u') = 1$ and $\mathcal{S}_{G,1}(u) = \mathcal{V}\backslash\{u\}$, $\mathcal{S}_{H,1}(u') = \mathcal{V}\backslash\{u'\}$. Now assume $u$ and $u'$ are both cut vertices. We first claim that

$$\{\!\{\chi_G(w) : w \in \bigcup_{i \in \mathcal{M}_G(u)} \mathcal{S}_{G,i}(u)\}\!\} = \{\!\{\chi_H(w) : w \in \bigcup_{i \in \mathcal{M}_H(u')} \mathcal{S}_{H,i}(u')\}\!\}. \tag{16}$$

To prove the claim, it suffices to prove that for each color $c \in \mathcal{C}$,

$$\left| \bigcup_{i \in \mathcal{M}_G(u)} \mathcal{S}_{G,i}(u) \cap \chi_G^{-1}(c) \right| = \left| \bigcup_{i \in \mathcal{M}_H(u')} \mathcal{S}_{H,i}(u') \cap \chi_H^{-1}(c) \right|. \tag{17}$$

Note that $|\chi_G^{-1}(c)| = |\chi_H^{-1}(c)|$. Also note that by Lemma C.48, for any two nodes $w_1 \in \bigcup_{i \in \mathcal{M}_G(u)} \mathcal{S}_{G,i}(u) \cap \chi_G^{-1}(c)$ and $w_2 \in \bigcup_{i \notin \mathcal{M}_G(u)} \mathcal{S}_{G,i}(u) \cap \chi_G^{-1}(c)$, we have $\text{dis}_G^R(u, w_1) < \text{dis}_G^R(u, w_2)$. In other words, the following two sets does not intersect:

$$\mathcal{D}_G(u, c) := \{\!\{\text{dis}_G^R(w, u) : w \in \bigcup_{i \in \mathcal{M}_G(u)} \mathcal{S}_{G,i}(u) \cap \chi_G^{-1}(c)\}\!\},$$

$$\widetilde{\mathcal{D}}_G(u, c) := \{\!\{\text{dis}_G^R(w, u) : w \in \bigcup_{i \notin \mathcal{M}_G(u)} \mathcal{S}_{G,i}(u) \cap \chi_G^{-1}(c)\}\!\}.$$

Since $\chi_G(u) = \chi_H(u')$, we have $\mathcal{D}_G(u, c) \cup \widetilde{\mathcal{D}}_G(u, c) = \mathcal{D}_H(u', c) \cup \widetilde{\mathcal{D}}_H(u', c)$. Then $\mathcal{D}_G(u, c) \cap \widetilde{\mathcal{D}}_G(u, c) = \mathcal{D}_H(u', c) \cap \widetilde{\mathcal{D}}_H(u', c) = \emptyset$ implies that $\mathcal{D}_G(u, c) = \mathcal{D}_H(u', c)$ and $\widetilde{\mathcal{D}}_G(u, c) = \widetilde{\mathcal{D}}_H(u', c)$. This proves (17) and thus (16) holds.

We next claim that

$$\{\!\{\{\!\{\chi_G(w) : w \in \mathcal{S}_{G,i}(u)\}\!\} : i \in \mathcal{M}_G(u)\}\!\} = \{\!\{\{\!\{\chi_H(w) : w \in \mathcal{S}_{H,i}(u')\}\!\} : i \in \mathcal{M}_H(u')\}\!\}. \tag{18}$$

This simply follows by using (16) and Corollary C.49. Finally, (18) already yields the desired conclusion because:

- If $|\mathcal{M}_G(u)| = m_G(u)$, then (16) implies that

$$\left| \bigcup_{i \in \mathcal{M}_H(u')} \mathcal{S}_{H,i}(u') \right| = \left| \bigcup_{i \in \mathcal{M}_G(u)} \mathcal{S}_{G,i}(u) \right| = |\mathcal{V}| - 1$$

  and thus $|\mathcal{M}_H(u')| = m_H(u')$.

- If $|\mathcal{M}_G(u)| = m_G(u) - 1$, then analogously $|\mathcal{M}_H(u')| = m_H(u) - 1$. Furthermore,

  $$\{\!\{\{\!\{\chi_G(w) : w \in \mathcal{S}_{G,i}(u)\}\!\} : i \notin \mathcal{M}_G(u)\}\!\} = \{\!\{\{\!\{\chi_G(w) : w \in \mathcal{S}_{H,i}(u')\}\!\} : i \notin \mathcal{M}_H(u')\}\!\}$$

  because $\{\!\{\chi_G(w) : w \in \mathcal{V}\backslash\{u\}\}\!\} = \{\!\{\chi_H(w) : w \in \mathcal{V}\backslash\{u'\}\}\!\}$.

In both cases, Corollary C.51 holds. $\square$

We are now ready to prove that $\{\!\{\chi_G(w) : w \in \mathcal{V}\}\!\} = \{\!\{\chi_H(w) : w \in \mathcal{V}\}\!\}$ implies $\text{BCVTree}(G) \simeq \text{BCVTree}(H)$. Recall that in a block cut-vertex tree $\text{BCVTree}(G)$, there are two types of nodes: all cut vertices of $G$, and all biconnected components of $G$. Each edge in $\text{BCVTree}(G)$ is connected between a cut vertex $u \in \mathcal{V}$ and a biconnected component $\mathcal{B} \subset \mathcal{V}$ such that $u \in \mathcal{B}$.

Given a fixed RD-WL graph representation $\mathcal{R}$, consider any graph $G = (\mathcal{V}, \mathcal{E}_G)$ satisfying $\{\!\{\chi_G(w) : w \in \mathcal{V}\}\!\} = \mathcal{R}$. First, all cut vertices of $G$ can be determined purely from $\mathcal{R}$ using the node colors. We denote the cut vertex color multiset as $\mathcal{C}^V := \{\!\{\chi_G(u) : u \text{ is a cut vertex of } G\}\!\}$. Next, the number $m_G(u)$ for each cut vertex $u$ can be determined only by its color $\chi_G(u)$ (by Corollary C.51), which is equal to the degree of node $u$ in $\text{BCVTree}(G)$. We now give a procedure to construct $\text{BCVTree}(G)$, which purely depends on $\mathcal{R}$ rather than the specific graph $G$.

We examine the multisets $\mathcal{T}(u) := \{\!\{\{\!\{\chi_G(w) : w \in \mathcal{S}_{G,i}(u)\}\!\}\}\!\}_{i=1}^{m_G(u)}$ for all cut vertices $u$, which only depends on $\mathcal{R}$ and $\chi_G(u)$ rather than the specific graph $G$ or node $u$ by Corollary C.51. See Figure 11(b) for an illustration of $\mathcal{T}_u$ for four types of cut vertices $u$. In the first step, we find all cut vertices $u$ such that $\sum_{\mathcal{S} \in \mathcal{T}(u)} \mathbf{1}[\mathcal{C}^V \cap \mathcal{S} \neq \emptyset] \leq 1$ where $\mathbf{1}[\cdot]$ is the indicator function. In other words, we find cut vertices $u$ such that there is at most one connected component $\mathcal{S}_{G,i}(u)$ that contains cut vertices. These cut vertices $u$ will serve as "leaf (cut vertex) nodes" in $\text{BCVTree}(G)$, in the sense that it connects to at most one internal node in $\text{BCVTree}(G)$. The number of BCVTree leaf

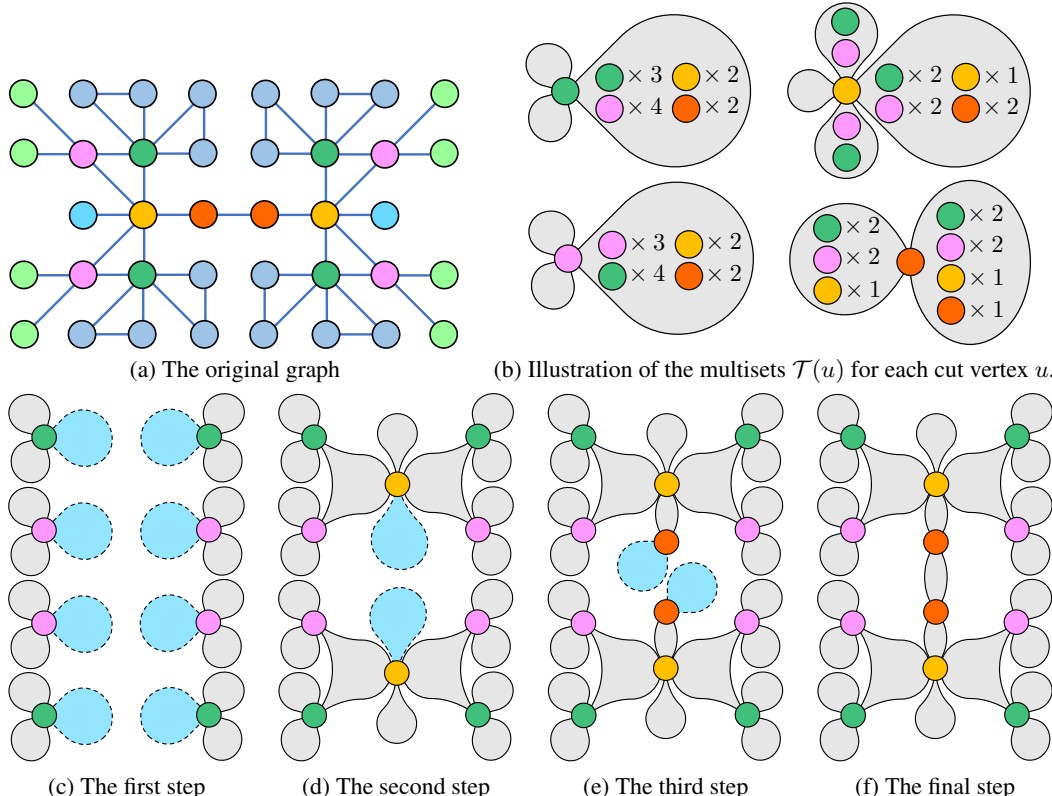

Figure 11: Illustrations for constructing the BCVTree given the graph representation $\mathcal{R}$.

nodes that connect to $u$ are also determined by Corollary C.51. See Figure 11(c) for an illustration. After finding all the "leaf (cut vertex) nodes", we can then find cut vertex nodes $v$ such that when removing all "leaf (cut vertex) nodes" in the BCVTree, $v$ will serve as a "leaf (cut vertex) node". To do this, we compute for each cut vertex $v$ and each biconnected component $\mathcal{B}_v$ associated with $v$, whether $\mathcal{B}_v$ has no cut vertex or all cut vertices in $\mathcal{B}_v$ correspond to the "leaf (cut vertex) nodes" in BCVTree($G$). Then, we check whether a cut vertex $v$ satisfies $\sum_{\mathcal{S}\in\mathcal{T}(v)} \mathbf{1}[(\mathcal{C}^{\mathrm{V}}\cap\mathcal{S})\backslash\mathcal{C}_u^{\mathrm{V}} \neq \emptyset] \leq 1$, where the set $\mathcal{C}_u^{\mathrm{V}}$ contains all colors corresponding to "leaf (cut vertex) nodes". These vertices $v$ will serve as new "leaf (cut vertex) nodes" when removing all "leaf (cut vertex) nodes" in the BCVTree, and the connection between such vertices $v$ and "leaf (cut vertex) nodes" can also be determined (see Figure 11(d) for an illustration). The procedure can be recursively executed until the full BCVTree is constructed (see Figure 11(f)), and the whole procedure does not depend on the specific graph $G$ and only depends on $\mathcal{R}$, which completes the proof.

## C.6 PROOF OF THEOREM 4.5

Given a graph $G = (\mathcal{V}, \mathcal{E})$, let $\chi_G^t$ be the 2-FWL color mapping after the $t$-th iteration (see Algorithm 2 for details), and let $\chi_G$ be the stable 2-FWL color mapping. The following result is useful for the subsequent proof:

**Lemma C.52.** *Let $u_1, u_2, v_1, v_2 \in \mathcal{V}$ be nodes in graph $G$ and $t$ be an integer. The following holds:*

- *If $\chi_G^t(u_1, v_1) = \chi_G^t(u_2, v_2)$, then $u_1 = v_1$ if and only if $u_2 = v_2$;*

- *If $\chi_G^t(u_1, v_1) = \chi_G^t(u_2, v_2)$, then $\{u_1, v_1\} \in \mathcal{E}$ if and only if $\{u_2, v_2\} \in \mathcal{E}$;*

- *If $\chi_G^t(u_1, v_1) = \chi_G^t(u_2, v_2)$ and $t \geq 1$, then $\deg_G(u_1) = \deg_G(u_2)$ and $\deg_G(v_1) = \deg_G(v_2)$.*

*Proof.* By the initial coloring (6) of 2-FWL, $\chi_G^0(u, v)$ can have the following three types of values:

$$\chi_G^0(u_1, v_1) = \begin{cases} c_{\text{same}} & \text{if } u = v \\ c_{\text{edge}} & \text{if } u \neq v \text{ and } \{u, v\} \in \mathcal{E} \\ c_{\text{other}} & \text{if } u \neq v \text{ and } \{u, v\} \notin \mathcal{E} \end{cases}$$

where $c_{\text{same}}, c_{\text{edge}}, c_{\text{other}}$ are three different colors. Therefore, if $\chi_G^0(u_1, v_1) = \chi_G^0(u_2, v_2)$, then $u_1 = v_1$ if and only if $u_2 = v_2$, and $\{u_1, v_1\} \in \mathcal{E}$ if and only if $\{u_2, v_2\} \in \mathcal{E}$. For the update step,

$$\chi_G^t(u, v) = \text{hash}\left(\chi_G^{t-1}(u, v), \{\!\!\{(\chi_G^{t-1}(u, w), \chi_G^{t-1}(w, v)) : w \in \mathcal{V}\}\!\!\}\right). \tag{19}$$

If $\chi_G^1(u_1, v_1) = \chi_G^1(u_2, v_2)$, then (19) implies that $\{\!\!\{\chi_G^0(u_1, w) : w \in \mathcal{V}\}\!\!\} = \{\!\!\{\chi_G^0(u_2, w) : w \in \mathcal{V}\}\!\!\}$ and thus $|\{\!\!\{w \in \mathcal{V} : \{u_1, w\} \in \mathcal{E}\}\!\!\}| = |\{\!\!\{w \in \mathcal{V} : \{u_2, w\} \in \mathcal{E}\}\!\!\}|$, namely $\deg_G(u_1) = \deg_G(u_2)$. We can similarly prove that $\deg_G(v_1) = \deg_G(v_2)$.

Finally, note that $\chi_G^t(u_1, v_1) = \chi_G^t(u_2, v_2)$ implies $\chi_G^{t-1}(u_1, v_1) = \chi_G^{t-1}(u_2, v_2)$ using (19). This concludes the proof of the case $t \geq 1$ by a simple induction. $\square$

For a path $P = (x_0, \cdots, x_d)$ (not necessarily simple) in graph $G$ of length $d \geq 1$, define $\omega(P) := (\deg_G(x_1), \cdots, \deg_G(x_{d-1}))$ which is a tuple of length $d - 1$. We have the following key lemma:

**Lemma C.53.** *Let $t \in \mathbb{N}$ be a non-negative integer. Given nodes $u_1, u_2, v_1, v_2 \in \mathcal{V}$, if $\chi_G^t(u_1, v_1) = \chi_G^t(u_2, v_2)$, then the following holds:*

- *Denote $\mathcal{P}_d(u, v)$ be the set of all paths (not necessarily simple) from node $u$ to node $v$ of length $d$. Then $|\mathcal{P}_{t+1}(u_1, v_1)| = |\mathcal{P}_{t+1}(u_2, v_2)|$.*

- *Denote $\mathcal{Q}_d(u, v)$ be the set of all hitting paths (not necessarily simple) from node $u$ to node $v$ of length $d$. Then, $\{\!\!\{\omega(Q) : Q \in \mathcal{Q}_{t+1}(u_1, v_1)\}\!\!\} = \{\!\!\{\omega(Q) : Q \in \mathcal{Q}_{t+1}(u_2, v_2)\}\!\!\}$, and $\{\!\!\{\omega(Q) : Q \in \mathcal{Q}_{t+1}(v_1, u_1)\}\!\!\} = \{\!\!\{\omega(Q) : Q \in \mathcal{Q}_{t+1}(v_2, u_2)\}\!\!\}$.*

*Proof.* We prove the lemma by induction over iteration $t$. We first prove the base case $t = 0$.

- If $u_1 = v_1$, then by Lemma C.52 $u_2 = v_2$. Note that obviously $|\mathcal{P}_1(u, u)| = 0$ and $|\mathcal{Q}_1(u, u)| = 0$ for any node $u$, namely $|\mathcal{P}_1(u_1, u_1)| = |\mathcal{P}_1(u_2, u_2)|$ and $\mathcal{Q}_1(u_1, u_1) = \mathcal{Q}_1(u_2, u_2) = \emptyset$.

- Similarly, if $u_1 \neq v_1$ and $\{u_1, v_1\} \notin \mathcal{E}$, then by Lemma C.52 $u_2 \neq v_2$ and $\{u_2, v_2\} \notin \mathcal{E}$. We also have $|\mathcal{P}_1(u_1, v_1)| = |\mathcal{P}_1(u_2, v_2)| = 0$ and $\mathcal{Q}_1(u_1, v_1) = \mathcal{Q}_1(u_2, v_2) = \emptyset$.

- If $u_1 \neq v_1$ and $\{u_1, v_1\} \in \mathcal{E}$, then by Lemma C.52 $u_2 \neq v_2$ and $\{u_2, v_2\} \in \mathcal{E}$. Then $|\mathcal{P}_1(u_1, v_1)| = |\mathcal{P}_1(u_2, v_2)| = 1$ and $\mathcal{Q}_1(u_1, v_1) = \mathcal{Q}_1(u_2, v_2)$ where both sets have a single element that is an empty tuple (0-dimension).

Now suppose that the conclusion of Lemma C.53 holds in iteration $t$, we will prove that it also holds in iteration $t + 1$. First note that for any two nodes $u, v$, $|\mathcal{P}_{t+1}(u, v)| = \sum_{w \in \mathcal{N}_G(v)} |\mathcal{P}_{t+1}(u, w)|$. If $\chi_G^{t+1}(u_1, v_1) = \chi_G^{t+1}(u_2, v_2)$, then by definition of 2-FWL update formula (19)

$$\{\!\!\{(\chi_G^t(u_1, w), \chi_G^t(w, v_1)) : w \in \mathcal{V}\}\!\!\} = \{\!\!\{(\chi_G^t(u_2, w), \chi_G^t(w, v_2)) : w \in \mathcal{V}\}\!\!\}.$$

which implies that $\{\!\!\{\chi_G^t(u_1, w) : w \in \mathcal{N}_G(v_1)\}\!\!\} = \{\!\!\{\chi_G^t(u_2, w) : w \in \mathcal{N}_G(v_2)\}\!\!\}$ due to Lemma C.52. Therefore,

- By induction, $\{\!\!\{|\mathcal{P}_{t+1}(u_1, w)| : w \in \mathcal{N}_G(v_1)\}\!\!\} = \{\!\!\{|\mathcal{P}_{t+1}(u_2, w)| : w \in \mathcal{N}_G(v_2)\}\!\!\}$. It follows that $\sum_{w \in \mathcal{N}_G(v_1)} |\mathcal{P}_{t+1}(u_1, w)| = \sum_{w \in \mathcal{N}_G(v_2)} |\mathcal{P}_{t+1}(u_2, w)|$ and thus we have $|\mathcal{P}_{t+2}(u_1, v_1)| = |\mathcal{P}_{t+2}(u_2, v_2)|$.

- By induction, $\{\!\!\{(\chi_G^t(u_1, w), \chi_G^t(w, v_1), \{\!\!\{\omega(Q) : Q \in \mathcal{Q}_{t+1}(w, v_1)\}\!\!\}) : w \in \mathcal{N}_G(u_1)\}\!\!\} = \{\!\!\{(\chi_G^t(u_2, w), \chi_G^t(w, v_2), \{\!\!\{\omega(Q) : Q \in \mathcal{Q}_{t+1}(w, v_2)\}\!\!\}) : w \in \mathcal{N}_G(u_2)\}\!\!\}$. Since Lemma C.52 says that $\chi_G^t(w, v) \neq \chi_G^t(v, v)$ if $w \neq v$, we have

$$\{\!\!\{(\chi_G^t(u_1, w), \{\!\!\{\omega(Q) : Q \in \mathcal{Q}_{t+1}(w, v_1)\}\!\!\}) : w \in \mathcal{N}_G(u_1) \backslash \{v_1\}\}\!\!\}$$
$$= \{\!\!\{(\chi_G^t(u_2, w), \{\!\!\{\omega(Q) : Q \in \mathcal{Q}_{t+1}(w, v_2)\}\!\!\}) : w \in \mathcal{N}_G(u_2) \backslash \{v_2\}\}\!\!\}$$

Further using the third bullet of Lemma C.52 and rearranging the two multisets yields

$$\{\!\{(\deg_G(w), \omega(Q)) : w \in \mathcal{N}_G(u_1)\backslash\{v_1\}, Q \in \mathcal{Q}_{t+1}(w, v_1)\}\!\}$$
$$=\{\!\{(\deg_G(w), \omega(Q)) : w \in \mathcal{N}_G(u_2)\backslash\{v_2\}, Q \in \mathcal{Q}_{t+1}(w, v_2)\}\!\}.$$

Equivalently, $\{\!\{\omega(Q) : Q \in \mathcal{Q}_{t+2}(u_1, v_1)\}\!\} = \{\!\{\omega(Q) : Q \in \mathcal{Q}_{t+2}(u_2, v_2)\}\!\}$. We can similarly prove that $\{\!\{\omega(Q) : Q \in \mathcal{Q}_{t+2}(v_1, u_1)\}\!\} = \{\!\{\omega(Q) : Q \in \mathcal{Q}_{t+2}(v_2, u_2)\}\!\}$.

This concludes the proof of the induction step. $\qquad\square$

The above lemma directly yields the following corollary:

**Corollary C.54.** *Given nodes* $u_1, u_2, v_1, v_2 \in \mathcal{V}$, *if* $\chi_G(u_1, v_1) = \chi_G(u_2, v_2)$, *then* $\mathrm{dis}_G(u_1, v_1) = \mathrm{dis}_G(u_2, v_2)$ *and* $\mathrm{dis}_G^R(u_1, v_1) = \mathrm{dis}_G^R(u_2, v_2)$.

*Proof.* If $\chi_G(u_1, v_1) = \chi_G(u_2, v_2)$, then $\chi_G^t(u_1, v_1) = \chi_G^t(u_2, v_2)$ holds for all $t \geq 0$. By Lemma C.53 $|\mathcal{P}_t(u_1, v_1)| = |\mathcal{P}_t(u_2, v_2)|$ holds for all $t \geq 0$ (the case $t = 0$ trivially holds). Since $\mathrm{dis}_G(u, v) = \min\{t : |\mathcal{P}_t(u_1, v_1)| > 0\}$, we conclude that $\mathrm{dis}_G(u_1, v_1) = \mathrm{dis}_G(u_2, v_2)$. As for the Resistance Distance $\mathrm{dis}_G^R$, it is equivalent to the Commute Time Distance multiplied by a constant (Chandra et al., 1996, see also Appendix E.2), i.e. $\mathrm{dis}_G^C(u, w) = 2|\mathcal{E}|\mathrm{dis}_G^R(u, w)$. Since $\mathrm{dis}_G^C(u, v) = \sum_{i=0}^{\infty} i \cdot \left(\sum_{P \in \mathcal{Q}_i(u,v)} q(P) + \sum_{P \in \mathcal{Q}_i(v,u)} q(P)\right)$ where $\mathcal{Q}_i(u, v)$ is the set containing all hitting paths of length $i$ from $u$ to $v$, and $q(P) = 1/\left(\deg_G(u)\prod_{i=1}^{d-1}\deg(x_i)\right)$ for a path $P = (x_0, \cdots, x_d)$. By Lemma C.53, we have $\sum_{P \in \mathcal{Q}_i(u_1,v_1)} q(P) = \sum_{P \in \mathcal{Q}_i(u_2,v_2)} q(P)$ and $\sum_{P \in \mathcal{Q}_i(v_1,u_1)} q(P) = \sum_{P \in \mathcal{Q}_i(v_2,u_2)} q(P)$ for all $i \geq 0$ (the case $i = 0$ trivially holds) and thus $\mathrm{dis}_G^C(u_1, v_1) = \mathrm{dis}_G^C(u_2, v_2)$, namely $\mathrm{dis}_G^R(u_1, v_1) = \mathrm{dis}_G^R(u_2, v_2)$. $\qquad\square$

We are now ready to prove Theorem 4.5.

**Theorem C.55.** *The 2-FWL algorithm is more powerful than both SPD-WL and RD-WL. Formally, given a graph G, let* $\chi_G^{\mathrm{2FWL}}$, $\chi_G^{\mathrm{SPDWL}}$ *and* $\chi_G^{\mathrm{RDWL}}$ *be the vertex color mappings for these algorithms, respectively. Then the partition induced by* $\chi_G^{\mathrm{2FWL}}$ *is finer than both* $\chi_G^{\mathrm{SPDWL}}$ *and* $\chi_G^{\mathrm{RDWL}}$.

*Proof.* Note that by definition (see Appendix B.2), we have $\chi_G(v) := \chi_G(v, v)$ for any node $v \in \mathcal{V}$. If $\chi_G(v_1) = \chi_G(v_2)$, then by definition of 2-FWL aggregation formula,

$$\{\!\{(\chi_G(v_1, w), \chi_G(w, v_1)) : w \in \mathcal{V}\}\!\} = \{\!\{(\chi_G(v_2, w), \chi_G(w, v_2)) : w \in \mathcal{V}\}\!\}.$$

Using Lemma C.6, if $\chi_G(v_1, w_1) = \chi_G(v_2, w_2)$ for some nodes $w_1$ and $w_2$, then $\chi_G(w_1) = \chi_G(w_2)$. Therefore, by using Corollary C.54 we obtain that if $\chi_G(v_1) = \chi_G(v_2)$, then

$$\{\!\{(\chi_G(w), \mathrm{dis}_G(w, v_1)) : w \in \mathcal{V}\}\!\} = \{\!\{(\chi_G(w), \mathrm{dis}_G(w, v_2)) : w \in \mathcal{V}\}\!\},$$
$$\{\!\{(\chi_G(w), \mathrm{dis}_G^R(w, v_1)) : w \in \mathcal{V}\}\!\} = \{\!\{(\chi_G(w), \mathrm{dis}_G^R(w, v_2)) : w \in \mathcal{V}\}\!\}.$$

The above equantions show that the partition induced by $\chi_G^{\mathrm{2FWL}}$ is finer than both $\chi_G^{\mathrm{SPDWL}}$ and $\chi_G^{\mathrm{RDWL}}$ and conclude the proof. $\qquad\square$

Finally, the following proposition trivially holds and will be used to prove Corollary 4.6.

**Proposition C.56.** *Given a graph* $G = (\mathcal{V}, \mathcal{E}_G)$, *let* $\chi_G$ *and* $\tilde{\chi}_G$ *be two color mappings induced by two different (general) color refinement algorithms, respectively. If the vertex partition induced by the mapping* $\chi_G$ *is finer than that of* $\tilde{\chi}_G$, *then:*

- *The mapping* $\chi_G$ *can distinguish cut vertices/edges if* $\tilde{\chi}_G$ *can distinguish cut vertices/edges;*

- *The mapping* $\chi_G$ *can distinguish the isomorphism type of* $\mathrm{BCVTree}(G)/\mathrm{BCETree}(G)$ *if* $\tilde{\chi}_G$ *can distinguish the isomorphism type of* $\mathrm{BCVTree}(G)/\mathrm{BCETree}(G)$.

Corollary 4.6 is a simple consequence of Theorem 4.5 and Proposition C.56.

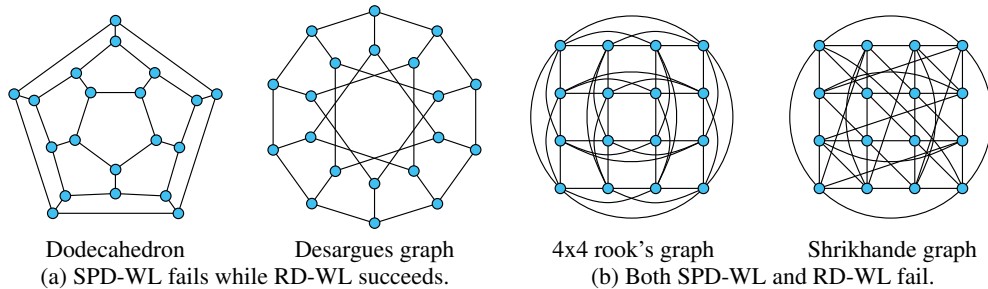

Dodecahedron          Desargues graph          4x4 rook's graph          Shrikhande graph
(a) SPD-WL fails while RD-WL succeeds.              (b) Both SPD-WL and RD-WL fail.

Figure 12: Illustration of non-isomorphic distance-regular graphs.

## C.7 REGARDING DISTANCE-REGULAR GRAPHS

In this subsection, we give more fine-grained theoretical results on the expressiveness upper bound of GD-WL by considering the special problem of distinguishing *distance-regular graphs*, a class of hard graphs that are highly relevant to the GD-WL framework. We provide a full characterization of what types of distance-regular graphs different GD-WL algorithms can or cannot distinguish, with both proofs and counterexamples.

Given a graph $G = (\mathcal{V}, \mathcal{E})$, let $\mathcal{N}_G^i(u) = \{w \in \mathcal{V} : \mathrm{dis}_G(u, w) = i\}$ be the $i$-hop neighbors of $u$ in $G$ and let $D(G) := \max_{u,v \in \mathcal{V}} \mathrm{dis}_G(u, v)$ be the diameter of $G$. We say $G$ is distance-regular if for all $i, j \in [D(G)]$ and all nodes $u, v, w, x \in \mathcal{V}$ with $\mathrm{dis}_G(u, v) = \mathrm{dis}_G(w, x)$, we have $|\mathcal{N}_G^i(u) \cap \mathcal{N}_G^j(v)| = |\mathcal{N}_G^i(w) \cap \mathcal{N}_G^j(x)|$. From the definition, it is straightforward to see that for all $u, v \in \mathcal{V}$ and $i \in [D(G)]$, $|\mathcal{N}_G^i(u)| = |\mathcal{N}_G^i(v)|$, i.e., the number of $i$-hop neighbors is the same for all nodes. We thus denote $\kappa(G) = (k_1, \cdots, k_{D(G)})$ as the $k$-hop-neighbor array where $k_i := |\mathcal{N}_G^i(u)|$ with $u \in \mathcal{V}$ chosen arbitrarily. We next define another important array:

**Definition C.57.** (**Intersection array**) The intersection array of a distance-regular graph $G$ is denoted as $\iota(G) = \{b_0, \cdots, b_{D(G)-1}; c_1, \cdots, c_{D(G)}\}$ where $b_i = |\mathcal{N}_G(u) \cap \mathcal{N}_G^{i+1}(v)|$ and $c_i = |\mathcal{N}_G(u) \cap \mathcal{N}_G^{i-1}(v)|$ with $\mathrm{dis}_G(u, v) = i$.

We now present our main results.

**Theorem C.58.** *Let $G$ and $H$ be two connected distance-regular graphs. Then the following holds:*

- *SPD-WL can distinguish the two graphs if and only if their $k$-hop-neighbor arrays differ, i.e. $\kappa(G) \neq \kappa(H)$.*

- *RD-WL can distinguish the two graphs if and only if their intersection arrays differ, i.e. $\iota(G) \neq \iota(H)$.*

- *2-FWL can distinguish the two graphs if and only if their intersection arrays differ, i.e. $\iota(G) \neq \iota(H)$.*

Theorem C.58 precisely characterizes the equivalence class of all distance-regular graphs for different types of algorithms. Combined the fact that $\iota(G) = \iota(H)$ implies $\kappa(G) = \kappa(H)$ (see e.g. van Dam et al. (2014, page 8)), we immediately arrive at the following corollary:

**Corollary C.59.** *RD-WL is strictly more powerful than SPD-WL in distinguishing non-isomorphic distance-regular graphs. Moreover, RD-WL is as powerful as 2-FWL in distinguishing non-isomorphic distance-regular graphs.*

**Counterexamples**. We provide representitive counterexamples in Figure 12 for both SPD-WL and RD-WL. In Figure 12(a), both the Dodecahedron and the Desargues graph have 20 vertices and the same $k$-hop-neighbor array $(3, 6, 6, 3, 1)$, and thus SPD-WL cannot distinguish them. However, they have the different intersection array (i.e., $\{3, 2, 1, 1, 1; 1, 1, 1, 2, 3\}$ for Dodecahedron and $\{3, 2, 2, 1, 1; 1, 1, 2, 2, 3\}$ for the Desargues graph), and thus RD-WL can distinguish them. In Figure 12(b), we make use of the well-known 4x4 rook's graph and the Shrikhande graph, both of which are strongly regular and thus distance-regular. They have the same intersection array $\{6, 3; 1, 2\}$ and thus both RD-WL and 2-FWL cannot distinguish them although they are non-isomorphic.

### C.7.1 PROOF OF THEOREM C.58

We first present a lemma that links the definition of distance-regular graph to its intersection array. The proof is based on the Bose-Mesner algebra and its association scheme, and please refer to van Dam et al. (2014, Sections 2.5 and 2.6) for details.

**Lemma C.60.** *Let $G$ and $H$ be two graphs with the same intersection array, and suppose nodes $u, v, w, x$ satisfy $\mathrm{dis}_G(u, v) = \mathrm{dis}_G(w, x)$. Then $|\mathcal{N}_G^i(u) \cap \mathcal{N}_G^j(v)| = |\mathcal{N}_H^i(w) \cap \mathcal{N}_H^j(x)|$ for all $i, j \in \mathbb{N}$.*

*Proof of the first item of Theorem C.58.* This part is straightforward. Consider the SPD-WL color mapping $\chi_G^1$ of graph $G$ after the first iteration. Then for two graphs $G, H$ with $n$ nodes, $\chi_G^1(u) = \chi_H^1(v)$ if and only if $|\mathcal{N}_G^i(u)| = |\mathcal{N}_H^i(v)|$ for all $i \in [n-1]$. Therefore, if $\kappa(G) \neq \kappa(H)$, then for any node $u$ in $G$ and $v$ in $H$, $|\mathcal{N}_G^j(u)| = |\mathcal{N}_H^j(v)|$ holds for some $j \in [\max(D(G), D(H))]$ and thus $\chi_G^1(u) \neq \chi_H^1(v)$. Namely, $\chi_G(u) \neq \chi_H(v)$ for all nodes $u$ in $G$ and $v$ in $H$, implying that SPD-WL can distinguish the two graphs. On the other hand, if $\kappa(G) = \kappa(H)$, then for any node $u$ in $G$ and $v$ in $H$ we have $\chi_G^1(u) = \chi_H^1(v)$. Similarly, $\chi_G^t(u) = \chi_H^t(v)$ for any iteration $t \in \mathbb{N}$, and thus SPD-WL cannot distinguish the two graphs. $\qquad\square$

*Proof of the second item of Theorem C.58.* The key insight is that given a distance-regular graph, the resistance distance between a pair of nodes $(u, v)$ only depends on its SPD. Formally, for any nodes $u, v, w, x$ in a distance-regular graph $G$, $\mathrm{dis}_G(u, v) = \mathrm{dis}_G(w, x)$ implies that $\mathrm{dis}_G^{\mathrm{R}}(u, v) = \mathrm{dis}_G^{\mathrm{R}}(w, x)$. Actually, we have the following stronger result:

**Theorem C.61.** *For any two nodes $u, v$ in a connected distance-regular graph $G$, $\mathrm{dis}_G^{\mathrm{R}}(u, v) = r_{\mathrm{dis}_G(u,v)}$ where the sequence $\{r_d\}_{d=0}^{D(G)}$ is recursively defined as follows:*

$$
r_d = \begin{cases} 0 & \text{if } d = 0, \\ r_{d-1} + \dfrac{2}{nk_{d-1}b_{d-1}} \displaystyle\sum_{i=d}^{D(G)} k_i & \text{if } d \in [D(G)], \end{cases} \tag{20}
$$

*where $\iota(G) = \{b_0, \cdots, b_{D(G)-1}; c_1, \cdots, c_{D(G)}\}$ is the intersection array of $G$ and $\kappa(G) = (k_1, \cdots, k_{D(G)})$ is its $k$-hop-neighbor array.*

*Proof.* Let $\mathbf{R} \in \mathbb{R}^{n \times n}$ be the RD matrix. Based on Theorem E.1, $\mathbf{R}$ can be expressed as $\mathbf{R} = \mathrm{diag}(\mathbf{M})\mathbf{1}^\top + \mathbf{1}^\top \mathrm{diag}(\mathbf{M}) - 2\mathbf{M}$, where $\mathbf{M} = \left(\mathbf{L} + \frac{1}{n}\mathbf{1}\mathbf{1}^\top\right)^{-1}$ and $\mathbf{L}$ is the graph Laplacian matrix. Now let $\widetilde{\mathbf{R}} = [r_{\mathrm{dis}_G(u,v)}]_{u,v \in \mathcal{V}}$ be the matrix with elements defined in (20). The key step is to prove that $2\mathbf{M} = c\mathbf{1}\mathbf{1}^\top - \widetilde{\mathbf{R}}$ for some $c \in \mathbb{R}$. This will yield

$$
\mathbf{R} = \frac{1}{2}\left(\mathrm{diag}(c\mathbf{1}\mathbf{1}^\top - \widetilde{\mathbf{R}})\mathbf{1}^\top + \mathbf{1}^\top\mathrm{diag}(c\mathbf{1}\mathbf{1}^\top - \widetilde{\mathbf{R}})\right) - c\mathbf{1}\mathbf{1}^\top + \widetilde{\mathbf{R}} = \widetilde{\mathbf{R}}
$$

(since $\mathrm{diag}(\widetilde{\mathbf{R}}) = \mathbf{O}$) and finish the proof.

We now prove $2\mathbf{M} = c\mathbf{1}\mathbf{1}^\top - \widetilde{\mathbf{R}}$ for some $c \in \mathbb{R}$, namely $\left(\mathbf{L} + \frac{1}{n}\mathbf{1}\mathbf{1}^\top\right)\left(c\mathbf{1}\mathbf{1}^\top - \widetilde{\mathbf{R}}\right) = 2\mathbf{I}$. Note that $\widetilde{\mathbf{R}}$ is a symmetric matrix and satisfy $\widetilde{\mathbf{R}}\mathbf{1} = c_1\mathbf{1}$ for some $c_1 \in \mathbb{R}$ because $G$ is distance-regular. Combined the fact that $\mathbf{L}\mathbf{1} = \mathbf{0}$, we have

$$
\left(\mathbf{L} + \frac{1}{n}\mathbf{1}\mathbf{1}^\top\right)\left(c\mathbf{1}\mathbf{1}^\top - \widetilde{\mathbf{R}}\right) = \left(c - \frac{c_1}{n}\right)\mathbf{1}\mathbf{1}^\top - \mathbf{L}\widetilde{\mathbf{R}}.
$$

It thus suffices to prove that $\mathbf{L}\widetilde{\mathbf{R}} = c\mathbf{1}\mathbf{1}^\top - 2\mathbf{I}$ for some $c \in \mathbb{R}$. Let us calculate each element $[\mathbf{L}\widetilde{\mathbf{R}}]_{uv}$ $(u, v \in \mathcal{V})$. We have

$$
[\mathbf{L}\widetilde{\mathbf{R}}]_{uv} = k_1 r_{\mathrm{dis}_G(u,v)} - \sum_{d=0}^{D(G)} r_d |\mathcal{N}_G(u) \cap \mathcal{N}_G^d(v)|. \tag{21}
$$

Now consider the following three cases:

- $u = v$. In this case, $r_{\operatorname{dis}_G(u,v)} = 0$ and we have

$$\sum_{d=1}^{D(G)} r_d |\mathcal{N}_G(u) \cap \mathcal{N}_G^d(v)| = r_1 k_1 = \frac{2(n-1)}{n}$$

by using $b_0 = k_1$ and $k_0 = 0$. Thus $[\mathbf{L}\widetilde{\mathbf{R}}]_{uv} = -\frac{2(n-1)}{n}$.

- $u \neq v$ and $\operatorname{dis}_G(u,v) < D(G)$. Denote $j = \operatorname{dis}_G(u,v)$. In this case, in (21) the term $\mathcal{N}_G(u) \cap \mathcal{N}_G^d(v) \neq \emptyset$ only when $d \in \{j-1, j, j+1\}$, and by definition of intersection array we have $|\mathcal{N}_G(u) \cap \mathcal{N}_G^{j-1}(v)| = c_j$, $|\mathcal{N}_G(u) \cap \mathcal{N}_G^{j+1}(v)| = b_j$, and $|\mathcal{N}_G(u) \cap \mathcal{N}_G^j(v)| = |\mathcal{N}_G(u)| - c_j - b_j = k_1 - c_j - b_j$. Therefore,

$$\begin{aligned}
[\mathbf{L}\widetilde{\mathbf{R}}]_{uv} &= k_1 r_j - r_{j-1} c_j - r_j(k_1 - b_j - c_j) - r_{j+1} b_j \\
&= c_j(r_j - r_{j-1}) + b_j(r_j - r_{j+1}) \\
&= \frac{2c_j}{nk_{j-1}b_{j-1}} \sum_{i=j}^{D(G)} k_i - \frac{2b_j}{nk_j b_j} \sum_{i=j+1}^{D(G)} k_i \\
&= \frac{2}{nk_j} \left( \sum_{i=j}^{D(G)} k_j - \sum_{i=j+1}^{D(G)} k_j \right) = \frac{2}{n},
\end{aligned}$$

where in the second last step we use the recursive relation of $r_j$, and in the last step we use the fact that $k_j = \frac{k_{j-1}b_{j-1}}{c_j}$ for any $j \in [D(G)]$ (see e.g. van Dam et al. (2014), page 8)).

- $u \neq v$ and $\operatorname{dis}_G(u,v) = D(G)$. This case is similar as the previous one. Denote $j = \operatorname{dis}_G(u,v)$, and $\mathcal{N}_G(u) \cap \mathcal{N}_G^d(v) \neq \emptyset$ only when $d \in \{j-1, j\}$. We have

$$\begin{aligned}
[\mathbf{L}\widetilde{\mathbf{R}}]_{uv} &= k_1 r_j - r_{j-1} c_j - r_j(k_1 - c_j) \\
&= c_j(r_j - r_{j-1}) \\
&= \frac{2c_j}{nk_{j-1}b_{j-1}} k_j = \frac{2}{n},
\end{aligned}$$

where we again use $k_j = \frac{k_{j-1}b_{j-1}}{c_j}$.

Combining the above three cases, we conclude that $\mathbf{L}\widetilde{\mathbf{R}} = \frac{2}{n}\mathbf{1}\mathbf{1}^\top - 2\mathbf{I}$, which finishes the proof. $\qquad\square$

We are now ready to prove the main result. Let $G = (\mathcal{V}_G, \mathcal{E}_G)$ and $H = (\mathcal{V}_H, \mathcal{E}_H)$ be two distance-regular graphs. We first prove that if $\iota(G) = \iota(H)$, then RD-WL cannot distinguish the two graphs. This is a simple consequence of Theorem C.61. Combined with the fact that $\kappa(G) = \kappa(H)$, we have $\{\operatorname{dis}_G^{\mathrm{R}}(u,w) : w \in \mathcal{V}_G\} = \{\operatorname{dis}_H^{\mathrm{R}}(v,w) : w \in \mathcal{V}_H\}$ for any nodes $u \in \mathcal{V}_G$ and $v \in \mathcal{V}_H$. Therefore, after the first iteration, the RD-WL color mappings $\chi_G^1$ and $\chi_H^1$ satisfy $\chi_G^1(u) = \chi_H^1(v)$ for all $u \in \mathcal{V}_G$ and $v \in \mathcal{V}_H$. Similarly, after the $t$-th iteration we still have $\chi_G^t(u) = \chi_H^t(v)$ for all $u \in \mathcal{V}_G$ and $v \in \mathcal{V}_H$ and thus RD-WL cannot distinguish the two graphs.

It remains to prove that if $\iota(G) \neq \iota(H)$, then RD-WL can distinguish the two graphs. First observe that in Theorem C.61, $r_i < r_j$ holds for any $i < j$. Therefore, for any nodes $u \in \mathcal{V}_G$ and $v \in \mathcal{V}_H$, $\{\operatorname{dis}_G^{\mathrm{R}}(u,w) : w \in \mathcal{V}_G\} = \{\operatorname{dis}_H^{\mathrm{R}}(v,w) : w \in \mathcal{V}_H\}$ if and only if

$$\{\!\!\{r_i(G)\}\!\!\} \times k_i(G) : i \in [D(G)]\} = \{\!\!\{r_i(H)\}\!\!\} \times k_i(H) : i \in [D(H)]\}, \qquad (22)$$

where $\{\!\!\{r\}\!\!\} \times k$ is a multiset containing $k$ repeated elements of value $r$. If $\iota(G) \neq \iota(H)$, then there exists a minimal index $d$ such that $b_i(G) = b_i(H)$ and $c_{i+1}(G) = c_{i+1}(H)$ for all $i < d$ but $b_d(G) \neq b_d(H)$ or $c_{d+1}(G) \neq c_{d+1}(H)$. It follows by Theorem C.61 that $r_i(G) = r_i(H)$ and $k_i(G) = k_i(H)$ for all $i \leq d$ but either $r_{d+1}(G) \neq r_{d+1}(H)$ (if $b_d(G) \neq b_d(H)$) or $k_{d+1}(G) \neq k_{d+1}(H)$ (if $b_d(G) = b_d(H)$ and $c_{d+1}(G) \neq c_{d+1}(H)$). Therefore, (22) does not hold and $\chi_G^1(u) \neq \chi_H^1(v)$ for any $u \in \mathcal{V}_G$ and $v \in \mathcal{V}_H$, namely, RD-WL can distinguish the two graphs. $\qquad\square$

*Proof of the third item of Theorem C.58.* First, if $\iota(G) \neq \iota(H)$, then 2-FWL can distinguish graphs $G$ and $H$. This is simply due to the fact that 2-FWL is more powerful than RD-WL (Theorem 4.5). It remains to prove that if $\iota(G) = \iota(H)$, then 2-FWL cannot distinguish graphs $G$ and $H$.

Let $\chi_G^t : \mathcal{V}_G \times \mathcal{V}_G \to \mathcal{C}$ be the 2-FWL color mapping of graph $G$ after $t$ iterations. We aim to prove that for any nodes $u, v \in \mathcal{V}_G$ and $w, x \in \mathcal{V}_H$, if $\mathrm{dis}_G(u,v) = \mathrm{dis}_G(w,x)$, then $\chi_G^t(u,v) = \chi_H^t(w,x)$ for any $t \in \mathbb{N}$. We prove it by induction. The base case of $t = 0$ trivially holds. Now suppose the case of $t$ holds and let us consider the color mapping after $t + 1$ iterations. By the 2-FWL update rule (2),

$$\chi_G^{t+1}(u,v) = \mathrm{hash}\left(\chi_G^t(u,v), \{\!\{(\chi_G^t(u,z), \chi_G^t(z,v)) : z \in \mathcal{V}_G\}\!\}\right). \tag{23}$$

It thus suffices to prove that

$$\{\!\{(\chi_G^t(u,z), \chi_G^t(z,v)) : z \in \mathcal{V}_G\}\!\} = \{\!\{(\chi_H^t(w,z), \chi_H^t(z,x)) : z \in \mathcal{V}_H\}\!\}. \tag{24}$$

By Lemma C.60, we have

$$\{\!\{(\mathrm{dis}_G(u,z), \mathrm{dis}_G(z,v)) : z \in \mathcal{V}_G\}\!\} = \{\!\{(\mathrm{dis}_H(w,z), \mathrm{dis}_H(z,x)) : z \in \mathcal{V}_H\}\!\}.$$

This already yields (24) by the induction result of iteration $t$. We thus complete the proof. $\square$

# D  FURTHER DISCUSSIONS WITH PRIOR WORKS

## D.1  KNOWN METRICS FOR MEASURING THE EXPRESSIVE POWER OF GNNS

In this subsection, we review existing metrics used in prior works to measure the expressiveness of GNNs. We will discuss the limitations of these metrics and argue why biconnectivity may serve as a more reasonable and compelling criterion in designing powerful GNN architectures.

**WL hierarchy**. Since the discovery of the relationship between MPNNs and 1-WL test (Xu et al., 2019; Morris et al., 2019), the WL hierarchy has been considered as the most standard metric to guide designing expressive GNNs. However, achieving an expressive power that matches the 2-FWL test is already highly difficult. Indeed, each iteration of the 2-FWL algorithm already requires a complexity of $\Omega(n^3)$ time and $\Theta(n^2)$ space for a graph with $n$ vertices (Immerman & Lander, 1990). Therefore, it is impossible to design expressive GNNs using this metric while maintaining its computational efficiency. Moreover, whether achieving higher-order WL expressiveness is necessary and helpful for real-world tasks has been questioned by recent works (Veličković, 2022).

**Structural metrics**. Another line of works thus sought different metrics to measure the expressive power of GNNs. Several popular choices are the ability of counting substructures (Arvind et al., 2020; Chen et al., 2020; Bouritsas et al., 2022), detecting cycles (Loukas, 2020; Vignac et al., 2020; Huang et al., 2023), calculating the graph diameter (Garg et al., 2020; Loukas, 2020) or other graph-related (combinatorial) problems (Sato et al., 2019). Yet, all these metrics have a common drawback: the corresponding problems may be *too hard* for GNNs to solve. Indeed, we show in Table 4 that solving any above task requires a computation complexity that grows super-linear w.r.t. the graph size even using advanced algorithms. Therefore, it is quite natural that standard MPNNs are not expressive for these metrics, since *no GNNs can solve these tasks while being efficient*. Consequently, instead of using GNNs to directly *learn* these metrics, these works had to use a precomputation step which can be costly in the worst case.

Table 4: The best computational complexity of known algorithms for solving different graph problems. Here $n$ and $m$ are the number of nodes and edges of a given graph, respectively.

| Metric | Complexity | Reference |
|---|---|---|
| $k$-FWL | $\Omega(n^{k+1})$ | (Immerman & Lander, 1990) |
| Counting/detecting triangles | $O(\min(n^{2.376}, m^{3/2}))$ | (Alon et al., 1997) |
| Detecting cycles of an odd length $k \geq 3$ | $O(\min(n^{2.376}, m^2))$ | (Alon et al., 1997) |
| Detecting cycles of an even length $k \geq 4$ | $O(n^2)$ | (Yuster & Zwick, 1997) |
| Calculating the graph diameter | $O(nm)$ | – |
| Detecting cut vertices | $\Theta(n + m)$ | (Tarjan, 1972) |
| Detecting cut edges | $\Theta(n + m)$ | (Tarjan, 1972) |

Due to the lack of proper metrics, most subsequent works mainly justify the expressive power of their proposed GNNs by focusing on regular graphs (Li et al., 2020; Bevilacqua et al., 2022; Bodnar

et al., 2021b; Feng et al., 2022; Velingker et al., 2022, to list a few), which hardly appear in practice. In contrast, the biconnectivity metrics proposed in this paper are different from all prior metrics, in that (i) it is a basic graph property and has significant values in both theory and applications; (i) it can be efficiently calculated with a complexity *linear* in the graph size, and thus it is reasonable to expect that these metrics should be learned by expressive GNNs.

## D.2 GNNs WITH DISTANCE ENCODING

In this subsection, we review prior works that are related to our proposed GD-WL. In the research field of expressive GNNs, the idea of incorporating distance first appeared in Li et al. (2020), where the authors mainly considered using distance encoding as *node features* and showed that distance can help distinguish regular graphs. They also considered an approach similar to $k$-hop aggregation by incorporating distance into the message-passing procedure (but without a systematic study). Zhang & Li (2021) designed a subgraph GNN that also uses (generalized) distance encoding as node features in each subgraph. Ying et al. (2021a) designed a Transformer architecture that incorporates distance information and *empirically* showed excellent performance. Very recently, Feng et al. (2022) formally studied the expressive power of $k$-hop GNNs. Yet, they still restricted the analysis to regular graphs. The concurrent work of Abboud et al. (2022) designed the shortest path network which is highly similar to our proposed SPD-WL. They showed the resulting model can alleviate the bottlenecks and over-squashing problems for MPNNs (Alon & Yahav, 2021; Topping et al., 2022) due to the increased receptive field.

Compared with prior works, our contribution lies in the following three aspects:

- We formalize the principled and more expressive GD-WL framework, which comprises SPD-WL as a special case. Our framework is theoretically clean and generalizes all prior works in a unified manner.

- We systematically and theoretically analyze the expressive power of SPD-WL for *general* graphs and highlight a fundamental advantage in distinguishing edge-biconnectivity.

- We design a Transformer-based GNN that is *provably as expressive as* GD-WL. Thus, our framework is not only for theoretical analysis, but can also be easily implemented with good empirical performance on real-world tasks.

**Discussions with the concurrent work of Velingker et al. (2022)**. After the initial submission, we became aware of a concurrent work (Velingker et al., 2022) which also explored the use of Resistance Distance to enhance the expressiveness of standard MPNNs. Here, we provide a comprehensive comparison of these two works. Overall, the main difference is that their approach incorporates RD (and several related affinity measures) into *node/edge features* (like Zhang & Li (2021)), while we combine RD to design a new *WL aggregation* procedure. As for the theoretical analysis, they only give a few toy examples of regular graphs to justify the expressive power beyond the 1-WL test, while we give a systematic analysis of the power of RD-WL for *general* graphs and point out that it is fully expressive for vertex-biconnectivity. In Velingker et al. (2022), the authors also made comparisons to SPD and conjectured that RD may have additional advantages than SPD in terms of expressiveness. In fact, this question is formally answered in our work, by proving that RD-WL is expressive for vertex-biconnectivity while SPD-WL is not. Another important contribution of our work is that we provide an *upper bound* of the expressive power of RD-WL to be 2-FWL (3-WL), which reveals the limit of incorporating RD information. We also provide a precise and complete characterization for the expressiveness of RD-WL in distinguishing distance-regular graphs, which reveals that RD-WL can *match* the power of 2-FWL in distinguishing these hard graphs.

## E IMPLEMENTATION OF GENERALIZED DISTANCE WEISFEILER-LEHMAN

In this section, we give implementation details of GD-WL and our proposed GNN architecture. We also give detailed analysis of its computation complexity. Below, assume the input graph $G = (\mathcal{V}, \mathcal{E})$ has $n$ vertices and $m$ edges.

### E.1 PREPROCESSING SHORTEST PATH DISTANCE

Shortest Path Distance can be easily calculated using the Floyd-Warshall algorithm (Floyd, 1962), which has a complexity of $\Theta(n^3)$. For sparse graphs typically encountered in practice (i.e. $m = o(n^2)$), a more clever way is to use breadth-first search that computes the distance from a given node to all other nodes in the graph. The time complexity can be improved to $\Theta(nm)$.

### E.2 PREPROCESSING RESISTANCE DISTANCE

In this subsection, we first describe several important properties of Resistance Distance. Based on these properties, we give a simple yet efficient algorithm to calculate Resistance Distance.

**Equivalence between Resistance Distance (RD) and Commute Time Distance (CTD)**. Chandra et al. (1996) established an important relationship between RD and CTD, by proving that $\mathrm{dis}_G^{\mathrm{C}}(u,v) = 2m\,\mathrm{dis}_G^{\mathrm{R}}(u,v)$ holds for any graph $G$ and any nodes $u, v \in \mathcal{V}$. Here, the Commute Time Distance is defined as $\mathrm{dis}_G^{\mathrm{C}}(u,v) := h_G(u,v) + h_G(v,u)$ where $h_G(u,v)$ is the average *hitting time* from $u$ to $v$ in a random walk. Concretely, $h_G(u,v)$ is equal to the average number of edges passed in a random walk when starting from $u$ and reaching $v$ for the first time. Mathmatically, it satisfies the following recursive relation:

$$
h_G(u,v) = \begin{cases} 0 & \text{if } u = v, \\ \infty & \text{if } u \text{ and } v \text{ are in different connected components}, \\ 1 + \frac{1}{\deg_G(u)} \sum_{w \in \mathcal{N}_G(u)} h_G(u,v) & \text{otherwise.} \end{cases}
\tag{25}
$$

The above equation can be used to calculate CTD and thus RD, as we will show later.

**Resistance Distance is a graph metric**. We say a function $d_G : \mathcal{V} \times \mathcal{V} \to \mathbb{R}$ is a graph metric if it is non-negative, positive semidefinite, symmetric, and satisfies triangular inequality. Let $G$ be a connected graph. Then Resistance Distance $\mathrm{dis}_G^{\mathrm{R}}$ is a valid graph metric because:

- (Positive semidefiniteness) $\mathrm{dis}_G^{\mathrm{R}}(u,v) \geq 0$ holds for any $u, v \in \mathcal{V}$. Moreover, $\mathrm{dis}_G^{\mathrm{R}}(u,v) = 0$ iff $u = v$.
- (Symmetry) $\mathrm{dis}_G^{\mathrm{R}}(u,v) = \mathrm{dis}_G^{\mathrm{R}}(v,u)$ holds for any $u, v \in \mathcal{V}$.
- (Triangular Inequality) For any $u, v, w \in \mathcal{V}$, $\mathrm{dis}_G^{\mathrm{R}}(u,v) + \mathrm{dis}_G^{\mathrm{R}}(v,w) \geq \mathrm{dis}_G^{\mathrm{R}}(u,w)$. This can be seen from the definition of CTD, since $\mathrm{dis}_G^{\mathrm{C}}(u,v) + \mathrm{dis}_G^{\mathrm{C}}(v,w)$ is equal to the average hitting time from $u$ to $w$ under the condition of passing node $v$, which is obviously larger than $\mathrm{dis}_G^{\mathrm{R}}(u,w)$.

**Comparing RD with SPD**. It is easy to see that RD is always no larger than SPD, i.e. $\mathrm{dis}_G^{\mathrm{R}}(u,v) \leq \mathrm{dis}_G(u,v)$. This is because for any subgraph $G'$ of $G$, we have $\mathrm{dis}_G^{\mathrm{R}}(u,v) \leq \mathrm{dis}_{G'}^{\mathrm{R}}(u,v)$, and when $G'$ is chosen to contain only the edges that belong to the shortest path between $u$ and $v$, we have $\mathrm{dis}_{G'}^{\mathrm{R}}(u,v) = \mathrm{dis}_G(u,v)$. Therefore, the range of RD is the same as SPD, i.e. $0 \leq \mathrm{dis}_G^{\mathrm{R}}(u,v) \leq n-1$. However, unlike SPD which is an integer, RD can be a general rational number. RD can thus be seen as a more fine-grained distance metric than SPD. Nevertheless, RD is still discrete and there are only finitely many possible values of $\mathrm{dis}_G^{\mathrm{R}}(u,v)$ when $n$ is fixed.

**Relationship to graph Laplacian**. We have the following theorem:

**Theorem E.1.** *Let $G = (\mathcal{V}, \mathcal{E})$ be a connected graph, $\mathcal{V} = [n]$, and let $\mathbf{L} \in \mathbb{S}^n$ be the graph Laplacian. Then*

$$
\mathrm{dis}_G^{\mathrm{R}}(i,j) = M_{i,i} + M_{j,j} - 2M_{i,j},
$$

*where $\mathbf{M} \in \mathbb{S}^n$ is a symmetric matrix defined as*

$$
\mathbf{M} = \left( \mathbf{L} + \frac{1}{n}\mathbf{1}\mathbf{1}^\top \right)^{-1}.
$$

*Proof.* Denote $\boldsymbol{d} = (\deg_G(1), \cdots, \deg_G(n))^\top$. Define the probability matrix $\mathbf{P}$ such that $P_{ij} = 0$ if $\{i,j\} \notin \mathcal{E}$ and $P_{ij} = 1/\deg_G(i)$ if $\{i,j\} \in \mathcal{E}$. Then for any $i \neq j$, (25) can be equivalently

written as

$$h(i, j) = 1 + \sum_{k=1}^{n} P_{ik} h(k, j) - P_{ij} h(j, j). \tag{26}$$

Now define a matrix $\tilde{\mathbf{H}}$ such that $\tilde{H}_{ij} = 1 + \sum_{k=1}^{n} P_{ik} \tilde{H}_{kj} - P_{ij} \tilde{H}_{jj}$, then $\tilde{H}_{ij} = h(i, j)$ for all $i \neq j$ (although $\tilde{H}_{ii} \neq 0 = h(i, i)$). $\tilde{\mathbf{H}}$ can be equivalently written as

$$\tilde{\mathbf{H}} = \mathbf{1}\mathbf{1}^\top + \mathbf{P}\tilde{\mathbf{H}} - \mathbf{P} \operatorname{diag}(\tilde{\mathbf{H}}), \tag{27}$$

where $\operatorname{diag}(\tilde{\mathbf{H}})$ is the diagnal matrix with elements $\tilde{H}_{ii}$ for $i \in [n]$.

We first calculate $\operatorname{diag}(\tilde{\mathbf{H}})$. Noting that $\boldsymbol{d}^\top \mathbf{P} = \boldsymbol{d}$, we have

$$\boldsymbol{d}^\top \tilde{\mathbf{H}} = \boldsymbol{d}^\top \mathbf{1}\mathbf{1}^\top + \boldsymbol{d}^\top (\tilde{\mathbf{H}} - \operatorname{diag}(\tilde{\mathbf{H}})),$$

and thus $\boldsymbol{d}^\top \operatorname{diag}(\tilde{\mathbf{H}}) = \boldsymbol{d}^\top \mathbf{1}\mathbf{1}^\top$, namely

$$\tilde{H}_{ii} = \frac{1}{d_i} \boldsymbol{d}^\top \mathbf{1} = \frac{2m}{d_i}. \tag{28}$$

Now define $\mathbf{H} = \tilde{\mathbf{H}} - \operatorname{diag}(\tilde{\mathbf{H}})$, then $H_{ij} = h(i, j)$ for all $i, j \in [n]$. We will calculate $\mathbf{H}$ in the following proof. We first write (27) equivalently as $\mathbf{H} + \operatorname{diag}(\tilde{\mathbf{H}}) = \mathbf{1}\mathbf{1}^\top + \mathbf{P}\mathbf{H}$. Then by multiplying $\mathbf{D}$, we have

$$\mathbf{D}(\mathbf{I} - \mathbf{P})\mathbf{H} = \mathbf{D}\mathbf{1}\mathbf{1}^\top - \mathbf{D} \operatorname{diag}(\tilde{\mathbf{H}}). \tag{29}$$

Using the fact that $\mathbf{D}(\mathbf{I} - \mathbf{P}) = \mathbf{L}$ and (28), we obtain

$$\mathbf{L}\mathbf{H} = \mathbf{D}\mathbf{1}\mathbf{1}^\top - 2m\mathbf{I}. \tag{30}$$

Next, noting that $\mathbf{L}\mathbf{1} = \mathbf{0}$, we have

$$\mathbf{L} = \left( \mathbf{L} + \frac{1}{n}\mathbf{1}\mathbf{1}^\top \right) \left( \mathbf{I} - \frac{1}{n}\mathbf{1}\mathbf{1}^\top \right). \tag{31}$$

One important property is that the matrix $\left( \mathbf{L} + \frac{1}{n}\mathbf{1}\mathbf{1}^\top \right)$ is invertible (see Gutman & Xiao (2004, Theorem 4) for a proof). Combining (30) and (31) we have

$$\left( \mathbf{I} - \frac{1}{n}\mathbf{1}\mathbf{1}^\top \right) \mathbf{H} = \left( \mathbf{L} + \frac{1}{n}\mathbf{1}\mathbf{1}^\top \right)^{-1} \left( \mathbf{D}\mathbf{1}\mathbf{1}^\top - 2m\mathbf{I} \right) = \mathbf{M} \left( \mathbf{D}\mathbf{1}\mathbf{1}^\top - 2m\mathbf{I} \right). \tag{32}$$

By taking diagonal elements and noting that $\operatorname{diag}(\mathbf{H}) = \mathbf{O}$, we otain

$$-\frac{1}{n} \operatorname{diag} \left( \mathbf{1}\mathbf{1}^\top \mathbf{H} \right) = \operatorname{diag} \left( \mathbf{M}\mathbf{D}\mathbf{1}\mathbf{1}^\top \right) - 2m \operatorname{diag} (\mathbf{M}) \tag{33}$$

Namely,

$$\frac{1}{n} \mathbf{H}^\top \mathbf{1} = -\mathbf{M}\mathbf{D}\mathbf{1} + 2m \operatorname{diag} (\mathbf{M}) \mathbf{1}. \tag{34}$$

Substituting (34) into (32) yields

$$\mathbf{H} = \mathbf{M} \left( \mathbf{D}\mathbf{1}\mathbf{1}^\top - 2m\mathbf{I} \right) - \mathbf{1}\mathbf{1}^\top \mathbf{D}\mathbf{M} + 2m\mathbf{1}\mathbf{1}^\top \operatorname{diag} (\mathbf{M}). \tag{35}$$

Therefore,

$$\mathbf{H} + \mathbf{H}^\top = 2m(\mathbf{1}\mathbf{1}^\top \operatorname{diag} (\mathbf{M}) + \operatorname{diag} (\mathbf{M}) \mathbf{1}\mathbf{1}^\top - 2\mathbf{M}). \tag{36}$$

This finally yields $\operatorname{dis}_G^{\mathrm{R}}(i, j) = \frac{1}{2m} \operatorname{dis}_G^{\mathrm{C}}(i, j) = \frac{1}{2m}(\mathbf{H} + \mathbf{H}^\top) = M_{i,i} + M_{j,j} - 2M_{i,j}$ and concludes the proof. $\square$

**Computational Complexity**. The graph Laplacian can be calculated in $O(n^2)$ time, and $\mathbf{M}$ can be calculated by matrix inversion which requires $O(n^3)$ time. Therefore, the overall computational complexity is $O(n^3)$ (or $O(n^{2.376})$ using advanced matrix multiplication algorithms).

For sparse graphs typically encountered in practice (i.e. $m = o(n^2)$), one may similarly ask whether a complexity that depends on $m$ can be achieved. We conjecture that it should be possible. Below,

we give another algorithm to calculate $\left(\mathbf{L} + \frac{1}{n}\mathbf{1}\mathbf{1}^\top\right)^{-1}$. Note that the graph Laplacian $\mathbf{L}$ can be equivalently written as $\mathbf{L} = \mathbf{E}\mathbf{E}^\top$, where $\mathbf{E} \in \mathbb{R}^{n \times m}$ is defined as

$$E_{ij} = \begin{cases} 1 & \text{if } \epsilon_j = \{i, k\} \text{ and } k > i \\ -1 & \text{if } \epsilon_j = \{i, k\} \text{ and } k < i \\ 0 & \text{if } i \notin \epsilon_j \end{cases} \tag{37}$$

where we denote $\mathcal{E} = \{\epsilon_1, \cdots, \epsilon_m\}$. Let $\mathbf{E} = [\boldsymbol{e}_1, \cdots, \boldsymbol{e}_m]$ where $\boldsymbol{e}_i \in \mathbb{R}^n$, then $\mathbf{M} = \left(\frac{1}{n}\mathbf{1}\mathbf{1}^\top + \sum_{i=1}^m \boldsymbol{e}_i \boldsymbol{e}_i^\top\right)^{-1}$. Noting that each $\boldsymbol{e}_i$ is highly sparse with only two non-zero elements. We suspect that one can obtain an $O(nm)$ complexity using techniques similar to the Sherman-Morrison-Woodbury update. We leave it as an open problem.

### E.3    TRANSFORMER-BASED IMPLEMENTATION

**Graphormer-GD**. The model is built on the Graphormer (Ying et al., 2021a) model, which use the Transformer (Vaswani et al., 2017) as the backbone network. A Transformer block consists of two layers: a self-attention layer followed by a feed-forward layer, with both layers having normalization (e.g., LayerNorm (Ba et al., 2016)) and skip connections (He et al., 2016). Denote $\mathbf{X}^{(l)} \in \mathbb{R}^{n \times d}$ as the input to the $(l+1)$-th block and define $\mathbf{X}^{(0)} = \mathbf{X}$, where $n$ is the number of nodes and $d$ is the feature dimension. For an input $\mathbf{X}^{(l)}$, the $(l+1)$-th block works as follows:

$$\mathbf{A}^h(\mathbf{X}^{(l)}) = \text{softmax}\left(\mathbf{X}^{(l)}\mathbf{W}_Q^{l,h}(\mathbf{X}^{(l)}\mathbf{W}_K^{l,h})^\top\right); \tag{38}$$

$$\hat{\mathbf{X}}^{(l)} = \mathbf{X}^{(l)} + \sum_{h=1}^H \mathbf{A}^h(\mathbf{X}^{(l)})\mathbf{X}^{(l)}\mathbf{W}_V^{l,h}\mathbf{W}_O^{l,h}; \tag{39}$$

$$\mathbf{X}^{(l+1)} = \hat{\mathbf{X}}^{(l)} + \text{GELU}(\hat{\mathbf{X}}^{(l)}\mathbf{W}_1^l)\mathbf{W}_2^l, \tag{40}$$

where $\mathbf{W}_O^{l,h} \in \mathbb{R}^{d_H \times d}$, $\mathbf{W}_Q^{l,h}, \mathbf{W}_K^{l,h}, \mathbf{W}_V^{l,h} \in \mathbb{R}^{d \times d_H}$, $\mathbf{W}_1^l \in \mathbb{R}^{d \times r}$, $\mathbf{W}_2^l \in \mathbb{R}^{r \times d}$, $H$ is the number of attention heads, $d_H$ is the dimension of each head, and $r$ is the dimension of the hidden layer. $\mathbf{A}^h(\mathbf{X})$ is usually referred to as the attention matrix.

Note that the self-attention layer and the feed-forward layer introduced in (39) and (40) do not encode any structural information of the input graph. As stated in Section 4, we incorporate the distance information into the attention layers of our Graphormer-GD model. The calculation of the attention matrix in (38) is modified as:

$$\mathbf{A}^h(\mathbf{X}^{(l)}) = \phi_1^{l,h}(\mathbf{D}) \odot \text{softmax}\left(\mathbf{X}^{(l)}\mathbf{W}_Q^{l,h}(\mathbf{X}^{(l)}\mathbf{W}_K^{l,h})^\top + \phi_2^{l,h}(\mathbf{D})\right); \tag{41}$$

where $\mathbf{D} \in \mathbb{R}^{n \times n}$ is the distance matrix such that $D_{uv} = d_G(u, v)$, $\phi_1^h$ and $\phi_2^h$ are element-wise functions applied to $\mathbf{D}$, and $\odot$ denotes the element-wise multiplication. In this way, the graph structural information can be captured by our Graphormer-GD model.

As stated in Section 4, we mainly consider two distance metrics: Shortest Path Distance $\text{dis}_G$ and Resistance Distance $\text{dis}_G^{\text{R}}$. For SPD, we follow Ying et al. (2021a) to use their shortest path distance encoding. Formally, let $\mathbf{D}^{\text{SPD}}$ be the SPD matrix such that $D_{uv}^{\text{SPD}} = \text{dis}_G(u, v)$. The function $\phi_1$ and $\phi_2$ can simply be parameterized by two learnable vectors $\boldsymbol{v}^1$ and $\boldsymbol{v}^2$, so that $\phi_1(D_{uv}^{\text{SPD}})$ is a learnable scalar corresponding to $v_{D_{uv}^{\text{SPD}}}^1$ (and similarly for $\phi_2$). If two nodes $u$ and $v$ are not in the same connected component, i.e., $D_{uv}^{\text{SPD}} = \infty$, a special learnable scalar is assigned. For RD, we use the Gaussian Basis kernels (Scholkopf et al., 1997) to encode the value since it may not be an integer. The encoded values from different Gaussian Basis kernels are concatenated and further transformed by a two-layer MLP. We integrate both the SPD encoding and the RD encoding to obtain $\phi_1^{l,h}(\mathbf{D})$ and $\phi_2^{l,h}(\mathbf{D})$. Note that these two matrices are parameterized by different sets of parameters. Following Ying et al. (2021a), we also incorporate the degree of each node in the input layer using a degree embedding.

**Relationship between Graphormer-GD and GD-WL**. As stated in Section 4, the expressive power of Graphormer-GD is at most as powerful as GD-WL. We will prove that it is actually as powerful as GD-WL under mild assumptions. We first restate the Lemma 5 from Xu et al. (2019), which shows that sum aggregators can represent injective functions over multisets.

**Lemma E.2.** *(Xu et al., 2019, Lemma 5) Assume the set $\mathcal{X}$ is countable. Then there exists a function $f : \mathcal{X} \to \mathbb{R}^n$ so that the function $h(\hat{\mathcal{X}}) := \sum_{x \in \hat{\mathcal{X}}} f(x)$ is unique for each multiset $\hat{\mathcal{X}} \subset \mathcal{X}$ of bounded size. Moreover, any multiset function $g$ can be decomposed as $g(\hat{\mathcal{X}}) = \phi(\sum_{x \in \hat{\mathcal{X}}} f(x))$ for some function $\phi$.*

We are now ready to present the detailed proof of the Theorem 4.4, which is restated as follows:

**Theorem E.3.** *Graphormer-GD is at most as powerful as GD-WL. Moreover, when choosing proper functions $\phi_1^h$ and $\phi_2^h$ and using a sufficiently large number of heads and layers, Graphormer-GD is as powerful as GD-WL.*

*Proof.* Consider all graphs with no more than $n$ nodes. The total number of possible values of both SPD and RD are thus finite and depends on $n$ (see Appendix E.2). Let

$$\mathcal{D}_n = \{(\mathrm{dis}_G(u,v), \mathrm{dis}_G^{\mathrm{R}}(u,v)) : G = (\mathcal{V}, \mathcal{E}), |\mathcal{V}| \le n, u,v \in \mathcal{V}\}$$

denote the set of all possible pairs $(\mathrm{dis}_G(u,v), \mathrm{dis}_G^{\mathrm{R}}(u,v))$. Since $\mathcal{D}_n$ is finite, we can list its elements as $\mathcal{D}_n = \{d_{G,1}, \cdots, d_{G,|\mathcal{D}_n|}\}$. Without abuse of notation, denote $d_G(u,v) = (\mathrm{dis}_G(u,v), \mathrm{dis}_G^{\mathrm{R}}(u,v))$. Then the GD-WL aggregation in (3) can be reformulated as follows:

$$\chi_G^t(v) := \mathrm{hash}\left((\chi_G^{t,1}(v), \chi_G^{t,2}(v), \cdots, \chi_G^{t,|\mathcal{D}_n|}(v))\right),$$

$$\text{where } \chi_G^{t,k}(v) := \{\!\{\chi_G^{t-1}(u) : u \in \mathcal{V}, d_G(u,v) = d_{G,k}\}\!\}. \tag{42}$$

Intuitively, this reformulation indicates that in each iteration, GD-WL updates the color of node $v$ by hashing a tuple of color multisets, where each multiset is obtained by injectively aggregating the colors of all nodes $u \in \mathcal{V}$ with certain distance configuration to node $v$. Therefore, to express GD-WL, the model suffices to update the representation of each node following the above procedure.

We show Graphormer-GD can achieve this goal. Recall that for the $h$-th head, the attention matrix is defined as $\phi_1^h(\mathbf{D}) \odot \mathrm{softmax}\left(\mathbf{X}\mathbf{W}_Q^h(\mathbf{X}\mathbf{W}_K^h)^\top + \phi_2^h(\mathbf{D})\right)$. For the function $\phi_1^h$, we define it to be the indicator function $\phi_1^h(d) := \mathbb{I}(d = d_{G,h})$. For the function $\phi_2^h$, we set it to be constant irrespective to the matrix $\mathbf{D}$. Let $\mathbf{W}_Q^h, \mathbf{W}_K^h$ be zero matrices. It can be seen that the term $\mathrm{softmax}\left(\mathbf{X}\mathbf{W}_Q^h(\mathbf{X}\mathbf{W}_K^h)^\top + \phi_2^h(\mathbf{D})\right)$ reduces to $\frac{1}{|\mathcal{V}|}\mathbf{1}\mathbf{1}^\top$, and thus for each node $v$, the output in the $h$-th attention head is the sum aggregation of representations of node $u$ satisfying $d_G(u,v) = d_{G,h}$. Formally,

$$\left[\mathbf{A}^h(\mathbf{X}^{(l)})\mathbf{X}^{(l)}\right]_v = \frac{1}{|\mathcal{V}|} \sum_{d_G(u,v)=d_{G,h}} \left[\mathbf{X}^{(l)}\right]_u.$$

Note that the constant $\frac{1}{|\mathcal{V}|}$ can be extracted with an additional head and be concatenated to the node representations. Moreover, the node representation $\mathbf{X}$ is processed via the feed-forward network in the previous layer (see (40). Thus, we can invoke Lemma E.2 and prove that the $h$-th attention head in Graphormer-GD can implement an injective aggregating function for $\{\!\{\chi_G^{t-1}(u) : u \in \mathcal{V}, d_G(u,v) = d_{G,h}\}\!\}$. Therefore, by using a sufficiently large number of attention heads, the multiset representations $\chi_G^{t,k}, k \in [|\mathcal{D}_n|]$ can be injectively obtained.

Finally, the multi-head attention defined in (39) is equivalent to first concatenating the output of each attention head and then using a linear mapping to transform the results. Thus, the concatenation is clearly an injective mapping of the tuple of multisets $\left(\chi_G^{t,1}, \chi_G^{t,2}, ..., \chi_G^{t,|\mathcal{D}_n|}\right)$. When the linear mapping has irrelational weights, the projection will also be injective. Therefore, one attention layer followed by the feed-forward network can implement the aggregation formula (42). Thus, our Graphormer-GD is able to simulate the GD-WL when using a sufficiant number of layers, which concludes the proof. □

# F EXPERIMENTAL DETAILS

## F.1 SYNTHETIC TASKS

**Data Generation and Evaluation Metrics.** We carefully design several graph generators to examine the expressive power of compared models on graph biconnectivity tasks. First, we include the

two families of graphs presented in Examples C.9 and C.10 (Appendix C.2). We further introduce a rich family of regular graphs with both cut vertices and cut edges. Each graph in this family is constructed by first randomly generating several connected components and then linking them via cut edges while simultaneously ensuring that each node has the same degree. Combining the above three families of hard graphs, we online generate data instances to train the compared models. For each data instance, the total number of nodes is upper bounded by 120. We use graph-level accuracy as the metric. That is, for each graph, the prediction of the model is considered correct only when all and only the cut vertices/edges are correctly identified. We use different seeds to repeat the experiments 5 times and report the average accuracy.

**Baselines.** We choose several baselines with their expressive power being at different levels. First, we consider classic MPNNs including GCN (Kipf & Welling, 2017), GAT (Veličković et al., 2018), and GIN (Bouritsas et al., 2022). The expressive power of these GNNs is proven to be at most as powerful as the 1-WL test (Xu et al., 2019). We also compare the Graph Substructure Network (Bouritsas et al., 2022), which extracts graph substructures to improve the expressive power of MPNNs. The substructure counts are incorporated into node features or the aggregation procedure. Lastly, we also compare the Graphormer model (Ying et al., 2021a), which achieved impressive performance in several world competitions (Ying et al., 2021b; Shi et al., 2022; Luo et al., 2022a).

**Settings.** We employ a 6-layer Graphormer-GD model. The dimension of hidden layers and feed-forward layers is set to 768. The number of Gaussian Basis kernels is set to 128. The number of attention heads is set to 64. The batch size is set to 32. We use AdamW (Kingma & Ba, 2014) as the optimizer and set its hyperparameter $\epsilon$ to 1e-8 and $(\beta_1, \beta_2)$ to (0.9, 0.999). The peak learning rate is set to 9e-5. The model is trained for 100k steps with a 6K-step warm-up stage. After the warm-up stage, the learning rate decays linearly to 0. All models are trained on 1 NVIDIA Tesla V100 GPU.

### F.2 REAL-WORLD TASKS

We conduct experiments on the popular benchmark dataset: ZINC from Benchmarking-GNNs (Dwivedi et al., 2020). It is a real-world dataset that consists of 250K molecular graphs. The task is to predict the constrained solubility of a molecule, which is an important chemical property for drug discovery. We train our models on both the ZINC-Full and ZINC-Subset (12K selected graphs following Dwivedi et al. (2020)).

**Baselines.** For a fair comparison, we set the parameter budget of the model to be around 500K following Dwivedi et al. (2020). We compare our Graphormer-GD with several competitive baselines, which mainly fall into five categories: Message Passing Neural Networks (MPNNs), High-order GNNs, Substructure-based GNNs, Subgraph GNNs, and Graph Transformers.

First, we compare several classic MPNNs including Graph Convolution Network (GCN) (Kipf & Welling, 2017), Graph Isomorphism Network (GIN) (Xu et al., 2019), Graph Attention Network (GAT) (Veličković et al., 2018), GraphSAGE (Hamilton et al., 2017) and MPNN(sum) (Gilmer et al., 2017). Besides, we also include several popularly used models. Mixture Model Network (MoNet) (Monti et al., 2017) introduces a general architecture to learn on graphs and manifolds using the Bayesian Gaussian Mixture Model. Gated Graph ConvNet (GatedGCN) considers residual connections, batch normalization, and edge gates to design an anisotropic variant of GCN. We compare the GatedGCN with positional encodings. Principal Neighborhood Aggregation (PNA) (Corso et al., 2020) combines multiple aggregators with degree-scalers.

Second, we compare two higher-order Graph Neural Networks: RingGNN (Chen et al., 2019) and 3WLGNN (Maron et al., 2019a) following Dwivedi et al. (2020). RingGNN extends the family of order-2 Graph $G$-invariant Networks without going into higher order tensors and is able to distinguish between non-isomorphic regular graphs where order-2 $G$-invariant networks provably fail. 3WLGNN uses rank-2 tensors to build the neural network and is proved to be equivalent to the 3-WL test on graph isomorphism problems.

Third, we compare two representative types of substructure-based GNNs. The Graph Substructure Network (Bouritsas et al., 2022) extracts graph substructures to improve the expressive power of MPNNs. The substructure counts are incorporated into node features or the aggregation procedure. We also compare the Cellular Isomorphism Network (Bodnar et al., 2021a), which extends theo-

retical results on Simplicial Complexes to regular Cell Complexes. Such generalization provides a powerful set of graph "lifting" transformations with a hierarchical message passing procedure.

Moreover, we compare several Subgraph GNNs. Nested Graph Neural Network (NGNN) (Zhang & Li, 2021) represents a graph with rooted subgraphs instead of rooted subtrees. It extracts a local subgraph around each node and applies a base GNN to each subgraph to learn a subgraph representation. The whole-graph representation is then obtained by pooling these subgraph representations. GNN-AK (Zhao et al., 2022) follows a similar manner to develop Subgraph GNNs with different generation policies. Equivariant Subgraph Aggregation Networks (ESAN) (Bevilacqua et al., 2022) develops a unified framework that includes per-layer aggregation across subgraphs, which are generated using pre-defined policies like edge deletion and ego-networks. Subgraph Union Network (SUN) (Frasca et al., 2022) is developed based on the symmetry analysis of a series of existing Subgraph GNNs and an upper bound on their expressive power, which theoretically unifies previous architectures and performs well across several graph representation learning benchmarks.

Last, we compare several Graph Transformer models. GraphTransformer (GT) (Dwivedi & Bresson, 2021) uses the Transformer model on graph tasks, which only aggregates the information from neighbor nodes to ensure graph sparsity, and proposes to use Laplacian eigenvector as positional encoding. Spectral Attention Network (SAN) (Kreuzer et al., 2021) uses a learned positional encoding (LPE) that can take advantage of the full Laplacian spectrum to learn the position of each node in a given graph. Graphormer (Ying et al., 2021a) develops the centrality encoding, spatial encoding, and edge encoding to incorporate the graph structure information into the Transformer model. Universal RPE (URPE) (Luo et al., 2022b) first shows that there exist continuous sequence-to-sequence functions which RPE-based Transformers cannot approximate, and develops a novel and universal attention module called Universal RPE-based Attention. The effectiveness of URPE has been verified across language and graph benchmarks (e.g., the ZINC dataset).

**Settings.** Our Graphormer-GD consists of 12 layers. The dimension of hidden layers and feed-forward layers are set to 80. The number of Gaussian Basis kernels is set to 128. The number of attention heads is set to 8. The batch size is selected from [128, 256, 512]. We use AdamW (Kingma & Ba, 2014) as the optimizer, and set its hyperparameter $\epsilon$ to 1e-8 and $(\beta_1, \beta_2)$ to (0.9, 0.999). The peak learning rate is selected from [4e-4, 5e-4]. The model is trained for 600k and 800k steps with a 60K-step warm-up stage for ZINC-Subset and ZINC-Full respectively. After the warm-up stage, the learning rate decays linearly to zero. The dropout ratio is selected from [0.0, 0.1]. The weight decay is selected from [0.0, 0.01]. All models are trained on 4 NVIDIA Tesla V100 GPUs.

### F.3 MORE TASKS

**Node-level Tasks.** We further conduct experiments on real-world node-level tasks. Following Li et al. (2020), we benchmark our model on two real-world graphs: Brazil-Airports and Europe-Airports, both of which are air traffic networks and are collected by Ackland et al. (2005) from the government websites. The nodes in each graph represent airports and each edge represents that there are commercial flights between the connected nodes. The Brazil-Airports graph has 131 nodes, 1038 edges in total and its diameter is 5. The Europe-Airports graph has 399 nodes, 5995 edges in total and its diameter is 5. The airport nodes are divided into 4 different levels according to the annual passenger flow distribution by 3 quantiles: 25%, 50%, and 75%. The task is to predict the level of each airport node. We follow Li et al. (2020) to split the nodes of each graph into train/validation/test subsets with the ratio being 0.8/0.1/0.1, respectively. The test accuracy of the best checkpoint on the validation set is reported. We use different seeds to repeat the experiments 20 times and report the average accuracy.

Following Li et al. (2020), we choose several competitive baselines including classical MPNNs (GCN, GraphSAGE, GIN), Struc2vec and Distance-encoding based GNNs (DE-GNN-SPD, DE-GNN-LP, DEA-GNN-SPD). We refer interested readers to Li et al. (2020) for detailed descriptions of baselines. For our Graphormer-GD, the dimension of hidden layers and feed-forward layers are set to 80. The number of layers is selected from [3, 6]. The number of Gaussian Basis kernels is set to 128. The number of attention heads is set to 8. The batch size is selected from [4, 8, 16, 32]. We use AdamW (Kingma & Ba, 2014) as the optimizer, and set its hyperparameter $\epsilon$ to 1e-8 and $(\beta_1, \beta_2)$ to (0.9, 0.999). The peak learning rate is selected from [2e-4, 7e-5, 4e-5]. The total number of training steps is selected from [500, 1000, 2000]. The ratio of the warm-up stage is set to 10%.

After the warm-up stage, the learning rate decays linearly to zero. The dropout ratio is selected from [0.0, 0.1, 0.5]. All models are trained on 1 NVIDIA Tesla V100 GPUs.

The results are presented in Table 5. We can see that our model outperforms these baselines on both datasets with a slightly larger variance value due to the small scale of the datasets.

Table 5: Average Accuracy on Brazil-Airports and Europe-Airports datasets. Experiments are repeated for 20 times with different seeds. We use * to indicate the best performance.

| Model | Brazil-Airports | Europe-Airports |
|---|---|---|
| GCN (Kipf & Welling, 2017) | 64.55±4.18 | 54.83±2.69 |
| GraphSAGE (Hamilton et al., 2017) | 70.65±5.33 | 56.29±3.21 |
| GIN (Xu et al., 2019) | 71.89±3.60 | 57.05±4.08 |
| Struc2vec (Ribeiro et al., 2017) | 70.88±4.26 | 57.94±4.01 |
| DE-GNN-SPD (Li et al., 2020) | 73.28±2.47 | 56.98±2.79 |
| DE-GNN-LP (Li et al., 2020) | 75.10±3.80 | 58.41±3.20 |
| DEA-GNN-SPD (Li et al., 2020) | 75.37±3.25 | 57.99±2.39 |
| Graphormer-GD (ours) | 77.69±6.39* | 59.23±4.05* |

## F.4 Efficiency Evaluation

We further conduct experiments to measure the efficiency of our approach by profiling the time cost per training epoch. We compare the efficiency of Graphormer-GD with other baselines along with the number of model parameters on the ZINC-subset from Dwivedi et al. (2020). The number of layers and the hidden dimension of our Graphormer-GD are set to 12 and 80 respectively. The number of attention heads is set to 8. The batch size is set to 128, which is the same as the settings of all baselines. We run profiling of all models on a 16GB NVIDIA Tesla V100 GPU. For all baselines, we evaluate the time costs based on the publicly available codes of Dwivedi et al. (2020) and Ying et al. (2021a). The results are presented in Table 6.

From Table 6, we can draw the following conclusions. Firstly, the efficiency of Graphormer-GD is in the same order of magnitude as classic MPNNs despite the fact that the computation complexity of Graphormer-GD is higher than MPNNs (i.e., $\Theta(n^2)$ v.s. $\Theta(n + m)$ for a graph with $n$ nodes and $m$ edges). This may be due to the high parallelizability of the Transformer layers. Secondly, Graphormer-GD is much more efficient than higher-order GNNs as reflected by the computation complexity in Table 1. Finally, Graphormer-GD is almost as efficient as the original Graphormer, since the newly introduced module to encode the Resistance Distance takes negligible additional time compared to that of the whole architecture.

Table 6: Efficiency Evaluation of different GNN models. We report the time per training epoch (seconds) as well as the number of model parameters.

| Model | # Params | Time (s) |
|---|---|---|
| GCN (Kipf & Welling, 2017) | 505,079 | 5.85 |
| GraphSAGE (Hamilton et al., 2017) | 505,341 | 6.02 |
| MoNet (Monti et al., 2017) | 504,013 | 7.19 |
| GIN (Xu et al., 2019) | 509,549 | 8.05 |
| GAT (Veličković et al., 2018) | 531,345 | 8.28 |
| GatedGCN-PE (Bresson & Laurent, 2017) | 505,011 | 10.74 |
| RingGNN (Chen et al., 2019) | 527,283 | 178.03 |
| 3WLGNN (Maron et al., 2019a) | 507,603 | 179.35 |
| Graphormer (Ying et al., 2021a) | 489,321 | 12.26 |
| Graphormer-GD (ours) | 502,793 | 12.52 |

