# OpenReview forum: "Rethinking the Expressive Power of GNNs via Graph Biconnectivity"
_ICLR.cc/2023/Conference — ICLR 2023 notable top 5%_

### Official Review · Reviewer_FHtc · 2022-10-24

**Confidence:** 4
**Clarity, Quality, Novelty And Reproducibility:** The quality, clarity and originality …
**Correctness:** 4
**Technical Novelty And Significance:** 3
**Empirical Novelty And Significance:** 3
**Recommendation:** 10

**Strength And Weaknesses:**

Pros:
1. The motivation of proposing such a metric is natural and fundamental, since graph isomorphism testing is too abstract to guide practical design of GNNs. And I agree that biconnectivity plays a key role in justifying any meaningful high-order substructures, such as those in bio-chemistry.
2. The theoretical of recent popular frameworks is pretty comprehensive. It includes almost all popular works with detailed discussion of counter-examples and proof.
3. The proposed GD-WL is efficient but still powerful enough to solve biconnectivity. It Graphormer implementation enjoys splendid performance on synthetic experiments and real-world benchmarks against strong baselines.

Con:
1. Since there are so many results in the main text, it would be great to present all these results in a comprehensive table near the introduction, to make it easier for any audience to get the message precisely.

Typo:
1. In appendix B.4, bolded ``Note`` -> ``Node``



**Summary Of The Paper:**

This work is proposing a metric to measure and compare the expressivity of GNNs - the capacity of solving vertex/edge biconnectivity. It carefully revisits several popular frameworks of designing powerful GNNs with comprehensive and rigorous analysis of their capacity on biconnectivity. Then, beyond the expressive but expensive previous works, it proposes a cheaper and powerful framework - GD-WL - with its Graphormer implementation. The proposed method is evaluated with synthetic experiments of biconnectivity and real-world benchmarks.

**Summary Of The Review:**

I would like to recommend a strong acceptance since this work carefully presents everything necessary in this framework of biconnectivity. On one hand, it is making a significant contribution regarding theoretically understanding expressivity to the GNN community. On the other hand, instead of simply presenting counter-examples on a limited range of models like some common papers these days, it presents theoretical results of the most general frameworks with proof, which would stand a role model for future works.

=== Acknowledgement ===

Due to time limit, I only read the theorems in the main text but did not check the proof in the appendix.

---

> ### Author Response · Authors · 2022-11-14
> **Response to Reviewer FHtc**
>
> We sincerely thank Reviewer FHtc for the recognition of the significance and the potential impact of our work as well as the valuable suggestion. We have followed your advice and added a comprehensive table near the introduction to list all theoretical results in a clear way (Table 1). This includes $(\mathrm{i})$ a summary of theoretical results on the expressive power of different GNN models for various biconnectivity problems; $(\mathrm{ii})$ links to relevant theorems; and $(\mathrm{iii})$ a comparison of computation/memory costs of different WL algorithms. We also thank you for pointing out the typo and we have fixed it.

---

### Official Review · Reviewer_kMJa · 2022-10-25

**Confidence:** 4
**Correctness:** 4
**Technical Novelty And Significance:** 4
**Empirical Novelty And Significance:** 3
**Recommendation:** 8

**Clarity, Quality, Novelty And Reproducibility:**

Although I have no major concerns regarding the clarity and quality of the paper, here are some questions and comments:
- Although the paper claims that the computational overhead of GD-WL is negligible, it would be helpful to report a time comparison between the proposals and the baselines (including vanilla MPNN).
- The paper only considers one molecular dataset for graph-level tasks. Since higher expressivity doesn't imply better performance on real-world problems, I'd like to see the performance of the proposed methods on more datasets, including node-level tasks.
- Any particular reason to use ZINC? Is there any connection to biconnectivity properties?
- Similarly to DE-GNN (Li et al., 2020), is the class of distance regular graphs pathological for GD-WL?
- Based on Table 3, should we expect that GD-WL cannot decide properties such as diameter, radius, conjoint cycle, and the total number of cycles? Have you proved this?

Overall, the theoretical results in the paper are not trivial, and the **technical novelty is sound**.

**Reproducibility**: The code is not available during the reviewing process, but the authors provide implementation details in the Appendix (section F).


**Strength And Weaknesses:**

**Strengths**
- The paper is well-written, and the authors did a good job reviewing the related literature.
- The introduced biconnectivity metric relates to important concepts in network science and is well-motivated.
- The paper provides theoretical insights into existing methods and represents an excellent contribution to the graph ML community.
- The proposed Transformer-based architecture is simple, relatively novel, and naturally derives from the paper's findings.

**Weaknesses**
- Very limited empirical evaluation setup.

**Summary Of The Paper:**

The paper proposes using the computation of node- and edge-level biconnectivity properties as expressivity metrics for GNNs. It analyzes four families of GNNs and finds that they fail to decide these biconnectivity metrics, except for ESAN. The paper provides insights into the power of ESAN. Based on these insights, the paper proposes a new color refinement algorithm that leverages distance between nodes to achieve expressivity in terms of the biconnectivity metrics. Experiments on the ZINC dataset show the efficacy of the proposed method in a real-world graph-level task.


**Summary Of The Review:**

The paper is well-written and consists of a solid contribution to the graph ML community. The results and insights from this paper should definitely be used in further developments in the field. My main concerns revolve around empirical evaluation. Therefore, I am recommending acceptance.

---

> ### Author Response · Authors · 2022-11-14
> **Response to Reviewer kMJa (Part 3/3)**
>
> **Regarding the time comparison of Graphormer-GD and baselines**.
>
> Thanks for the suggestion. We further conduct experiments to profile the time cost per epoch. We compare the efficiency of Graphormer-GD with other baselines along with the number of model parameters on the ZINC-subset from [2]. The number of layers and the hidden dimension of our Graphormer-GD are set to 12 and 80, respectively. The number of attention heads is set to 8. The batch size is set to 128, which is the same as the settings of all baselines. We profile all models on a 16GB NVIDIA Tesla V100 GPU. For all baselines, we evaluate their time costs based on the publicly available codes of [2] and [3]. The results are presented in Table 2.
>
> From Table 2, we can draw the following conclusions. Firstly, the efficiency of Graphormer-GD is in the same order of magnitude as classic MPNNs despite the fact that the computation complexity of Graphormer-GD is higher than MPNNs. This may be due to the high parallelizability of the Transformer layers. Secondly, Graphormer-GD is much more efficient than higher-order GNNs since the computation complexity is lower ($\Theta(n^2)$ v.s. $\Theta(n^3)$). Finally, Graphormer-GD is almost as efficient as the original Graphormer, since the newly introduced module to encode the Resistance Distance takes negligible additional time compared to that of the whole architecture.
>
> | Model                                                        | # Params | Time per Epoch (s) |
> | ------------------------------------------------------------ | -------- | ------------------ |
> | GCN                                                          | 505,079  | 5.85               |
> | GraphSAGE                                                    | 505,341  | 6.02               |
> | GIN                                                          | 509.549  | 8.05               |
> | GAT                                                          | 531,345  | 8.28               |
> | MoNet                                                        | 504,013  | 7.19               |
> | GatedGCN-PE                                                  | 505,011  | 10.74              |
> | RingGNN                                                      | 505,011  | 178.03             |
> | 3WLGNN                                                       | 527,283  | 179.35             |
> | Graphormer                                                   | 489,321  | 12.26              |
> | Graphormer-GD (ours)                                         | 502,793  | 12.52              |
> | Table 2. Efficiency evaluation of our Graphormer-GD and other GNN baselines. |          |                    |
>
>
> **Regarding the reason to use ZINC dataset.**
>
> It is a good question. There are two main reasons to use ZINC dataset for empirical evaluation. One major reason is that ZINC is a popular benchmark and the relevant baselines are quite comprehensive, which comprises most of prior GNN achitectures such as MPNN, GSN, CIN, various subgraph GNNs, and Graphormer. This enables a fair comparison between our approach and various baselines, which we believe can better demonstrate the effectiveness of our proposed method. Second, the task of ZINC dataset is to predict the constrained solubility of each molecule. It is known that the strength of intramolecular bonding determines the solubility of a solute in a given solvent ([see this page](https://chem.libretexts.org/Courses/Northern_Alberta_Institute_of_Technology/CHEM1130_Principles_in_Chemistry_I/3%3A_Chemical_Bonding._Solubility/3.8%3A_Solubility_and_Structure)). Note that the property of intramolecular bonding is highly relevant to edge-biconnectivity. Therefore, the prediction target is related to the biconnectivity of the molecular graph, which fits our work well.
>
> [1] Li, Pan, et al. "Distance encoding: Design provably more powerful neural networks for graph representation learning." Advances in Neural Information Processing Systems 33 (2020): 4465-4478.
>
> [2] Dwivedi, Vijay Prakash, et al. "Benchmarking graph neural networks." arXiv preprint arXiv:2003.00982 (2020).
>
> [3] Ying, Chengxuan, et al. "Do transformers really perform badly for graph representation?." Advances in Neural Information Processing Systems 34 (2021): 28877-28888.

---

> ### Author Response · Authors · 2022-11-14
> **Response to Reviewer kMJa (Part 2/3)**
>
> **Regarding the performance of Graphormer-GD on other datasets.**
>
> Thanks for the suggestion. Following [1], we conduct additional node-level experiments on two real-world graphs: the Brazil-Airports and the Europe-Airports, both of which are air traffic networks. The nodes in each graph represent airports and each edge represents that there are commercial flights between the connected nodes. According to the annual passenger flow distribution, the airport nodes are divided into 4 different levels. The task is to predict the level of each airport node. We follow [1] to split the nodes of each graph into train/validation/test subsets with the ratio being 0.8/0.1/0.1, respectively. The test accuracy of the best checkpoint on the validation set is reported. We use different seeds to repeat the experiments 20 times and report the average accuracy. We choose several competitive baselines and report their performance according to [1]. The results are presented in the following table.
>
>
> | Method                                                       | Bra.-Airports                       | Eur.-Airports                       |
> | ------------------------------------------------------------ | ----------------------------------- | ----------------------------------- |
> |                                                              | nodes: 131 edges: 1,038 diameter: 5 | nodes: 399 edges: 5,995 diameter: 5 |
> | GCN                                                          | 64.55±4.18                          | 54.83±2.69                          |
> | GraphSAGE                                                    | 70.65±5.33                          | 56.29±3.21                          |
> | GIN                                                          | 71.89±3.60                          | 57.05±4.08                          |
> | Struc2vec                                                    | 70.88±4.26                          | 57.94±4.01                          |
> | DE-GNN-SPD                                                   | 73.28±2.47                          | 56.98±2.79                          |
> | DE-GNN-LP                                                    | 75.10±3.80                          | 58.41±3.20                          |
> | DEA-GNN-SPD                                                  | 75.37±3.25                          | 57.99±2.39                          |
> | Graphormer-GD (ours)                                         | **77.69±6.39**                      | **59.23±4.05**                      |
> | Table 1. Averaged accuracy of Graphormer-GD on node-level tasks of Brazil and Europe Airports. |                                     |
>
> It can be seen that our model outperforms baselines on both datasets with a slightly larger variance due to the small scale of the datasets. The experimental results and detailed settings can also be found in Appendix F.3 in the updated version of our paper.
>
> Due to the time limitation and the restriction of computational resources, up to now we have just finished the above two node-level experiments. We will conduct more experiments on other graph-level and node-level datasets and add the new results to the next version of our paper.

---

> ### Author Response · Authors · 2022-11-14
> **Response to Reviewer kMJa (Part 1/3)**
>
> We sincerely thank Reviewer kMJa for the positive feedback and insightful comments. Below we would like to give detailed responses to each of your comments.
>
> **Regarding the expressive power of GD-WL on the class of Distance Regular Graphs.**
>
> Thanks for raising this valuable question. Since GD-WL crucially relies on distance information, it is interesting to study its expressiveness regarding worst-case scenarios where distance information may not help distinguish certain non-isomorphic graphs. Following your advice, we have fully figured out the power of GD-WL in distinguishing the class of distance-regular graphs during the discussion period. Below, we provide a *precise and complete* characterization of what types of distance-regular graphs SPD-WL/RD-WL/2-FWL can distinguish, with both theoretical results and counterexamples. Details and proofs can be found in the updated version of our paper (Appendix D.7).
>
> > **Theorem**: (Theorem C.58 in Appendix D.7)
> > - SPD-WL can distinguish two non-isomorphic distance-regular graphs if and only if their $k$-hop neighbor arrays differ.
> > - RD-WL can distinguish two non-isomorphic distance-regular graphs if and only if their intersection arrays differ.
> > - 2-FWL can distinguish two non-isomorphic distance-regular graphs if and only if their intersection arrays differ.
>
> The proof for RD-WL and 2-FWL is a bit involved, which uses the theory of Bose-Mesner algebra and its association scheme. Several important discussions are listed below:
> - **RD-WL is strictly more powerful than SPD-WL in distinguishing non-isomorphic distance-regular graphs**. This is due to the fact that two distance-regular graphs with the same intersection array must have the same number of $k$-hop neighbors for each $k$. The strictness can be seen from the following counterexample: $(\mathrm{i})$ the Dodecahedron and $(\mathrm{ii})$ the Desargues graph. See Figure 11 in Appendix D.7 for illustrations. Both graphs have 20 vertices and the same $k$-hop neighbor array (3, 6, 6, 3, 1), but they have different intersection arrays (i.e., {3, 2, 1, 1, 1; 1, 1, 1, 2, 3} and {3, 2, 2, 1, 1; 1, 1, 2, 2, 3}, respectively).
> - **RD-WL is as powerful as 2-FWL in distinguishing non-isomorphic distance-regular graphs**. This shows that RD-WL is highly expressive in distinguishing these hard graphs. It might be a bit counter-intuitive since RD-WL is much more efficient than 2-FWL in terms of both computation and space complexity. To our knowledge, this is the first result showing that a more efficient algorithm can *match* the expressive power of 2-FWL in distinguishing distance-regular graphs. Well-known counterexamples are the 4x4 rook’s graph and the Shrikhande graph, both of which are strongly regular with the same intersection array {6, 3; 1, 2} and cannot be distinguished by both RD-WL and 2-FWL.
>
>
> **Regarding the expressive power of GD-WL on other graph properties.**
>
> Thanks for the interesting question. Investigating the power of GD-WL through various metrics such as diameter, radius, and cycle counts is an exciting direction. As the reviewer pointed out, solving most of these tasks needs a computational complexity higher than the per-iteration cost of GD-WL, according to the table in Appendix D.2. But this does not directly imply that GD-WL must be unable to solve these problems. The key reason is that GD-WL involves a preprocessing step that calculates the pairwise distance between nodes, which has a computational cost of $O(mn)$. Such a complexity is not less than the complexity of several problems such as calculating the diameter, radius, or counting triangles of a graph. During the discussion period, we have obtained some preliminary results which show that GD-WL can indeed calculate the diameter and radius of the graph, and we will continue studying more aspects. However, since these studies go far beyond the scope of the current manuscript (i.e., the expressiveness power of GNNs via biconnectivity), we would like to leave the investigation as an independent work in the future.

---

### Official Review · Reviewer_yHMN · 2022-10-25

**Confidence:** 3
**Correctness:** 4
**Technical Novelty And Significance:** 4
**Empirical Novelty And Significance:** 3
**Recommendation:** 8

**Clarity, Quality, Novelty And Reproducibility:**

The paper is written in good quality. Although some heavy theoretical details make the paper make the paper hard to follow, I still believe it can be easily improved with minor modifications. The novelty of the paper is significant on graph theory and results on molecular graph regression is encouraging as well.

**Strength And Weaknesses:**

Strength:
1. Very interesting and innovative study on biconnectivity problem.
2. The theoretical discussion on ESAN and proposed Graphomer-GD are comprehensive and clear.

Weakness:
1. The organization of the paper is a bit hard to follow. I am not fully understand of explaining ESAN in the main paper since eventually author proposes a more efficient variant of graphomer. I suggest focus on Section 4 since section 3 and 4 seems quite parallel to me...
2. The synthetic experiment is missing some details such as number of graphs and size of the graphs. The numbers reported in Table 1 lack of number of repetitions and variations.
3. The efficiency of the proposed method. Can you share with us the amount of parameters of Graphormer-GD and other baselines ? What's the empirical running time on ZINC?

**Summary Of The Paper:**

This paper discusses the graph biconnectivity and the potential of existing GNNs on recognizing it. It first proves most of the existing GNNs and 1-WL cannot distinguish biconnectivity. The author proves ESAN are powerful enough for this problem due to the subgraph aggregation policies. Furthermore, author discusses the

**Summary Of The Review:**

I think this is a great paper discussing the capacity of graph neural network on one of the fundamental problem for graph. Despite I found the structure of the paper can be improved, I still think it's an excellent paper to the community.

---

> ### Author Response · Authors · 2022-11-14
> **Response to Reviewer yHMN**
>
> We sincerely thank Reviewer yHMN for the positive feedback and valuable suggestions. Below we would like to give detailed responses to each of your comments.
>
> **Regarding the organization of this paper and the analysis of ESAN.**
>
> Thanks for pointing out this concern. We have followed your advice (and also that of Reviewer FHtc) to improve the organization of this paper. In particular, we have given a summary of our theoretical results in a comprehensive table near the introduction for readers to grasp the message. We also revised some sentences in Section 3.2 to make our motivation clearer.
>
> We fully agree with you that the main contribution of this paper lies in Section 4, which provides deep insights into the relationship between biconnectivity and distance, and by virtue of this finding we propose and analyze an efficient Graphormer-like architecture. Yet, we would like to clarify that Section 3.2 plays an important role and is closely related to Section 4. The reasons are as follows.
>
> - **As a review of previous works**. In this work, we propose a new class of expressivity metrics. For the sake of completeness, we start from reviewing previous GNN architectures on this new metric. This is exactly what Sections 3.1 and 3.2 are about, where negative/positive results are provided separately. We believe these results can be beneficial for the GNN community.
>
> - **On Subgraph GNN**. Subgraph GNN is an emerging topic in this year with many architectures proposed. Despite the elegant design and empirical success, the expressive power of subgraph GNNs is largely unclear. Our theoretical results in Section 3.2 provide an answer to this important and fundamental research question, which we believe are helpful for future research of Subgraph GNNs.
>
> - **Technical connection to Section 4**. The theoretical analysis and proofs for ESAN are highly related to the finding in Section 4 and further justify our key result that *distance information is crucial for distinguishing biconnectivity metrics*. Technically, the proof of the first part of Theorem 3.2 (i.e., identifying cut vertices) has a similar structure to the proof of Theorem 4.2 with a series of similar lemmas. The proof of the second part (i.e., identifying cut edges) is also based on Theorem 4.1.
>
> Overall, the insights and techniques of the two parts are not quite separated and the analysis of ESAN completes our theoretical results.
>
> **Regarding the details of empirical evaluation**.
>
> Thanks for the suggestion. We have updated additional details on the empirical evaluation in the new version of our paper. For the synthetic tasks, the data generation and evaluation metrics are presented in Appendix F.1. We use various families of hard graphs such as Examples C.9 and C.10 mentioned in Section C.2 and also introduce a rich family of regular graphs. The data instances are *online generated* in a stochastic manner, and the accuracy numbers are reported when training converges. For each data instance, the number of nodes is upper bounded by 120. We use different seeds and repeat the experiments 5 times. In the updated version, we report both the average accuracy and the standard deviation as you suggested. We find that GD-WL can consistently achieve 100% accuracy over multiple runs.
>
> For the efficiency evaluation of Graphormer-GD and other GNN baselines, we further conduct experiments to profile the time cost per epoch and provide the number of model parameters of compared models on the ZINC-subset. The number of layers and the hidden dimension of our Graphormer-GD are set to 12 and 80 respectively. The number of attention heads is set to 8. The batch size is set to 128, which is the same as the settings of all baselines. We profile all the models on the same 16GB NVIDIA Tesla V100 GPU. For all baselines, we evaluate their time costs based on the publicly available codes of [1] and [2]. The results are presented in Table 1.
>
> | Model                   | # Params | Time per Epoch (s) |
> | ----------------------- | -------- | ------------------ |
> | GCN                     | 505,079  | 5.85        |
> | GraphSAGE         | 505,341  | 6.02        |
> | GIN                      | 509.549  | 8.05         |
> | GAT                       | 531,345  | 8.28       |
> | MoNet                   | 504,013  | 7.19       |
> | GatedGCN-PE       | 505,011  | 10.74     |
> | RingGNN               | 505,011  | 178.03   |
> | 3WLGNN               | 527,283  | 179.35   |
> | Graphormer           | 489,321  | 12.26      |
> | Graphormer-GD (ours)  | 502,793  | 12.52   |
> | Table 1. Efficiency Evaluation of our Graphormer-GD and other GNN baselines. |          |       |
>
> [1] Dwivedi, Vijay Prakash, et al. "Benchmarking graph neural networks." arXiv preprint arXiv:2003.00982 (2020).
>
> [2] Ying, Chengxuan, et al. "Do transformers really perform badly for graph representation?." Advances in Neural Information Processing Systems 34 (2021): 28877-28888.

---

### Author Response · Authors · 2022-11-14
**General Response: New Results & Paper Updates**

We sincerely thank all the reviewers and the area chair for their efforts in reviewing our paper. Below, we would like to highlight some new results and paper updates during the discussion period.

**New results.** Following the question raised by Reviewer kMJa, we have fully figured out the power of GD-WL in distinguishing the class of *distance-regular graphs*, which can be regarded as a worst-case analysis of GD-WL for graphs with high distance symmetry. Our new results provide additional insights into the strengths and weaknesses of different algorithms (e.g., SPD-WL, RD-WL, and even the classic 2-FWL), which we believe can further enhance this work. Below, we provide a *precise and complete* characterization of what types of distance-regular graphs SPD-WL/RD-WL/2-FWL can distinguish, with both theoretical results and counterexamples. Details and proofs can be found in the updated version of our paper (Appendix D.7).

> Theorem: (Theorem C.58 in Appendix D.7)
>
> - SPD-WL can distinguish two non-isomorphic distance-regular graphs if and only if their $k$-hop neighbor arrays differ.
> - RD-WL can distinguish two non-isomorphic distance-regular graphs if and only if their intersection arrays differ.
> - 2-FWL can distinguish two non-isomorphic distance-regular graphs if and only if their intersection arrays differ.

The proof for RD-WL and 2-FWL is a bit involved, which uses the theory of Bose-Mesner algebra and its association scheme [1]. Several important corollaries are listed below:

- **RD-WL is strictly more powerful than SPD-WL in distinguishing non-isomorphic distance-regular graphs**. This is due to the fact that two distance-regular graphs with the same intersection array must have the same number of $k$-hop neighbors for each $k$. We also provide counterexamples to show the strictness: the Dodecahedron and the Desargues graph. Both graphs have 20 vertices and see Figure 11 in Appendix D.7 for illustrations. They can be distinguished by RD-WL but not SPD-WL.
- **RD-WL is as powerful as 2-FWL in distinguishing non-isomorphic distance-regular graphs**. This shows that RD-WL is highly expressive in distinguishing these hard graphs. To our knowledge, this is the first result showing that a more efficient WL algorithm can *match* the expressive power of 2-FWL in distinguishing distance-regular graphs.

-----

**Paper updates**. Below we highlight the major updates of the revised submission.
- **Section 1**: add a comprehensive table that gives a summary of our theoretical results for readers to grasp the message. (Reviewers FHtc and yHMN)
- **Appendix C.7**: provide a systematic study of what types of distance-regular graphs SPD-WL/RD-WL/2-FWL can distinguish with both theoretical results and counterexamples. (Reviewer kMJa)
- **Appendix F.3**: add additional node-level experiments on two real-world graphs: the Brazil-Airports and the Europe-Airports. (Reviewer kMJa)
- **Appendix F.4**: add the efficiency comparison of Graphormer-GD with baselines.  (Reviewers yHMN and kMJa)
- **Appendix F.1**: add more details of the synthetic experiment.  (Reviewers yHMN)
- **Appendix D.2**: add discussions with a concurrent work (Velingker et al.,2022) [2].

[1] van Dam, Edwin R. Distance-regular graphs. arXiv preprint arXiv:1410.6294, 2014.

[2] Velingker, Ameya, et al. Affinity-aware graph networks. arXiv preprint arXiv:2206.11941, 2022.

---

### Public Comment · ~Lingxiao_Zhao1 · 2023-02-10
**Great work**

First I really like your work, it brings deeper understanding for the GNN community. As the author of GNN-AK, I also want to know that your analysis over GNN-AK+. It seems that the paper only mentions the naive version without considering contextual and distance encoding in the paper. For the GNN-AK+, shouldn't it be powerful enough for the biconnectivity problem? Hope you can describe more about this in paper.

---

> ### Author Response · Authors · 2023-02-12
> **Thanks for the question**
>
> Dear Lingxiao:
>
> Thank you for your interest in our work and raising the good question. We have further investigated the two advanced versions of GNN-AK proposed in your work during these days: the GNN-AK-ctx and the GNN-AK+. Our results are as follows:
> - By further incorporating the contextual encoding, the GNN-AK-ctx still cannot identify cut vertices. The counterexample is illustrated in Figure 2(c). The reason is precisely the same as GNN-AK, in that it uses the ego network policy without marking.
> - On the other hand, when using distance encoding, the resulting GNN-AK+ can distinguish both vertex-biconnectivity and edge-biconnectivity. This is also as expected, since distance encoding is more powerful than node marking and the contextual encoding is similar to the cross-graph aggregation in DSS-WL.
>
> We hope our response can clarify your concerns, and feel free to ask if you have other questions. Thank you!

---

### Comment · Program_Chairs · 2023-03-21
**Congratulations on winning the ICLR 2023 Outstanding Paper Award!!**

Hello authors,

Big congratulations! For more details about the award this year, please see https://blog.iclr.cc/2023/03/21/announcing-the-iclr-2023-outstanding-paper-award-recipients/

Best,

Mengdi, on behalf of ICLR 2023 PCs

---

### Decision · Program_Chairs · 2023-01-20

**Decision:**

Accept: notable-top-5%

**Justification For Why Not Higher Score:**

N/A

**Justification For Why Not Lower Score:**

The reviewers were consistent in their high scores and appreciated the contributions of this work.

**Metareview: Summary, Strengths And Weaknesses:**

This work received unanimously very high scores from all reviewers. All reviewers appreciated the theoretical contribution of analyzing GNNs from the biconnectivity perspective and the current/future practical benefits of this work. The authors further responded to remaining requests and critique of reviewers.

**Note From Pc:**

if the above contains the word "oral" or "spotlight" please see: "oral" presentation means -> notable-top-5% and "spotlight" means -> notable-top-25%. As stated in our emails, we are disassociating presentation type from AC recommendations